# Compressed Decentralized Momentum Stochastic Gradient Methods for Nonconvex Optimization

**Wei Liu**                                                      *lwdsdqqb@gmail.com*
*Department of Mathematical Sciences*
*Rensselaer Polytechnic Institute*

**Anweshit Panda**                                                 *pandaa2@rpi.edu*
*Department of Computer Science*
*Rensselaer Polytechnic Institute*

**Ujwal Pandey**                                                    *pandeu@rpi.edu*
*Department of Computer Science*
*Rensselaer Polytechnic Institute*

**Christopher Brissette**                                            *brissc@rpi.edu*
*Department of Computer Science*
*Rensselaer Polytechnic Institute*

**Yikang Shen**                                                *yikang.shn@gmail.com*
*MIT-IBM Watson AI Lab, IBM Research*

**George M. Slota**                                                 *slotag@rpi.edu*
*Department of Computer Science*
*Rensselaer Polytechnic Institute*

**Naigang Wang**                                                 *nwang@us.ibm.com*
*IBM T. J. Watson Research Center*

**Jie Chen**                                                    *chenjie@us.ibm.com*
*MIT-IBM Watson AI Lab, IBM Research*

**Yangyang Xu**[*]                                                    *xuy21@rpi.edu*
*Department of Mathematical Sciences*
*Rensselaer Polytechnic Institute*

## Abstract

In this paper, we design two compressed decentralized algorithms for solving nonconvex stochastic optimization under two different scenarios. Both algorithms adopt a momentum technique to achieve fast convergence and a message-compression technique to save communication costs. Though momentum acceleration and compressed communication have been used in literature, it is highly nontrivial to theoretically prove the effectiveness of their composition in a decentralized algorithm that can maintain the benefits of both sides, because of the need to simultaneously control the consensus error, the compression error, and the bias from the momentum gradient.

For the scenario where gradients are bounded, our proposal is a compressed decentralized adaptive method. To the best of our knowledge, this is the first decentralized adaptive stochastic gradient method with compressed communication. For the scenario of data heterogeneity without bounded gradients, our proposal is a compressed decentralized heavy-ball

---

[*]Corresponding author

method, which applies a gradient tracking technique to address the challenge of data hetero-geneity. Notably, both methods achieve an optimal convergence rate, and they can achieve linear speed up and adopt topology-independent algorithmic parameters within a certain regime of the user-specified error tolerance. Superior empirical performance is observed over state-of-the-art methods on training deep neural networks (DNNs) and Transformers.

# 1 Introduction

In this paper, we consider multi-agent nonconvex stochastic optimization problems in the form of

$$\min_{\mathbf{x} \in \mathbb{R}^d} f(\mathbf{x}) := \frac{1}{n} \sum_{i=1}^{n} f_i(\mathbf{x}), \quad \text{with } f_i(\mathbf{x}) = \mathbb{E}_{\xi_i \sim \mathcal{D}_i} \left[ F_i \left( \mathbf{x}, \xi_i \right) \right]. \tag{1}$$

Here, we suppose that there are $n$ agents on a connected graph $\mathcal{G}$; for each $i \in [n] := \{1, \ldots, n\}$, the functions $f_i$ and $F_i$ are owned and can be accessed directly only by the $i$-th agent; $\{\mathcal{D}_i\}_{i=1}^{n}$ are possibly non-i.i.d data distributions or represent heterogeneous datasets; the $n$ agents collaboratively solve (1) by communicating messages to their 1-hop neighbors on $\mathcal{G}$. Applications of training DNNs with pre-collected data can be formulated into (1), where the data $\{\mathcal{D}_i\}_{i=1}^{n}$ are collected by (or distributed onto) $n$ agents. Similarly, the data for training large language models consist of diverse corpora that are non-i.i.d. We are interested in problems that are in the form of (1) and satisfy the following structural assumption.

**Assumption 1** *In (1), for each $i \in [n]$, the function $f_i$ is $L$-smooth, i.e., $\|\nabla f_i(\mathbf{x}) - \nabla f_i(\mathbf{y})\| \leq L\|\mathbf{x} - \mathbf{y}\|$, for any $\mathbf{x}, \mathbf{y} \in \mathbb{R}^d$, and $f$ is lower bounded, i.e., $f^* \triangleq \min_{\mathbf{x}} f(\mathbf{x}) > -\infty$.*

## 1.1 Motivation and challenges

The stochastic gradient method (SGM) (Robbins & Monro, 1951; Nemirovski et al., 2009; Ghadimi & Lan, 2013; Bottou et al., 2018) and its various momentum-based or adaptive variants (Kingma & Ba, 2014; Reddi et al., 2018; Duchi et al., 2011; Xu & Xu, 2023; Cutkosky & Orabona, 2019) are now the workhorse for training DNNs and Transformers. Compared to the classic SGM, the momentum or adaptive SGMs are usually significantly faster to reach the same level of accuracy. On the other hand, for solving multi-agent stochastic optimization problems in the form of (1), decentralized SGMs have been designed and analyzed in many papers. However, most existing decentralized methods are about classic SGMs, e.g., Lian et al. (2017); Tang et al. (2018); Lian et al. (2018); Zhao et al. (2022a); Yan et al. (2023) and few about momentum or adaptive SGMs (Nazari et al., 2022; Chen et al., 2023b; Xiao et al., 2023). To save communication cost, a certain message-compression technique has been adopted in many works (Yan et al., 2023; Zhao et al., 2022a; Koloskova et al., 2019; Song et al., 2022; Taheri et al., 2020; Seide et al., 2014; Tang et al., 2019b) for distributed algorithms. It remains unknown whether a compression technique can be incorporated into a *momentum or adaptive decentralized* SGM, to simultaneously achieve fast convergence, low communication cost, and linear speedup for solving nonconvex multi-agent optimization problems.

Unlike in a centralized environment, it is difficult to keep at the same status the information held by the multiple agents in a decentralized system, due to the communication restriction. This difficulty makes it challenging to ensure the convergence or show an optimal complexity result of a momentum-based or adaptive decentralized SGM, especially when data are heterogenious across the agents.

## 1.2 Our contributions

Our contributions are three-fold. First, on solving multi-agent nonconvex stochastic optimization, we design two decentralized momentum-based SGMs with the option of message compression to save communication costs. Our first method, called DAMSCo and given in Alg. 1, is a compressed decentralized version of AMSGrad (Reddi et al., 2018) that adopts adaptive updates for fast convergence; our second method, called DaSHCo and given in Alg. 2, is designed by using the heavy-ball acceleration technique (Polyak, 1964; Ochs et al., 2014; Xu et al., 2022). To the best of our knowledge, our methods are the first *decentralized*

SGMs that adopt a *compression* technique together with a *momentum-based* or *adaptive* learning rate to accelerate empirical convergence. Compared to the decentralized adaptive methods in Chen et al. (2023b), our method DAMSCo needs only one round of communication per update and enables message compression, leading to significant higher communication efficiency while maintaining fast convergence. Though the method in Nazari et al. (2022) also needs only a single round of communication per update, it relies on full-message communication and more problematically, it does not have guaranteed convergence to stationarity for nonconvex optimization, as pointed out in Chen et al. (2023b). Compared to the gradient tracking based decentralized method in Xiao et al. (2023), our method DaSHCo is more communication efficient by using two rounds of communication (compared to three rounds in Xiao et al. (2023)) per update and enabling compressed communication.

Second, we analyze the convergence rate of DAMSCo under the assumption of bounded sample gradients — standard for analyzing adaptive SGMs (Kingma & Ba, 2014; Reddi et al., 2018; Chen et al., 2023b; 2019; Xu et al., 2023), and the assumption of nonexpansiveness of the compression error. The analysis is highly non-trivial as it needs to handle the challenges raised by the coexistence of nonconvexity, decentralization, compression, stochasticity variance, and the adaptive update. By meticulously controlling the errors caused by stochasticity, decentralization, compression, and adaptive learning rate, we show that DAMSCo can achieve a convergence rate of $O(T^{-\frac{1}{2}})$ in terms of expected norm-square of the objective's gradient and consensus deviation at all agents' local iterates, where $T$ is the number of updates. The rate is optimal for smooth nonconvex stochastic optimization and matches with the lower bound established in Arjevani et al. (2023). In addition, DAMSCo enjoys linear speed up with respect to the number $n$ of agents in a certain region of $T$, and its learning rate can be independent of the topology of the communication graph. Empirically, we observe significantly faster convergence of DAMSCo over existing decentralized adaptive SGMs on homogeneous data, in terms of the communication rounds.

Third, we analyze the convergence rate of DaSHCo in the scenario of heterogeneous data. By using the gradient tracking technique, we relax the assumption of bounded sample gradients, which is required by DAMSCo and most existing adaptive SGMs. With the nonexpansiveness condition of the compression error, we show that DaSHCo also converges to a stationary and consensual point, in expectation, at the optimal rate of $O(T^{-\frac{1}{2}})$. Similar to DAMSCo, DaSHCo achieves linear speed up with respect to the number of agents and can use topology-independent learning rate in a regime of $T$. In addition to the momentum-based acceleration, the linear speed up is another significant advantage of DaSHCo over the compressed decentralized method CDProxSGT in Yan et al. (2023). Empirically DaSHCo exhibits significantly faster convergence than CDProxSGT on both homogeneous and heterogeneous data, in particular for complicated neural architectures.

To highlight the advantage and novelty of our methods, we compare them to a few closely relevant methods in Table 1. More detailed comparisons are given in Appendix D. In addition to these theoretical highlights, the proposed methods significantly improve the training efficiency of large-scale neural networks. The theoretical and empirical findings contribute to a broader understanding of decentralized adaptive gradient techniques and may inspire further research.

## 1.3 Notations

We denote $[l] = \{1, 2, \ldots, l\}$. $\|\cdot\|$ is used for the Euclidean norm of a vector and the Frobenius norm of a matrix and $\|\mathbf{A}\|_2$ is for the spectral norm of a matrix $\mathbf{A}$. For two vectors $\boldsymbol{a}$ and $\boldsymbol{b}$ of the same size, $\frac{\boldsymbol{a}}{\boldsymbol{b}}$ and $\boldsymbol{a} \circ \boldsymbol{b}$ respectively denote the componentwise division and multiplication, and $\sqrt{\boldsymbol{c}}$ takes a componentwise square root of a nonnegative vector $\boldsymbol{c}$.

To solve (1) in a decentralized manner, we introduce a local copy of variable, denoted as $\boldsymbol{x}_i$, for the $i$-th agent for each $i \in [n]$. We let $\mathbf{X} = [\mathbf{x}_1, \mathbf{x}_2, \ldots, \mathbf{x}_n] \in \mathbb{R}^{d \times n}$ concatinate all local variables. $\mathbf{I}$ is reserved for the identity matrix and $\mathbf{1}$ for an all-one vector of appropriate size. Also, we denote

$$\nabla \mathbf{f}^t = \left[\nabla f_1\left(\mathbf{x}_1^t\right), \ldots, \nabla f_n\left(\mathbf{x}_n^t\right)\right], \quad \nabla \mathbf{F}^t = \left[\nabla F_1\left(\mathbf{x}_1^t, \xi_1^t\right), \ldots, \nabla F_n\left(\mathbf{x}_n^t, \xi_n^t\right)\right],$$

| Methods | CMP | (AG, MMT) | DH | LS | TI | CMR | strong condition |
|---|---|---|---|---|---|---|---|
| DADAM (Nazari et al., 2022) | ✗ | (✓, ✓) | ✗ | ✓ | ✓ | 1 | bounded gradient |
| DAGM (Chen et al., 2023b) | ✗ | (✓, ✓) | ✗ | ✓ | ✓ | 2 | bounded gradient |
| Choco-SGD (Koloskova et al., 2019) | ✓ | (✗, ✗) | ✗ | ✓ | ✓ | 1 | bounded gradient strong convexity |
| BEER (Zhao et al., 2022a) | ✓ | (✗, ✗) | ✓ | ✓ | ✓ | 2 | big batch |
| CDProxSGT (Yan et al., 2023) | ✓ | (✗, ✗) | ✓ | ✓ | ✓ | 2 | none |
| DAMSCo (this paper) | ✓ | (✓, ✓) | ✗ | ✓ | ✓ | 1 | bounded gradient |
| DaSHCo (this paper) | ✓ | (✗, ✓) | ✓ | ✓ | ✓ | 2 | none |

Table 1: Comparison between proposed methods and a few existing ones. "CMP" designates the use of compressed communication. In our convergence bounds, the parameter $\eta$ quantifies the compression error: smaller $\eta$ values correspond to higher compression ratios. "AG" is for whether an adaptive gradient update is used. "MMT" represents whether momentum is used. "DH" is for data heterogeneity. "LS" refers to linear speedup, which indicates that the convergence rate scales proportionally with the number of workers. Specifically, the $\frac{1}{n}$ factor in the complexity bound explicitly demonstrates this linear speedup, showing that using $n$ workers yields an $n$-fold acceleration compared to the single-worker case. "TI" stands for topology independence, which refers to the algorithm's robustness with respect to the underlying communication network. "CMR" denotes the communication round per iteration.

and we define $\mathbf{J} = \frac{\mathbf{1}\mathbf{1}^\top}{n}$ as the averaging matrix, and

$$\overline{\mathbf{x}} = \frac{1}{n}\mathbf{X}\mathbf{1}, \quad \overline{\mathbf{X}} = \mathbf{X}\mathbf{J} = \overline{\mathbf{x}}\mathbf{1}^\top, \quad \mathbf{X}_\perp = \mathbf{X}(\mathbf{I} - \mathbf{J}), \quad \overline{\nabla \mathbf{F}}^t = \frac{1}{n}\mathbf{F}^t\mathbf{1}, \quad \overline{\nabla \mathbf{f}}^t = \frac{1}{n}\mathbf{f}^t\mathbf{1}.$$

We use $\mathbb{E}_t$ for the expectation about $\{\xi_i^t\}_{i \in [n]}$ conditional on the $t$-th iterate and $\mathbb{E}$ for the full expectation. $\mathcal{Q}$ is used for a (random) compression operator and $\mathbb{E}_{\mathcal{Q}}$ for its expectation.

## 2 Related Work

In this section, we review existing works on distributed SGMs in either a centralized or decentralized setting for solving nonconvex problems.

### 2.1 Decentralized Nonadaptive Stochastic Gradient Methods

Many decentralized stochastic methods have been designed for solving problem (1), and most of them adopt nonadaptive updates, such as decentralized SGMs in Lian et al. (2017); Assran et al. (2019); Tang et al. (2018), gradient tracking-based methods (Koloskova et al., 2019; Lu et al., 2019), and momentum-based variance reduction methods (Xin et al., 2021).

A huge number of parameters are usually involved in training very large-scale neural networks, especially Transformers, and a high communication cost will incur to communicate the models or gradients among multiple agents. Consequently, reducing the communication cost is critical in designing efficient distributed methods. Techniques such as 1-bit SGD (Seide et al., 2014), SignSGD (Bernstein et al., 2018), QSGD (Alistarh et al., 2017), TernGrad (Wen et al., 2017), Random-$k$ (Stich et al., 2018), Top-$k$ (Aji & Heafield, 2017), Threshold-$v$ (Dutta et al., 2020) and ScaleCom (Chen et al., 2020a), have emerged as pivotal tools for compression, acting on either the gradients or model parameters, to achieve low communication cost. Tang et al. (2019b); Karimireddy et al. (2019) apply such a tool in a centralized setting. In decentralized scenarios, Tang et al. (2019a) applies the compression with error compensation to the communication of model parameters. Koloskova et al. (2019) proposes Choco-Gossip, which compresses a residue between the model parameter and its estimation, and it designs Choco-SGD by Choco-Gossip. BEER (Zhao et al., 2022a) and CDProxSGT (Yan et al., 2023) include gradient tracking and compress both tracked stochastic gradients and model parameters at each iteration by Choco-Gossip.

## 2.2 Centralized or Decentralized Adaptive Stochastic Gradient Methods

Three distinct categories of stochastic algorithms have emerged in literature: momentum-based, adaptive learning rate, and adaptive gradient methods that combine the ideas of the first two categories. In practice, adaptive SGMs like AdaGrad (Duchi et al., 2011), Adam (Kingma & Ba, 2014), and AMSGrad (Reddi et al., 2016) are more effective compared with a standard SGM.

Efforts have been made to incorporate the technique of adaptive gradient updates into distributed methods to achieve fast convergence. Hou et al. (2018) propose a distributed Adam for convex problems. Chen et al. (2020b); Zhao et al. (2022b) introduce locally adaptive algorithms for centralized distributed training. Chen et al. (2021) propose a quantized Adam, which considers error feedback in only one direction. Chen et al. (2023a) then presents a compressed distributed Adam that communicates the gradient error at each iteration. A centralized distributed AMSGrad is explored in Li et al. (2022), and its compressed version is presented in Wang et al. (2022).

Few explorations have been made to integrate momentum-based or adaptive gradient updates into a decentralized method, possibly because of the challenge of understanding their convergence behaviors. A decentralized ADAM (called DADAM) is developed in (Nazari et al., 2022) for online optimization. Though Nazari et al. (2022) gives regret bounds for DADAM on both convex and nonconvex problems, Chen et al. (2023b) points out that for offline nonconvex problems, DADAM may not converge to a stationary point. Then, Chen et al. (2023b) presents a framework of decentralized adaptive gradient method (DAGM), which communicates both model parameter and the second-momentum vector at each iteration. DAGM successfully integrates the use of adaptive gradient updates and decentralized communication protocols. While Nazari et al. (2022) and Chen et al. (2023b) are most closely related to our work, neither of them explore compression to reduce communication cost. Moreover, they both require boundedness of gradients and thus cannot handle data heterogeneity. In contrast, both algorithms that we propose can have the option of compressed communication, and our momentum-based method (in Alg. 2) can successfully address the challenge of data heterogeneity without requiring boundedness of gradients.

## 3 Decentralized Stochastic Methods with Compressed Communication

In this section, we present two decentralized SGMs with compressed communication for solving (1). The first method in Alg. 1 is designed for the scenario where gradients are bounded, while the second one in Alg. 2 can additionally handle data heterogeneity where gradients are unbounded.

In order for the $n$ agents to collaboratively solve (1), we let each agent maintain a local copy of the variable. For each $i \in [n]$, let $\boldsymbol{x}_i$ be the local copy by the $i$-th agent. To perform neighbor communication, we use a mixing (or called "gossip") matrix $\mathbf{W}$ that satisfies the following standard assumption.

**Assumption 2** *For the mixing matrix* $\mathbf{W}$*, it holds that (i)* $\mathbf{W}$ *is doubly stochastic, i.e.,* $\mathbf{W} \geq \mathbf{0}$*,* $\mathbf{W}\mathbf{1} = \mathbf{1}$ *and* $\mathbf{1}^\top \mathbf{W} = \mathbf{1}^\top$*; (ii)* $W_{ij} = 0$ *if* $i$ *and* $j$ *are not neighbors to each other; (iii)* $\mathrm{Null}(\mathbf{W} - \mathbf{I}) = \mathrm{span}\{\mathbf{1}\}$ *and* $\rho \triangleq \|\mathbf{W} - \mathbf{J}\|_2 < 1$*.*

Under Assumption 2(i), weighted averaging of local variables or sample gradients will be performed at each agent with its neighbors. Condition (ii) is enforced because message can be communicated directly only between immediate (a.k.a. 1-hop) neighbors. If the underlying communication graph $\mathcal{G}$ is connected, then a $\mathbf{W}$ can be chosen such that Assumption 2 is satisfied. The condition $\rho < 1$ is crucial to ensure consensus among the multiple agents through neighbor communication. Particularly, the value of $\rho$ depends on $\mathcal{G}$ and the choice of $\mathbf{W}$. Examples of $\mathbf{W}$ for different graphs are given in Koloskova et al. (2019); Nedić et al. (2018); Mancino-Ball et al. (2023). It is also shown in Xiao & Boyd (2004) that an optimal $\mathbf{W}$ can be designed subject to the constraints in Assumption 2(i), (ii) and $\mathrm{Null}(\mathbf{W} - \mathbf{I}) = \mathrm{span}\{\mathbf{1}\}$, such that $\rho$ is minimized.

### 3.1 Compressed Decentralized AMSGrad

With a mixing matrix $\mathbf{W}$ satisfying Assumption 2, we propose a compressed decentralized adaptive SGM for solving (1), where we adopt a similar adaptive update as AMSGrad (Reddi et al., 2018). The pseudocode is

shown in Alg. 1. Lines 5-6 follow AMSGrad and perform local update to the first and second momentum; Line 7 performs a local update to the model; $\underline{\mathbf{x}}_i$ is used to estimate the local model, while we compress the estimate error; Line 8 performs a neighbor communication, which can be realized through communicating the compressed vectors; see the discussions above Assumption 3. Notice that for simplicity, we take only one random sample $\xi_i^t$ at each iteration. In general, one can take a mini-batch of random samples, and all our theoretical results established in Section 4 remain valid.

---

**Algorithm 1: D**ecentralized **AMS**Grad with **Co**mpressed Communication (DAMSCo)

---

**1** **Input:** choose $\alpha > 0$, $0 < \beta_1 < 1$, $0 < \beta_2 < 1$, $\delta > 0$, $0 < \gamma \leq 1$, and a maximum number $T$ of updates,
  set $\boldsymbol{x}_i^0$, $\underline{\boldsymbol{x}}_i^0$, $\boldsymbol{m}_i^{-1}$, $\boldsymbol{g}_i^{-1}$, $\widehat{\boldsymbol{u}}_i^{-1}$, and $\boldsymbol{u}_i^{-1}$ to $\mathbf{0}$, and choose a mixing matrix $\mathbf{W}$.

**2** **for** $t = 0, 1, \cdots, T-1$ **do**

**3**   **for** *all nodes* $i \in [n]$ *in parallel* **do**

**4**     obtain one random sample $\xi_i^t$ and compute a stochastic gradient $\boldsymbol{g}_i^t \leftarrow \nabla \mathbf{F}_i(\mathbf{x}_i^t, \xi_i^t)$;

**5**     let $\boldsymbol{m}_i^t = \beta_1 \boldsymbol{m}_i^{t-1} + (1-\beta_1)\boldsymbol{g}_i^t$;

**6**     let $\widehat{\boldsymbol{u}}_i^t = \beta_2 \widehat{\boldsymbol{u}}_i^{t-1} + (1-\beta_2)\boldsymbol{g}_i^t \circ \boldsymbol{g}_i^t$ and $\boldsymbol{u}_i^t = \max\{\boldsymbol{u}_i^{t-1}, \widehat{\boldsymbol{u}}_i^t\}$;

**7**     update $\boldsymbol{x}_i^{t+\frac{1}{2}} = \boldsymbol{x}_i^t - \alpha \frac{\boldsymbol{m}_i^t}{\sqrt{\boldsymbol{u}_i^t + \delta}}$ and set $\underline{\boldsymbol{x}}_i^{t+1} = \underline{\boldsymbol{x}}_i^t + \mathcal{Q}[\boldsymbol{x}_i^{t+\frac{1}{2}} - \underline{\boldsymbol{x}}_i^t]$;

**8**     let $\boldsymbol{x}_i^{t+1} = \boldsymbol{x}_i^{t+\frac{1}{2}} + \gamma(\sum_{j=1}^n \mathbf{W}_{ji}\underline{\boldsymbol{x}}_j^{t+1} - \underline{\boldsymbol{x}}_i^{t+1})$.

---

Alg. 1 is the first decentralized adaptive SGM that incorporates compressed communication. Previous methods have either focused on adaptive updates with full-message communication (Chen et al., 2023b; Nazari et al., 2022) or have applied compression strategies to non-adaptive algorithms (Yan et al., 2023; Koloskova et al., 2019).

When $\mathcal{Q} = \mathbf{I}$, Alg. 1 reduces to a non-compressed decentralized version of AMSGrad. Note that even in this special non-compressed case, our algorithm differs from that in Chen et al. (2023b). First, we always use the most recent stochastic gradient $\boldsymbol{g}_i^t$ in the setting of $\boldsymbol{u}_i^t$ to update $\boldsymbol{x}_i$, while the decentralized AMSGrad in Chen et al. (2023b) uses $\boldsymbol{g}_i^t$ only to update its first moment vector but not the second momentum vector at the $t$-th step. Second, our algorithm only performs one round of communication for $\boldsymbol{x}_i$-variables, while the method in Chen et al. (2023b) needs additional communication for $\boldsymbol{u}_i$-iterates. Thus, our method can save half of the communication costs, even without using compression.

When $\mathcal{Q} \neq \mathbf{I}$, our algorithm can achieve higher communication efficiency. Note that by maintaining the weighted average $\boldsymbol{v}_i := \sum_{j=1}^n W_{ji}\underline{\boldsymbol{x}}_j$ for each $i$, we only need to communicate the compressed vectors $\left\{\mathcal{Q}[\boldsymbol{x}_j^{t+\frac{1}{2}} - \underline{\boldsymbol{x}}_j^t]\right\}$ to update $\boldsymbol{v}_i$ and thus obtain $\boldsymbol{x}_i^{t+1}$ by $\boldsymbol{v}_i^{t+1} = \boldsymbol{v}_i^t + \sum_{j=1}^n W_{ji}\mathcal{Q}[\boldsymbol{x}_j^{t+\frac{1}{2}} - \underline{\boldsymbol{x}}_j^t]$. To ensure reliability of the algorithm, $\mathcal{Q}$ must satisfy a certain quality condition. Throughout this paper, we assume the following condition.

**Assumption 3** *There exists* $\eta \in [0, 1)$ *such that* $\mathbb{E}_{\mathcal{Q}}\left[\|\mathbf{x} - \mathcal{Q}[\mathbf{x}]\|^2\right] \leq \eta^2 \|\mathbf{x}\|^2, \forall \mathbf{x} \in \mathbb{R}^d$.

The condition in Assumption 3 is satisfied by various compression operators in the literature, such as Random-$k$ (Stich et al., 2018), Top-$k$ (Aji & Heafield, 2017), and the rescaled quantizations (Chen et al., 2023a). In the appendix, we give a few concrete examples of $\mathcal{Q}$ that satisfy the above assumption. More examples can be found in Koloskova et al. (2019); Chen et al. (2023a).

### 3.2 Compressed Decentralized Heavy-ball Method

Similar to existing distributed adaptive SGMs, the boundedness condition on (sample) gradients is needed to ensure convergence of Alg. 1. Such a condition implies data similarity among the multiple agents, which is often measured by the quantity $\sup_{\boldsymbol{x}} \frac{1}{n} \sum_{i=1}^n \|\nabla f(\boldsymbol{x}) - \nabla f_i(\boldsymbol{x})\|^2$. To address the challenge of data heterogeneity and to achieve fast convergence and efficient communication, we propose to incorporate the techniques of gradient tracking (Lorenzo & Scutari, 2016; Nedic et al., 2017), momentum-based updates, and compressed communication, resulting in a compressed decentralized stochastic heavy-ball method in Alg. 2. This method

does not require bounded gradients and can successfully handle data heterogeneity. Note that we can also apply gradient tracking in Alg. 1 but a gradient boundedness assumption will still be required to ensure convergence.

In Alg. 2, we communicate not only model but also stochastic gradient. In addition, Line 5 uses the gradient tracking technique to handle data heterogeneity. Similar to the model compression, we use $\underline{\mathbf{g}}_i^t$ as an estimate of $\mathbf{g}_i^{t-\frac{1}{2}}$ and compress the estimate error. This technique helps control the impact of possibly unbounded gradients by allowing agents to track the global gradient. If the gradients are bounded, DaSHCo will not need to communicate gradients for convergence. Line 6 mixes local gradients and updates the first momentum. Line 7 performs the local update to model by using the first momentum term and then compresses the model estimate error. Line 8 performs a neighbor communication of local models.

---

**Algorithm 2: D**ecentralized **S**tochastic **H**eavy-ball Method with **Co**mpressed Communication (DaSHCo)

**1 Input:** choose $\alpha > 0$, $0 < \beta_1 < 1$, $\gamma_x, \gamma_g \in (0, 1]$ and a maximum number $T$ of updates, set $\boldsymbol{x}_i^0$, $\underline{\boldsymbol{x}}_i^0$,

$\boldsymbol{m}_i^{-1}$, $\boldsymbol{g}_i^{-1}$, $\widetilde{\boldsymbol{g}}_i^{-1}$, $\boldsymbol{g}_i^{-\frac{1}{2}}$ and $\underline{\boldsymbol{g}}_i^{-1}$ to $\mathbf{0}$, and choose a mixing matrix $\mathbf{W}$.

**2 for** $t = 0, 1, \cdots, T - 1$ **do**
**3**    **for** *all nodes $i \in [n]$ in parallel* **do**
**4**      obtain one random sample $\xi_i^t$ and compute a stochastic gradient $\widetilde{\boldsymbol{g}}_i^t \leftarrow \nabla \mathbf{F}_i(\mathbf{x}_i^t, \xi_i^t)$;
**5**      let $\boldsymbol{g}_i^{t-\frac{1}{2}} = \boldsymbol{g}_i^{t-1} - \widetilde{\boldsymbol{g}}_i^{t-1} + \widetilde{\boldsymbol{g}}_i^t$ and set $\underline{\boldsymbol{g}}_i^t = \underline{\boldsymbol{g}}_i^{t-1} + \mathcal{Q}[\boldsymbol{g}_i^{t-\frac{1}{2}} - \underline{\boldsymbol{g}}_i^{t-1}]$;
**6**      let $\boldsymbol{g}_i^t = \boldsymbol{g}_i^{t-\frac{1}{2}} + \gamma_g[\sum_{j=1}^n W_{ji}\underline{\boldsymbol{g}}_j^t - \underline{\boldsymbol{g}}_i^t]$ and $\boldsymbol{m}_i^t = \beta_1 \boldsymbol{m}_i^{t-1} + (1 - \beta_1)\boldsymbol{g}_i^t$;
**7**      update $\boldsymbol{x}_i^{t+\frac{1}{2}} = \boldsymbol{x}_i^t - \alpha \boldsymbol{m}_i^t$ and set $\underline{\boldsymbol{x}}_i^{t+1} = \underline{\boldsymbol{x}}_i^t + \mathcal{Q}[\boldsymbol{x}_i^{t+\frac{1}{2}} - \underline{\boldsymbol{x}}_i^t]$;
**8**      let $\boldsymbol{x}_i^{t+1} = \boldsymbol{x}_i^{t+\frac{1}{2}} + \gamma_x[\sum_{j=1}^n \mathbf{W}_{ji}\underline{\boldsymbol{x}}_j^{t+1} - \underline{\boldsymbol{x}}_i^{t+1}]$.

---

The heavy-ball (a.k.a. momentum) technique has been used in several distributed methods, e.g., Xu et al. (2022); Xiao et al. (2023). However, Alg. 2 is the first decentralized method that uses such a technique together with a compression technique and gradient tracking to simultaneously achieve fast convergence and efficient communication for handling data heterogeneity.

## 4 Convergence Analysis

In this section, we analyze the convergence of Alg. 1 and Alg. 2. We aim at establishing their convergence rate in terms of the violation of first-order optimality condition and consensus error. The analysis is challenging due to the nonconvexity of the problem, combined with the use of stochastic gradients, compressed decentralized communication, and momentum acceleration. Specifically, the coupling of compressed communication with momentum acceleration or adaptive gradient updates significantly complicates bounding both the consensus error and stationarity violation. To address this challenge, we carefully construct Lyapunov functions that explicitly separate compression error, consensus error, and stochastic noise. By delicately balancing these error terms through tailored choices of stepsizes and mixing parameters, we establish the first convergence guarantees for decentralized adaptive methods that simultaneously incorporate adaptive gradient updates and compressed communication (see Theorems A.1 and A.2). Main results are given below and detailed proofs in the appendix.

### 4.1 Convergence Results of Compressed Decentralized AMSGrad

Besides Assumptions 1–3, we make the next assumption that is standard for analyzing (distributed or non-distributed) adaptive SGMs (Reddi et al., 2018; Kingma & Ba, 2014; Xu et al., 2023; Chen et al., 2023b; 2019).

**Assumption 4** *The random samples $\{\xi_i^t\}_{i\in[n],t\geq 0}$ are independent. For each $t$ and $i$, it holds $\mathbb{E}_t[\boldsymbol{g}_i^t] = \nabla f_i(\boldsymbol{x}_i^t)$. In addition, there are constants $B$ and $B_\infty$ such that $\|\boldsymbol{g}_i^t\| \leq B$, $\|\boldsymbol{g}_i^t\|_\infty \leq B_\infty$ for any $i \in [n]$ and any $t$, and $\|\nabla f_i(\boldsymbol{x})\| \leq B$, $\|\nabla f_i(\boldsymbol{x})\|_\infty \leq B_\infty$ for all $\boldsymbol{x}$.*

The next two theorems give the convergence rate results. Their complete descriptions are given in Theorems A.1 and A.2 in the appendix, with proofs provided there.

**Theorem 4.1** *Under Assumptions 1–4, let $\{\boldsymbol{x}_i^t\}$ be generated by Alg. 1 with*

$$\alpha \leq \min\left\{\frac{\delta}{24L\sqrt{B_\infty^2+\delta}}, \frac{(1-\eta^2)^2}{32}\right\}, \quad \gamma \leq \min\left\{\frac{1-\widehat{\rho}^2}{60\eta}, \frac{1-\eta^2}{25}, \frac{\alpha}{\eta}\right\}, \tag{2}$$

*where $\widehat{\rho} = \|\gamma\mathbf{W} + (1-\gamma)\mathbf{I}\|_2 < 1$. Then with $C := \frac{12}{1-\widehat{\rho}^2}\left(\frac{4nB^2\widehat{\rho}^2}{\delta(1-\widehat{\rho}^2)} + \frac{nB^2}{45\delta}\right) + \frac{10nB^2}{\delta(1-\eta^2)^2}$, it holds that $\frac{1}{T}\sum_{t=0}^{T-1}\mathbb{E}\left[\|\mathbf{X}_\perp^t\|^2\right] \leq \alpha^2 C$ and*

$$\frac{\alpha}{4\sqrt{B_\infty^2+\delta}}\sum_{t=0}^{T-1}\mathbb{E}\left[\|\nabla f\left(\overline{\boldsymbol{x}}^t\right)\|^2\right]$$

$$\leq \left[f\left(\overline{\boldsymbol{x}}^0\right) - f^*\right] + \frac{\alpha\beta_1}{1-\beta_1}\frac{dB_\infty^2}{\sqrt{\delta}} + \frac{L\beta_1^2\alpha^2 dB_\infty^2}{\delta(1-\beta_1)^2} + \frac{24L\alpha^2}{n\delta}TB^2$$

$$+ \left(\frac{6L^3\alpha^2}{n\delta} + \frac{L^2\alpha}{n}\left(\frac{\sqrt{B_\infty^2+\delta}}{\delta} + \frac{1}{2\sqrt{\delta}}\right)\right)\alpha^2 CT + L\alpha^2 B_\infty^4\frac{2d}{\delta^2}\sum_{t=0}^{T-1}\left(1-\beta_2^{t+1}\right)$$

$$+ \alpha\sum_{t=0}^{T-1}\left(\frac{dB_\infty^4}{\delta^{\frac{3}{2}}}\left(1-\beta_2^{t+1}\right) + \frac{\alpha^2\beta_1^2 L^2 B^2}{\delta(1-\beta_1)^2}\left(\frac{\sqrt{B_\infty^2+\delta}}{\delta} + \frac{1}{2\sqrt{\delta}}\right)\right).$$

By Theorem 4.1, we specify the choice of $\alpha$ and $\delta$ and obtain the convergence rate as follows.

**Theorem 4.2** *Let $\alpha = \frac{4\theta\sqrt{n(B_\infty^2+\delta)}}{\sqrt{T}}$ satisfy (2) for some constant $\theta \in (\frac{n}{Le}, \frac{n}{L})$ and a large enough $T$. Suppose that all other conditions assumed in Theorem 4.1 also hold and suppose $n \leq T$. Then we have the following results.*

(i) *Let $\delta = \omega^2 B_\infty^2\frac{\sqrt{T}}{\sqrt{n}}$ for a universal constant $\omega > 0$ and $C$ defined in Theorem 4.1. Set $\beta_2 \in \left[\frac{T}{T+1}, (\frac{\theta L}{n})^{\frac{1}{T}}\right]$. Then*

$$\frac{1}{T}\sum_{t=0}^{T-1}\mathbb{E}\left[\|\nabla f\left(\overline{\boldsymbol{x}}^t\right)\|^2 + \frac{1}{n}\|\mathbf{X}_\perp^t\|^2\right] = O\left(\frac{1}{\sqrt{nT}} + \frac{n+C}{T}\right). \tag{3}$$

(ii) *Let $\delta = O(1)$ be an universal positive constant. Set $\beta_2 \in [\frac{pT}{pT+1}, 1)$ with $p = \sqrt{nT}$. Then (3) also holds.*

**Remark 1 (Linear speed up and topology independence)** *In Case (i) of Theorem 4.2, we have $C = O(1)$ and $\frac{n}{T} = O(\frac{1}{\sqrt{nT}})$ if $T$ is large enough such that $T = \Omega\left(\max\left\{\frac{n^3}{(1-\widehat{\rho}^2)^4}, \frac{n^{\frac{7}{3}}}{(1-\eta^2)^{\frac{8}{3}}}\right\}\right)$. Thus (2) holds, and*

$$\frac{1}{T}\sum_{t=0}^{T-1}\mathbb{E}\left[\|\nabla f\left(\overline{\boldsymbol{x}}^t\right)\|^2 + \frac{1}{n}\|\mathbf{X}_\perp^t\|^2\right] = O\left(\frac{1}{\sqrt{nT}}\right). \tag{4}$$

*For Case (ii) of Theorem 4.2, if $T$ is large enough such that $T = \Omega\left(\max\left\{\frac{n^3}{(1-\widehat{\rho}^2)^4}, \frac{n^2}{(1-\eta^2)^2}\right\}\right)$, it then holds $\frac{1+C}{T} = O(\frac{1}{\sqrt{nT}})$. Thus again we have (4). Therefore, if $T$ is large enough, we obtain linear speed up, and the learning rate is independent of $\rho$ defined in Assumption 2, i.e., topology-independent.*

## 4.2 Convergence Results of Compressed Decentralized Heavy-ball Method

Instead of Assumption 4, we assume a weaker condition, which is standard for analyzing SGMs.

**Assumption 5** *The random samples $\{\xi_i^t\}_{i\in[n],t\geq 0}$ are independent. In addition, there is $\sigma \geq 0$ such that $\mathbb{E}_t[\boldsymbol{g}_i^t] = \nabla f_i(\boldsymbol{x}_i^t)$ and $\mathbb{E}\left[\|\boldsymbol{g}_i^t - \nabla f_i(\boldsymbol{x}_i^t)\|^2\right] \leq \sigma^2$ for all $t$ and $i$.*

A complete description of the next theorem about the convergence rate is in Theorems B.3 and B.4.

**Theorem 4.3** *Suppose Assumptions 1–3 and 5 hold. Let $\alpha = \frac{\theta_1\sqrt{n}}{\sigma\sqrt{T}}$, $\gamma_g = \frac{\theta_2\sqrt{n}}{\sigma\sqrt{T}}$ and $\gamma_x = \frac{\theta_3\sqrt{n}}{\sigma\sqrt{T}}$ for some constants $\theta_1,\theta_2,\theta_3 \in (0,1)$ such that*

$$\gamma_x \leq \min\left\{\frac{1-\widehat{\rho}_x^2}{60\eta}, \frac{1-\eta^2}{25}, \frac{\alpha}{\eta}, \frac{\sqrt{2}-1}{2\eta}\right\},$$

$$\gamma_g \leq \min\left\{\frac{1-\widehat{\rho}_g^2}{25\eta}, \frac{1-\widehat{\rho}_g^2}{25L}, \frac{1-\eta^2}{25}, \frac{1-\eta^2}{25L}\right\},$$

$$\alpha \leq \min\left\{\frac{1}{16b}, \frac{\gamma_g(1-\eta^2)}{32}, \frac{\gamma_g(1-\eta^2)}{12L\sqrt{n}}, \frac{\gamma_g(1-\widehat{\rho}_x^2)}{12\sqrt{b}}, \frac{\gamma_g(1-\widehat{\rho}_g^2)}{12L\sqrt{n}}, \sqrt{\frac{(1-\beta_1)\gamma_g^2/(2L)}{\frac{6L\beta_1^2}{(1-\beta_1)^2} + \frac{b}{45(1-\widehat{\rho}_x^2)} + \frac{24L^2+1}{(1-\eta^2)^2} + \frac{20L^2}{(1-\widehat{\rho}_g^2)^2}}}\right\},$$

*with $b = 4\left(9L + 1 + \frac{72L^2+1}{(1-\eta^2)^2} + \frac{60L^2}{(1-\widehat{\rho}_g^2)^2}\right)$ and $\widehat{\rho}_s = \|\gamma_s\mathbf{W} + (1-\gamma_s)\mathbf{I}\|_2 < 1$ for $s \in \{x,g\}$. Then*

$$\frac{1}{T}\sum_{t=0}^{T-1}\mathbb{E}\left[\|\nabla f\left(\overline{\boldsymbol{x}}^t\right)\|^2 + \frac{1}{n}\|\mathbf{X}_{\perp}^t\|^2\right] = O\left(\frac{\sigma}{\sqrt{nT}} + \frac{n}{T}\left(\frac{1}{1-\widehat{\rho}_g^2} + \frac{1}{(1-\eta^2)^2}\right)\right).$$

**Remark 2 (Linear speed up and topology independence)** *From Theorem 4.3, if $T$ is large enough such that*

$$T = \Omega\Big(\max\Big\{\frac{n}{\sigma^2(1-\widehat{\rho}_g^2)^6}, \frac{n}{\sigma^2(1-\widehat{\rho}_x^2)^6}, \frac{n}{\sigma^2(1-\eta^2)^6},$$

$$\frac{n^2}{\sigma^2(1-\widehat{\rho}_g^2)^4}, \frac{n^2}{\sigma^2(1-\widehat{\rho}_x^2)^4}, \frac{n^3}{\sigma^2(1-\widehat{\rho}_g^2)^2}, \frac{n^3}{\sigma^2(1-\eta^2)^4}\Big\}\Big),$$

*we obtain that $\alpha, \gamma_g$ and $\gamma_x$ are all $O(\frac{\sqrt{n}}{\sigma\sqrt{T}})$, and $\frac{n}{T}\left(\frac{1}{(1-\widehat{\rho}_g^2)^2} + \frac{1}{(1-\eta^2)^2}\right) = O(\frac{\sigma}{\sqrt{nT}})$. Thus it holds*

$$\frac{1}{T}\sum_{t=0}^{T-1}\mathbb{E}\left[\|\nabla f\left(\overline{\boldsymbol{x}}^t\right)\|^2 + \frac{1}{n}\|\mathbf{X}_{\perp}^t\|^2\right] = O\left(\frac{\sigma}{\sqrt{nT}}\right),$$

*and we obtain linear speed up with $\alpha$ independent of $\rho$ defined in Assumption 2.*

## 5 Numerical Results

We now demonstrate the efficacy of the proposed algorithms over a set of numerical experiments. We consider three standard benchmarks, including training a convolutional neural network LeNet5 (LeCun et al., 1998) on the FashionMNIST dataset (Xiao et al., 2017), a restnet architecture Fixup-ResNet-20 (Zhang et al., 2019) on the CIFAR-10 dataset (Krizhevsky et al., 2009), and a small-scale GPT model, called NanoGPT (Andrej, 2022), on the tiny-shakespeare dataset. We will show the performance of our proposed methods (DAMSCo and DaSHCo) on homogeneous data and the ability of DaSHCo on handling heterogeneous data. Our test data and neural networks are selected to demonstrate practical performance. Our code is available for download at the following repository: `https://github.com/DecentralizedMethods/DAMSCo_DaSHCo`.

We primarily compare against three prior works: CDProxSGT (Yan et al., 2023), Distributed Adam (Nazari et al., 2022), and Distributed AdaGrad (Duchi et al., 2011) with adaptive learning rates (Chen et al., 2023b).

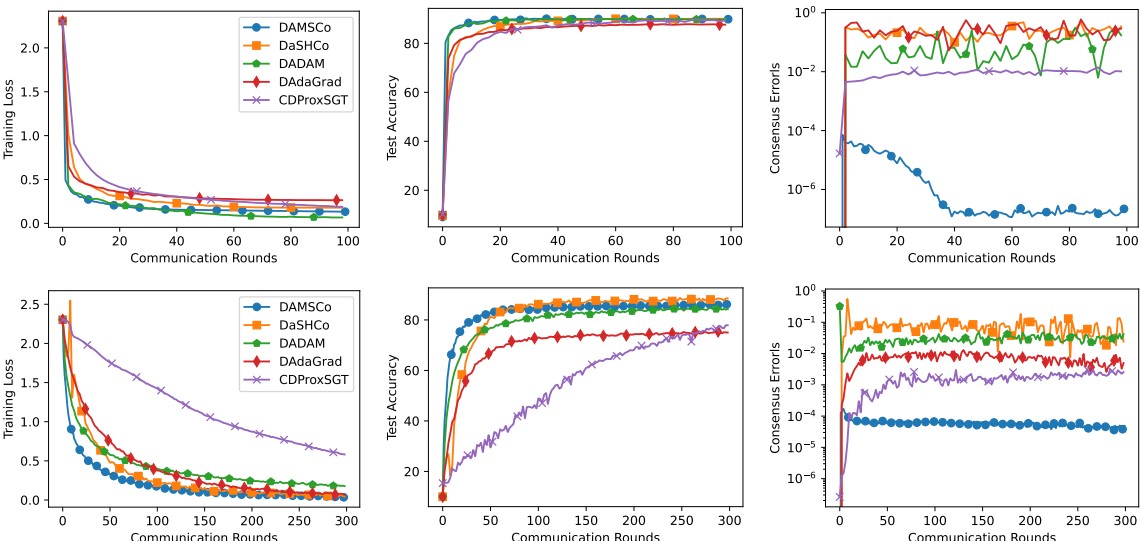

Figure 1: **Results with homogeneous data:** Plotted above are (from left to right) the training loss, test accuracy, and consensus error per communication round for the FashionMNIST (top) and CIFAR-10 (bottom) datasets, comparing DAMSCo and DaSHCo with CDProxSGT, Distributed AdaGrad, and Distributed Adam with Top-$k$ compression.

Recently, CDProxSGT was proposed as a proximal SGM for training on heterogeneous data in a compressed and decentralized setting. It demonstrated superior numeric performance to other competing methods (Xin et al., 2021; Koloskova et al., 2019; Zhao et al., 2022a). Chen et al. (Chen et al., 2023b) presented a general framework for decentralized adaptive gradient methods. They demonstrated that communicating the second-momentum vector information is critical to guaranteeing convergence of decentralized adaptive methods. Therefore, we incorporate their method into our implementation of Distributed Adam and AdaGrad. We consider these methods to be a practical representation of the state-of-the-art.

Our methods and the methods for comparison are implemented in Python with *PyTorch* and *MPI for Python* (mpi4py) and they will be open-sourced upon publication. For LeNet5 and Fixup-ResNet-20, we run our experiments on a CPU server. This server has two-way 64-core (256 threads) AMD EPYC 7742 CPUs at 2.25GHz and 2TB DDR4 memory. It runs Ubuntu 20.04 with PyTorch version 2.3.0+cu121, Python 3.8.10, and mpi4py version 3.0.3. For NanoGPT, we run the experiments on 4 NVIDIA A100 GPUs.

We run the experiments for LeNet5 and Fixup-ResNet-20 on $n = 5$ MPI ranks and for NanoGPT on $n = 4$ MPI ranks with the communication network in a ring topology. Each rank only communicates with their two neighbors. We measure and report objective loss on the full training data, accuracy on the full test data, and consensus error calculated as $\frac{1}{n}\|\mathbf{X}_\perp\|^2$. We plot these outputs against the number of communication rounds, under the assumption that these methods are targeting systems where communication is extremely costly. In the appendix, we show the same experimental results but as a function of epoch. In addition, we show results for larger ranks, a grid network topology, and QSGD compression (Alistarh et al., 2017) to demonstrate that our methods can generalize beyond the earlier computational results.

## 5.1 Results for Training LeNet5 and Fixup-ResNet-20 on Homogeneous Data

Here, we present results on the described optimizers with random homogeneous training data. For compression, we use the Top-$k$ compressor (Aji & Heafield, 2017) to compress gradient data and model parameter for CDProxSGT, DaSHCo, Distributed AdaGrad (DAdaGrad), and Distributed Adam (DADAM), while we only communicate parameter updates with DAMSCo. We set $\gamma_x = \gamma_g = \gamma = 1.0$ for all compressors to mirror prior experiments (Yan et al., 2023). For FashionMNIST, we use top-$k(0.3)$, communicating the largest 30% of values. For CIFAR-10, we use top-$k(0.4)$.

We train LeNet5 on the FashionMNIST dataset for 100 communication rounds for all optimizers. For CDProxSGT, we use the code published by the authors along with the same learning rate (0.02), batch size (8), and $\mu$ value for the regularizer ($10^{-4}$), as tuned by the authors. We mirror these hyperparameter settings for DaSHCo. For DADAM and DAMSCo, we mirror the batch size of 8, set the learning rate to the PyTorch default for Adam of 0.001, and similarly use the standard $\beta$ defaults of $\beta_1 = 0.9, \beta_2 = 0.999$. For DAdaGrad, we use the same batch size and the PyTorch default learning rate of 0.01.

We train Fixup-ResNet-20 on the CIFAR-10 dataset for 300 communication rounds. We again use the authors' tuned values for CDProxSGT, with the batch size increased to 64. We mirror the batch size of 64 and keep the other hyperparameter settings the same as described above for the other optimizers.

Results for homogeneous data are shown in Figure 9. We observe that our DAMSCo method is very competitive, converging at about the same rate on the FashionMNIST dataset as DADAM and much quicker on the more challenging CIFAR-10 dataset, with respect to the objective function and test accuracy. We note that both DAMSCo and DaSHCo demonstrate better performance than CDProxSGT on both datasets, which itself had recently given strong numerical performance against several other state-of-the-art methods (note that CDProxSGT was primarily developed for heterogeneous data).

## 5.2 Results for Training LeNet5 and Fixup-ResNet-20 on Heterogeneous Data

Next, we compare all methods on heterogeneous training data, with an equal number (i.e., 2) of label classes from each dataset distributed to each of the 5 MPI ranks. We lower the batch size to 32 to help improve generalization performance on CIFAR-10, but we otherwise reuse the same learning rates and other hyperparameters from the earlier experiments. We again run for 100 communication rounds on FashionMNIST and use 300 rounds on CIFAR-10. We demonstrate our results for these experiments in Figure 10.

We note several obvious differences with the results on homogeneous data. CDProxSGT is considerably more competitive, as it was primarily developed for the heterogeneous case. Our DaSHCo method converges most quickly with respect to testing accuracy on both FashionFMNIST and CIFAR-10. DAdaGrad, DADAM, and DAMSCo are unable to quickly converge in these experiments. This is expected with these types of methods, as data heterogeneity can break assumptions for convergence guarantees, such as bounded gradients and gradient similarities.

## 5.3 Results for Training NanoGPT

Lastly, we compare all methods on homogeneous training data using a customizable GPT model implemented by Andrej (2022). It is a standard Transformer architecture but we experiment with fewer parameters: 6 layers, 6 heads in each layer, 384 feature channels, and 256 context length. This totals to 10.67M parameters. We run training on 4 NVIDIA A100 GPUs and thus 4 MPI Ranks. We tune two hyperparameters: learning rate ($\alpha$) and batch-size. For all optimizers, we set the batch-size to 256 and $\alpha = 10^{-4}$ except for DaSHCo and CDProxSGT, in which we set $\alpha = 0.02$. We train the GPT model on the tiny-shakespeare dataset with top-$k(0.55)$ and use 650 communication rounds. After training, we achieve minimum validation losses of 2.321, 1.649, 1.641, 1.635, 1.620 for CDProxSGT, DAMSCo, DAdam, DAdaGrad, and DaSHCo respectively. We demonstrate our results for these experiments in Figure 11. We give a qualitative comparison of inference performance in the appendix.

## 6 Conclusion

We have presented two compressed decentralized stochastic gradient methods (SGMs), DAMSCo and DaSHCo, with a technique of either adaptive updates or heavy-ball (a.k.a. momentum) acceleration, for solving multi-agent nonconvex stochastic optimization. Our methods can simultaneously achieve fast convergence (by adaptive or heavy-ball updates) and efficient communication (by message compression). Both methods achieve the optimal convergence rate. Similar to existing distributed adaptive SGMs, the convergence of DAMSCo relies on the condition of bounded gradients, while DaSHCo does not require any bound-

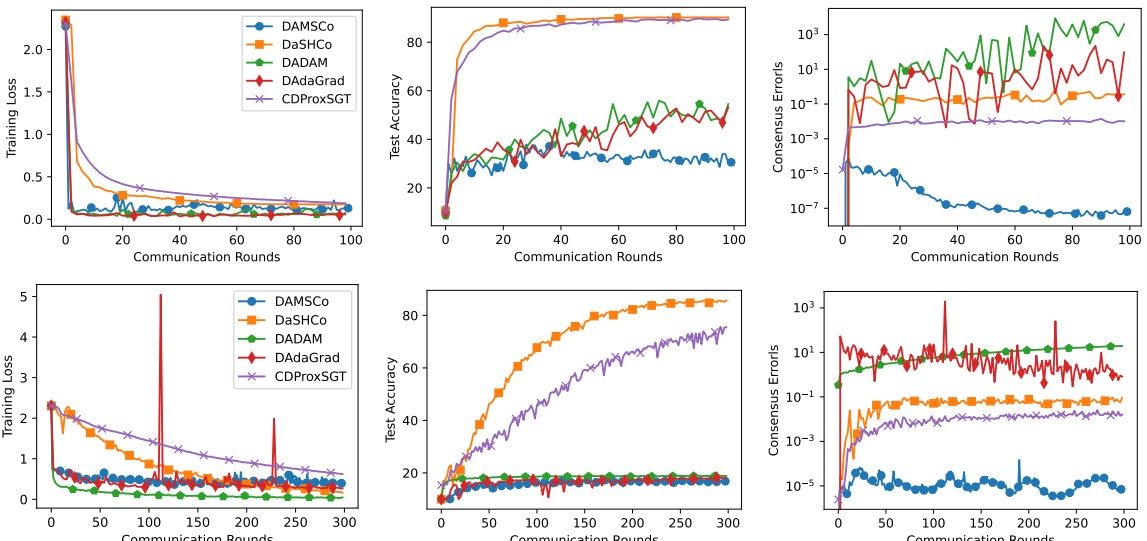

Figure 2: **Results with heterogeneous data:** Plotted above are (from left to right) the training loss, test accuracy, and consensus error per communication round for the FashionMNIST (top) and CIFAR-10 (bottom) datasets, comparing DAMSCo and DaSHCo with CDProxSGT, Distributed AdaGrad, and Distributed Adam with Top-$k$ compression.

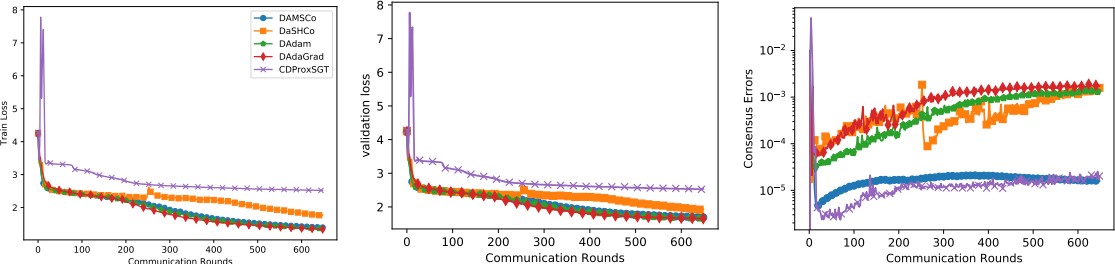

Figure 3: **GPT Results with homogeneous data:** Plotted above are (from left to right) the training loss, validation loss, and consensus error per communication round for the Shakespeare dataset, comparing DAMSCo and DaSHCo with CDProxSGT, Distributed AdaGrad, and Distributed Adam with Top-$k$ compression.

edness condition and can successfully address the challenge of data heterogeneity. Numerical experiments on training deep neural networks demonstrate the superiority of the proposed methods over state-of-the-art methods.

## Acknowledgements

The authors would like to thank three anonymous reviewers for their valuable comments. This work is partly supported by NSF grant DMS-2208394, ONR grant N000142212573, and also by IBM through the IBM-Rensselaer Future of Computing Research Collaboration.

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

## A   Convergence Rate Results of Algorithm 1

In this section, we analyze the convergence rate of Algorithm 1. By the notation introduced in Section 1.3, we can write the updates of Algorithm 1 in the more compact matrix form:

$$\mathbf{M}^t = \beta_1 \mathbf{M}^{t-1} + (1 - \beta_1)\mathbf{G}^t, \tag{5}$$

$$\widehat{\mathbf{U}}^t = \beta_2 \widehat{\mathbf{U}}^{t-1} + (1 - \beta_2)\mathbf{G}^t \circ \mathbf{G}^t, \tag{6}$$

$$\mathbf{U}^t = \max\left\{\widehat{\mathbf{U}}^t, \mathbf{U}^{t-1}\right\}, \tag{7}$$

$$\mathbf{Y}^t := \frac{\mathbf{M}^t}{\sqrt{\mathbf{U}^t + \delta}}, \tag{8}$$

$$\mathbf{X}^{t+\frac{1}{2}} = \mathbf{X}^t - \alpha \mathbf{Y}^t, \tag{9}$$

$$\underline{\mathbf{X}}^{t+1} = \underline{\mathbf{X}}^t + \mathcal{Q}\left[\mathbf{X}^{t+\frac{1}{2}} - \underline{\mathbf{X}}^t\right], \tag{10}$$

$$\mathbf{X}^{t+1} = \mathbf{X}^{t+\frac{1}{2}} + \gamma\underline{\mathbf{X}}^{t+1}(\mathbf{W} - \mathbf{I}), \tag{11}$$

where

$$\mathbf{G}^t = \left[\boldsymbol{g}_1^t, \boldsymbol{g}_2^t, \dots, \boldsymbol{g}_n^t\right], \ \mathbf{M}^t = \left[\boldsymbol{m}_1^t, \boldsymbol{m}_2^t, \dots, \boldsymbol{m}_n^t\right], \ \mathbf{U}^t = \left[\boldsymbol{u}_1^t, \boldsymbol{u}_2^t, \dots, \boldsymbol{u}_n^t\right].$$

Also we use similar notations as in Section 1.3 by letting

$$\overline{\mathbf{m}}^t = \frac{1}{n}\mathbf{M}^t \mathbf{1}, \ \overline{\mathbf{y}}^t = \frac{1}{n}\mathbf{Y}^t \mathbf{1}, \ \overline{\mathbf{Y}}^t := \mathbf{YJ}.$$

Let $\widehat{\mathbf{W}} = \gamma\mathbf{W} + (1 - \gamma)\mathbf{I}$. Then we can write (11) to

$$\mathbf{X}^{t+1} = \mathbf{X}^{t+\frac{1}{2}}\widehat{\mathbf{W}} + \gamma\left(\underline{\mathbf{X}}^{t+1} - \mathbf{X}^{t+\frac{1}{2}}\right)(\mathbf{W} - \mathbf{I}). \tag{12}$$

When $\mathbf{W}$ satisfies the conditions in Assumption 2, it can be shown that $\widehat{\mathbf{W}}$ also satisfies all three conditions. Indeed, we have

$$\widehat{\rho} := \left\|\widehat{\mathbf{W}} - \mathbf{J}\right\|_2 < 1.$$

Below we bound the sequence $\{\mathbf{M}^t\}$, $\{\mathbf{U}^t\}$ and $\{\mathbf{Y}^t\}$.

**Lemma A.1** *Under Assumption 4, it holds that for any $t \geq 0$,*

$$\|\mathbf{M}^t\| \leq \left(1 - \beta_1^{t+1}\right)\sqrt{n}B \leq \sqrt{n}B, \quad \|\mathbf{Y}^t\| \leq \sqrt{n}B\delta^{-\frac{1}{2}}, \tag{13}$$

$$\|\boldsymbol{m}_i^t\| \leq B, \quad \|\boldsymbol{m}_i^t\|_\infty \leq B_\infty, \quad \|\boldsymbol{u}_i^t\|_\infty \leq \left(1 - \beta_2^{t+1}\right)B_\infty^2 \leq B_\infty^2, \quad \forall i \in [n]. \tag{14}$$

*Proof.* From the update of $\boldsymbol{m}$, i.e., $\boldsymbol{m}_i^t = \beta_1 \boldsymbol{m}_i^{t-1} + (1-\beta_1)\boldsymbol{g}_i^t$, we have that for any $t \geq 0$ and each $i \in [n]$,

$$\|\boldsymbol{m}_i^t\| = \|\beta_1 \boldsymbol{m}_i^{t-1} + (1-\beta_1)\mathbf{g}_i^t\| \leq \beta_1 \|\boldsymbol{m}_i^{t-1}\| + (1-\beta_1)\|\mathbf{g}_i^t\| \leq \beta_1\|\boldsymbol{m}_i^{t-1}\|_\infty + (1-\beta_1)B,$$

where the second inequality holds from $\|\boldsymbol{g}_i^t\| \leq B$ by Assumption 4. Recursively applying the inequality above and noticing $\boldsymbol{m}_i^{-1} = \mathbf{0}$, we obtain

$$\|\boldsymbol{m}_i^t\| \leq \left(1 + \beta_1 + \beta_1^2 + \ldots + \beta_1^t\right)(1-\beta_1)B = \left(1 - \beta_1^{t+1}\right)B \leq B.$$

Hence, $\|\mathbf{M}^t\| \leq \left(1 - \beta_1^{t+1}\right)\sqrt{n}B$. Now by $\mathbf{U}^t \geq \mathbf{0}$, we immediately have $\|\mathbf{Y}^t\| = \left\|\frac{\mathbf{M}^t}{\sqrt{\mathbf{U}^t + \delta}}\right\| \leq \sqrt{n}B\delta^{-\frac{1}{2}}$.

In addition, we have that for any $t \geq 0$ and each $i \in [n]$,

$$\|\boldsymbol{m}_i^t\|_\infty = \|\beta_1 \boldsymbol{m}_i^{t-1} + (1-\beta_1)\mathbf{g}_i^t\|_\infty \leq \beta_1\|\boldsymbol{m}_i^{t-1}\|_\infty + (1-\beta_1)\|\mathbf{g}_i^t\|_\infty \leq \beta_1\|\boldsymbol{m}_i^{t-1}\|_\infty + (1-\beta_1)B_\infty,$$

where the second inequality follows from $\|\boldsymbol{g}_i^t\|_\infty \leq B_\infty$ by Assumption 4. Recursively applying the inequality above and noticing $\boldsymbol{m}_i^{-1} = \mathbf{0}$, we obtain

$$\|\boldsymbol{m}_i^t\|_\infty \leq \left(1 + \beta_1 + \beta_1^2 + \ldots + \beta_1^t\right)(1-\beta_1)B_\infty = \left(1 - \beta_1^{t+1}\right)B_\infty \leq B_\infty.$$

Similarly, by noticing $\widehat{\boldsymbol{u}}_i^{-1} = \mathbf{0}$ and $\|\boldsymbol{g}_i^t \circ \boldsymbol{g}_i^t\|_\infty \leq B_\infty^2$, we have $\|\widehat{\boldsymbol{u}}_i^t\|_\infty \leq \left(1 - \beta_2^{t+1}\right)B_\infty^2$ for each $i \in [n]$ and $t \geq 0$. Now by $\boldsymbol{u}_i^t = \max\{\widehat{\boldsymbol{u}}_i^t, \boldsymbol{u}_i^{t-1}\}$, it holds

$$\begin{aligned}
\|\boldsymbol{u}_i^t\|_\infty &\leq \max\{\|\widehat{\boldsymbol{u}}_i^t\|_\infty, \|\boldsymbol{u}_i^{t-1}\|_\infty\} \leq \max\{\|\widehat{\boldsymbol{u}}_i^t\|_\infty, \|\widehat{\boldsymbol{u}}_i^{t-1}\|_\infty, \|\boldsymbol{u}_i^{t-2}\|_\infty\} \\
&\leq \max\left\{\|\widehat{\boldsymbol{u}}_i^t\|_\infty, \|\widehat{\boldsymbol{u}}_i^{t-1}\|_\infty, \ldots, \|\widehat{\boldsymbol{u}}_i^0\|_\infty, \|\boldsymbol{u}_i^{-1}\|_\infty\right\}, \\
&\leq \max\left\{\left(1 - \beta_2^{t+1}\right)B_\infty^2, \left(1 - \beta_2^t\right)B_\infty^2, \ldots, (1-\beta_2)B_\infty^2, \|\boldsymbol{u}_i^{-1}\|_\infty\right\} = \left(1 - \beta_2^{t+1}\right)B_\infty^2,
\end{aligned}$$

where the equality holds because $\beta_2 \in (0,1)$ and $\boldsymbol{u}_i^{-1} = \mathbf{0}$. The proof is then completed. $\qquad\square$

The next lemma shows the bound of the consensus error of $\mathbf{X}$.

**Lemma A.2** *Under Assumptions 2 and 3, let $\alpha \leq \frac{(1-\eta^2)^2}{32}$ and $\gamma \leq \min\{\frac{1-\widehat{\rho}^2}{60\eta}, \frac{1-\eta^2}{25}, \frac{\alpha}{\eta}\}$. Then*

$$\frac{1}{T}\sum_{t=0}^{T-1}\mathbb{E}\left[\|\mathbf{X}_\perp^t\|^2\right] \leq \alpha^2 C, \text{ where } C := \frac{12}{1-\widehat{\rho}^2}\left(\frac{4nB^2\widehat{\rho}^2}{\delta(1-\widehat{\rho}^2)} + \frac{nB^2}{45\delta}\right) + \frac{320\alpha nB^2}{\delta(1-\eta^2)^2}. \tag{15}$$

*Proof.* By (12) and $(\mathbf{W} - \mathbf{I})(\mathbf{I} - \mathbf{J}) = \mathbf{W} - \mathbf{I}$, we have $\mathbf{X}_\perp^{t+1} = \mathbf{X}^{t+\frac{1}{2}}\widehat{\mathbf{W}}(\mathbf{I} - \mathbf{J}) + \gamma(\underline{\mathbf{X}}^{t+1} - \mathbf{X}^{t+\frac{1}{2}})(\mathbf{W} - \mathbf{I})$ and thus for any $\eta_1 > 0$, it holds

$$\begin{aligned}
\left\|\mathbf{X}_\perp^{t+1}\right\|^2 &\leq (1+\eta_1)\left\|\mathbf{X}^{t+\frac{1}{2}}\widehat{\mathbf{W}}(\mathbf{I} - \mathbf{J})\right\|^2 + \left(1 + \eta_1^{-1}\right)\left\|\gamma\left(\underline{\mathbf{X}}^{t+1} - \mathbf{X}^{t+\frac{1}{2}}\right)(\mathbf{W} - \mathbf{I})\right\|^2 \\
&\leq (1+\eta_1)\left\|\mathbf{X}^{t+\frac{1}{2}}\widehat{\mathbf{W}}(\mathbf{I} - \mathbf{J})\right\|^2 + 4\left(1 + \eta_1^{-1}\right)\gamma^2\left\|\left(\underline{\mathbf{X}}^{t+1} - \mathbf{X}^{t+\frac{1}{2}}\right)\right\|^2, \tag{16}
\end{aligned}$$

where the second inequality follows from $\|\mathbf{W} - \mathbf{I}\|_2 \leq 2$. In addition, by $\widehat{\mathbf{W}}\mathbf{J} = \mathbf{J}$, we have

$$\begin{aligned}
\left\|\mathbf{X}^{t+\frac{1}{2}}\widehat{\mathbf{W}}(\mathbf{I} - \mathbf{J})\right\|^2 &= \left\|\mathbf{X}^{t+\frac{1}{2}}(\widehat{\mathbf{W}} - \mathbf{J})\right\|^2 = \left\|(\mathbf{X}^t - \alpha\mathbf{Y}^t)(\widehat{\mathbf{W}} - \mathbf{J})\right\|^2 \\
&= \left\|\left((\mathbf{X}^t - \alpha\mathbf{Y}^t) - \left(\overline{\mathbf{X}}^t - \alpha\overline{\mathbf{Y}}^t\right)\right)(\widehat{\mathbf{W}} - \mathbf{J})\right\|^2 \leq \widehat{\rho}^2\left\|\mathbf{X}_\perp^t - \alpha\mathbf{Y}_\perp^t\right\|^2 \\
&\leq \left(\widehat{\rho}^2 + \frac{1-\widehat{\rho}^2}{2}\right)\left\|\mathbf{X}_\perp^t\right\|^2 + \left(\widehat{\rho}^2 + \frac{2\widehat{\rho}^4}{1-\widehat{\rho}^2}\right)\alpha^2\left\|\mathbf{Y}_\perp^t\right\|^2 \\
&= \frac{1+\widehat{\rho}^2}{2}\left\|\mathbf{X}_\perp^t\right\|^2 + \frac{1+\widehat{\rho}^2}{1-\widehat{\rho}^2}\widehat{\rho}^2\alpha^2\left\|\mathbf{Y}_\perp^t\right\|^2 \leq \frac{1+\widehat{\rho}^2}{2}\left\|\mathbf{X}_\perp^t\right\|^2 + \frac{2\widehat{\rho}^2\alpha^2}{1-\widehat{\rho}^2}\left\|\mathbf{Y}_\perp^t\right\|^2. \tag{17}
\end{aligned}$$

Notice
$$\|\mathbf{Y}_\perp^t\|^2 = \|\mathbf{Y}^t - \overline{\mathbf{Y}}^t\|^2 = \|\mathbf{Y}^t\|^2 - \|\overline{\mathbf{Y}}^t\|^2 \overset{(13)}{\leq} \frac{nB^2}{\delta}, \tag{18}$$
which together with (17) gives
$$\left\| \mathbf{X}^{t+\frac{1}{2}} \widehat{\mathbf{W}}(\mathbf{I} - \mathbf{J}) \right\|^2 \leq \frac{1 + \widehat{\rho}^2}{2} \left\| \mathbf{X}_\perp^t \right\|^2 + \frac{2nB^2\widehat{\rho}^2\alpha^2}{\delta(1 - \widehat{\rho}^2)}. \tag{19}$$

Moreover, by (10) and Assumption 3, it holds
$$\mathbb{E}\left[ \left\| \underline{\mathbf{X}}^{t+1} - \mathbf{X}^{t+\frac{1}{2}} \right\|^2 \right] = \mathbb{E}\left[ \mathbb{E}_Q \left[ \left\| Q\left[ \mathbf{X}^{t+\frac{1}{2}} - \underline{\mathbf{X}}^t \right] - \left( \mathbf{X}^{t+\frac{1}{2}} - \underline{\mathbf{X}}^t \right) \right\|^2 \right] \right]$$
$$\leq \eta^2 \mathbb{E}\left[ \left\| \mathbf{X}^{t+\frac{1}{2}} - \underline{\mathbf{X}}^t \right\|^2 \right] = \eta^2 \mathbb{E}\left[ \left\| \mathbf{X}^{t+\frac{1}{2}} - \mathbf{X}^t + \mathbf{X}^t - \underline{\mathbf{X}}^t \right\|^2 \right]$$
$$\leq \eta^2 \left( 1 + \eta_2 \right) \mathbb{E}\left[ \left\| \mathbf{X}^t - \underline{\mathbf{X}}^t \right\|^2 \right] + \eta^2 \left( 1 + \eta_2^{-1} \right) \mathbb{E}\left[ \left\| \mathbf{X}^{t+\frac{1}{2}} - \mathbf{X}^t \right\|^2 \right]$$
$$= \eta^2 \left( 1 + \eta_2 \right) \mathbb{E}\left[ \left\| \mathbf{X}^t - \underline{\mathbf{X}}^t \right\|^2 \right] + \eta^2 \left( 1 + \eta_2^{-1} \right) \alpha^2 \mathbb{E}\left[ \|\mathbf{Y}^t\|^2 \right] \tag{20}$$
$$\leq \eta^2 \left( 1 + \eta_2 \right) \mathbb{E}\left[ \left\| \mathbf{X}^t - \underline{\mathbf{X}}^t \right\|^2 \right] + \eta^2 \left( 1 + \eta_2^{-1} \right) \alpha^2 nB^2\delta^{-1}, \tag{21}$$
where $\eta_2 > 0$ is any positive number. Substituting (19) and (21) with $\eta_2 = 1$ into (16) after taking full expectation, we obtain
$$\mathbb{E}\left[ \|\mathbf{X}_\perp^{t+1}\|^2 \right] \leq (1 + \eta_1) \left( \frac{1 + \widehat{\rho}^2}{2} \mathbb{E}\left[ \|\mathbf{X}_\perp^t\|^2 \right] + \frac{2nB^2\widehat{\rho}^2\alpha^2}{\delta(1 - \widehat{\rho}^2)} \right)$$
$$+ 8\eta^2 \left( 1 + \eta_1^{-1} \right) \gamma^2 \left( \mathbb{E}\left[ \left\| \mathbf{X}^t - \underline{\mathbf{X}}^t \right\|^2 \right] + \alpha^2 nB^2\delta^{-1} \right). \tag{22}$$

Let $\eta_1 = \frac{7\eta\gamma}{1 - \widehat{\rho}^2}$ in (22). Then by $\gamma \leq \frac{1 - \widehat{\rho}^2}{60\eta}$, we obtain
$$1 + \eta_1 < 2, \quad (1 + \eta_1)\frac{1 + \widehat{\rho}^2}{2} \leq \frac{3 + \widehat{\rho}^2}{4}, \quad \eta^2 \left( 1 + \eta_1^{-1} \right) \gamma^2 \leq \frac{\eta\gamma(1 - \widehat{\rho}^2)}{6} < \frac{1}{360}. \tag{23}$$
Thus we have from (22) that
$$\mathbb{E}\left[ \|\mathbf{X}_\perp^{t+1}\|^2 \right] \leq \frac{3 + \widehat{\rho}^2}{4} \mathbb{E}\left[ \|\mathbf{X}_\perp^t\|^2 \right] + \frac{4nB^2\widehat{\rho}^2\alpha^2}{\delta(1 - \widehat{\rho}^2)} + \frac{4\eta\gamma(1 - \widehat{\rho}^2)}{3} \mathbb{E}\left[ \left\| \mathbf{X}^t - \underline{\mathbf{X}}^t \right\|^2 \right] + \frac{\alpha^2 nB^2}{45\delta}. \tag{24}$$

Now let us consider the compression error of $\mathbf{X}$, namely, $\mathbb{E}\left[ \left\| \mathbf{X}^t - \underline{\mathbf{X}}^t \right\|^2 \right]$. By (12), we have that for any $\eta_3 > 0$,
$$\mathbb{E}\left[ \left\| \mathbf{X}^{t+1} - \underline{\mathbf{X}}^{t+1} \right\|^2 \right] = \mathbb{E}\left[ \left\| \left( \underline{\mathbf{X}}^{t+1} - \mathbf{X}^{t+\frac{1}{2}} \right) (\gamma(\mathbf{W} - \mathbf{I}) - \mathbf{I}) + \gamma\mathbf{X}^{t+\frac{1}{2}}(\mathbf{I} - \mathbf{J})(\mathbf{W} - \mathbf{I}) \right\|^2 \right]$$
$$\leq (1 + \eta_3) \left( 1 + 2\gamma \right)^2 \mathbb{E}\left[ \left\| \underline{\mathbf{X}}^{t+1} - \mathbf{X}^{t+\frac{1}{2}} \right\|^2 \right] + \left( 1 + \eta_3^{-1} \right) 4\gamma^2 \mathbb{E}\left[ \left\| \mathbf{X}_\perp^{t+\frac{1}{2}} \right\|^2 \right], \tag{25}$$
where we have used $\mathbf{JW} = \mathbf{J}$ in the equality and $\|\gamma(\mathbf{W} - \mathbf{I}) - \mathbf{I}\|_2 \leq \gamma\|\mathbf{W} - \mathbf{I}\|_2 + \|\mathbf{I}\|_2 \leq 1 + 2\gamma$ and $\|\mathbf{W} - \mathbf{I}\|_2 \leq 2$ in the inequality. For the second term in the RHS of (25), we have
$$\mathbb{E}\left[ \left\| \mathbf{X}_\perp^{t+\frac{1}{2}} \right\|^2 \right] \overset{(9)}{=} \mathbb{E}\left[ \left\| \mathbf{X}_\perp^t - \alpha\mathbf{Y}_\perp^t \right\|^2 \right] \leq 2\mathbb{E}\left[ \|\mathbf{X}_\perp^t\|^2 \right] + 2\alpha^2\mathbb{E}\left[ \|\mathbf{Y}_\perp^t\|^2 \right]$$
$$\overset{(18)}{\leq} 2\mathbb{E}\left[ \|\mathbf{X}_\perp^t\|^2 \right] + 2\alpha^2 nB^2\delta^{-1}. \tag{26}$$

Plugging (26) and (21) with $\eta_2 = \frac{1-\eta^2}{2\eta^2}$ into (25), we have

$$\mathbb{E}\left[\left\|\mathbf{X}^{t+1} - \underline{\mathbf{X}}^{t+1}\right\|^2\right] \leq \left(1 + \eta_3^{-1}\right) 4\gamma^2 \left(2\mathbb{E}\left[\left\|\mathbf{X}_\perp^t\right\|^2\right] + 2\alpha^2 nB^2\delta^{-1}\right) \tag{27}$$
$$+ \left(1 + \eta_3\right)\left(1 + 2\gamma\right)^2 \left(\frac{1+\eta^2}{2}\mathbb{E}\left[\left\|\mathbf{X}^t - \underline{\mathbf{X}}^t\right\|^2\right] + \frac{2\eta^2}{1-\eta^2}\alpha^2 nB^2\delta^{-1}\right).$$

Let $\eta_3 = \frac{1-\eta^2}{12}$ in (27). Then by $\gamma \leq \frac{1-\eta^2}{25}$, we obtain

$$\left(1 + \eta_3^{-1}\right) 8\gamma^2 \leq \frac{1}{1-\eta^2}, \ \ \left(1 + \eta_3\right)\left(1 + 2\gamma\right)^2 \frac{1+\eta^2}{2} \leq \frac{3+\eta^2}{4}, \ \ \frac{2\left(1 + \eta_3\right)\left(1 + 2\gamma\right)^2}{1-\eta^2} \leq \frac{4}{1-\eta^2}. \tag{28}$$

Thus, we have from (27) that

$$\mathbb{E}\left[\left\|\mathbf{X}^{t+1} - \underline{\mathbf{X}}^{t+1}\right\|^2\right] \leq \frac{1}{1-\eta^2}\mathbb{E}\left[\left\|\mathbf{X}_\perp^t\right\|^2\right] + \frac{3+\eta^2}{4}\mathbb{E}\left[\left\|\mathbf{X}^t - \underline{\mathbf{X}}^t\right\|^2\right] + \frac{5\alpha^2 nB^2}{\delta(1-\eta^2)}. \tag{29}$$

Denote $\Omega^t = \left(\mathbb{E}\left[\left\|\mathbf{X}_\perp^t\right\|^2\right], \mathbb{E}\left[\left\|\mathbf{X}^t - \underline{\mathbf{X}}^t\right\|^2\right]\right)^\top$. Then (24) and (29) imply $\Omega^{t+1} \leq \mathbf{A}\Omega^t + \mathbf{c}$, where

$$\mathbf{A} = \begin{pmatrix} \frac{3+\widehat{\rho}^2}{4} & \frac{4\eta\gamma(1-\widehat{\rho}^2)}{3} \\ \frac{1}{1-\eta^2} & \frac{3+\eta^2}{4} \end{pmatrix}, \quad \mathbf{c} = \begin{pmatrix} \frac{4nB^2\widehat{\rho}^2\alpha^2}{\delta(1-\widehat{\rho}^2)} + \frac{\alpha^2 nB^2}{45\delta} \\ \frac{5\alpha^2 nB^2}{\delta(1-\eta^2)} \end{pmatrix}.$$

For any $\mathbf{q} = (q_1, q_2)^\top \geq \mathbf{0}$, we have

$$\mathbf{q}^\top\Omega^{t+1} \leq \mathbf{q}^\top\Omega^t + \left(\mathbf{q}^\top\mathbf{A} - \mathbf{q}^\top\right)\Omega^t + \mathbf{q}^\top\mathbf{c}.$$

Take $q_1 = \frac{3}{1-\widehat{\rho}^2}$ and $q_2 = \frac{16\alpha}{1-\eta^2}$. We have $\mathbf{q}^\top\mathbf{A} - \mathbf{q}^\top \leq \left(-\frac{1}{4}, 0\right)$ by $\gamma \leq \frac{\alpha}{\eta}$ and $\alpha \leq \frac{(1-\eta^2)^2}{32}$. Thus $\mathbf{q}^\top\Omega^{t+1} \leq \mathbf{q}^\top\Omega^t - \frac{1}{4}\mathbb{E}[\|\mathbf{X}_\perp^t\|^2] + \mathbf{q}^\top\mathbf{c}$. Summing up the above inequality for all $t = 0, 1, \ldots, T-1$, we obtain

$$\frac{1}{T}\sum_{t=0}^{T-1}\mathbb{E}\left[\left\|\mathbf{X}_\perp^t\right\|^2\right] \leq \frac{4}{T}\left(\mathbf{q}^\top\Omega^0 - \mathbf{q}^\top\Omega^T\right) + 4\mathbf{q}^\top\mathbf{c}. \tag{30}$$

From $\mathbf{x}_i^0 = \overline{\mathbf{x}}^0 = \underline{\mathbf{x}}_i^0 = \mathbf{0}$, $\forall i \in [n]$, we have $\|\mathbf{X}_\perp^0\|^2 = 0$ and $\mathbb{E}[\|\mathbf{X}^0 - \underline{\mathbf{X}}^0\|^2] = 0$. Thus by the nonnegativity of $\mathbb{E}[\|\mathbf{X}_\perp^t\|^2]$, we have $\mathbf{q}^\top\Omega^0 - \mathbf{q}^\top\Omega^T \leq 0$. Hence

$$\frac{1}{T}\sum_{t=0}^{T-1}\mathbb{E}\left[\left\|\mathbf{X}_\perp^t\right\|^2\right] \leq \frac{12\alpha^2}{1-\widehat{\rho}^2}\left(\frac{4nB^2\widehat{\rho}^2}{\delta(1-\widehat{\rho}^2)} + \frac{nB^2}{45\delta}\right) + \frac{320\alpha^3 nB^2}{\delta(1-\eta^2)^2}.$$

The proof is then completed. $\qquad\qquad\qquad\qquad\qquad\qquad\qquad\qquad\qquad\qquad\qquad\qquad\qquad\quad \square$

To prove the convergence of Algorithm 1, we define an auxiliary sequence as follows

$$\mathbf{z}^t = \overline{\mathbf{x}}^t + \frac{\beta_1}{1-\beta_1}\left(\overline{\mathbf{x}}^t - \overline{\mathbf{x}}^{t-1}\right), \forall t \geq 0, \tag{31}$$

with $\overline{\mathbf{x}}^{-1} = \overline{\mathbf{x}}^0$. The lemma below shows the difference of two consecutive $\mathbf{z}$-points.

**Lemma A.3** *Let $\{\mathbf{z}^t\}$ be defined in (31). It holds that for all $t \geq 0$,*

$$\mathbf{z}^{t+1} - \mathbf{z}^t = \frac{\beta_1}{1-\beta_1}\frac{\alpha}{n}\sum_{i=1}^n \mathbf{m}_i^{t-1} \circ \left(\frac{1}{\sqrt{\mathbf{u}_i^{t-1} + \delta}} - \frac{1}{\sqrt{\mathbf{u}_i^t + \delta}}\right) - \frac{\alpha}{n}\sum_{i=1}^n \frac{\mathbf{g}_i^t}{\sqrt{\mathbf{u}_i^t + \delta}}. \tag{32}$$

*Proof.* By (11) and $(\mathbf{W} - \mathbf{I})\mathbf{J} = \mathbf{0}$, we have $\overline{\boldsymbol{x}}^{t+1} = \overline{\boldsymbol{x}}^{t+\frac{1}{2}}$. Hence, it follows from (8) and (9) that

$$\overline{\boldsymbol{x}}^{t+1} = \overline{\boldsymbol{x}}^t - \frac{\alpha}{n} \sum_{i=1}^{n} \frac{\boldsymbol{m}_i^t}{\sqrt{\boldsymbol{u}_i^t + \delta}}. \tag{33}$$

Thus by (31), we have

$$
\begin{aligned}
\boldsymbol{z}^{t+1} - \boldsymbol{z}^t &= \overline{\boldsymbol{x}}^{t+1} - \overline{\boldsymbol{x}}^t + \frac{\beta_1}{1 - \beta_1} \left( \overline{\boldsymbol{x}}^{t+1} - \overline{\boldsymbol{x}}^t \right) - \frac{\beta_1}{1 - \beta_1} \left( \overline{\boldsymbol{x}}^t - \overline{\boldsymbol{x}}^{t-1} \right) \\
&= \frac{1}{1 - \beta_1} \left( \overline{\boldsymbol{x}}^{t+1} - \overline{\boldsymbol{x}}^t \right) - \frac{\beta_1}{1 - \beta_1} \left( \overline{\boldsymbol{x}}^t - \overline{\boldsymbol{x}}^{t-1} \right) \\
&= \frac{1}{1 - \beta_1} \left( -\frac{\alpha}{n} \sum_{i=1}^{n} \frac{\boldsymbol{m}_i^t}{\sqrt{\boldsymbol{u}_i^t + \delta}} \right) - \frac{\beta_1}{1 - \beta_1} \left( -\frac{\alpha}{n} \sum_{i=1}^{n} \frac{\boldsymbol{m}_i^{t-1}}{\sqrt{\boldsymbol{u}_i^{t-1} + \delta}} \right) \\
&= \frac{1}{1 - \beta_1} \left( -\frac{\alpha}{n} \sum_{i=1}^{n} \frac{\beta_1 \boldsymbol{m}_i^{t-1} + (1 - \beta_1) \boldsymbol{g}_i^t}{\sqrt{\boldsymbol{u}_i^t + \delta}} \right) - \frac{\beta_1}{1 - \beta_1} \left( -\frac{\alpha}{n} \sum_{i=1}^{n} \frac{\boldsymbol{m}_i^{t-1}}{\sqrt{\boldsymbol{u}_i^{t-1} + \delta}} \right) \\
&= \frac{\beta_1}{1 - \beta_1} \frac{\alpha}{n} \sum_{i=1}^{n} \boldsymbol{m}_i^{t-1} \circ \left( \frac{1}{\sqrt{\boldsymbol{u}_i^{t-1} + \delta}} - \frac{1}{\sqrt{\boldsymbol{u}_i^t + \delta}} \right) - \frac{\alpha}{n} \sum_{i=1}^{n} \frac{\boldsymbol{g}_i^t}{\sqrt{\boldsymbol{u}_i^t + \delta}},
\end{aligned}
$$

which is the desired result. $\qquad\square$

**Lemma A.4** *Under Assumptions 1 and 4, it holds*

$$\left\| \frac{1}{n} \sum_{i=1}^{n} \left( \nabla f_i \left( \boldsymbol{x}_i^t \right) - \nabla f_i \left( \overline{\boldsymbol{x}}^t \right) \right) \right\|^2 \leq \frac{L^2}{n} \| \mathbf{X}_\perp^t \|^2, \tag{34}$$

*and*

$$\| \nabla f \left( \boldsymbol{z}^t \right) - \nabla f(\overline{\boldsymbol{x}}^t) \|^2 \leq \frac{\alpha^2 L^2 \beta_1^2 B^2}{\delta (1 - \beta_1)^2}. \tag{35}$$

*Proof.* First, by the $L$-smoothness of $f_i$ for each $i \in [n]$ and Young's inequality, we have

$$\left\| \frac{1}{n} \sum_{i=1}^{n} \left( \nabla f_i \left( \boldsymbol{x}_i^t \right) - \nabla f_i \left( \overline{\boldsymbol{x}}^t \right) \right) \right\|^2 \leq \frac{L^2}{n} \sum_{i=1}^{n} \| \boldsymbol{x}_i^t - \overline{\boldsymbol{x}}^t \|^2,$$

which indicates (34) by the definition of $\mathbf{X}_\perp^t$. Also, by the $L$-smoothness of $f$, it follows

$$\| \nabla f \left( \boldsymbol{z}^t \right) - \nabla f(\overline{\boldsymbol{x}}^t) \|^2 \leq L^2 \| \boldsymbol{z}^t - \overline{\boldsymbol{x}}^t \|^2 \overset{(31)}{=} \frac{L^2 \beta_1^2}{(1 - \beta_1)^2} \| \overline{\boldsymbol{x}}^t - \overline{\boldsymbol{x}}^{t-1} \|^2$$

$$\overset{(33)}{=} \frac{L^2 \beta_1^2}{(1 - \beta_1)^2} \left\| \frac{\alpha}{n} \sum_{i=1}^{n} \frac{\boldsymbol{m}_i^{t-1}}{\sqrt{\boldsymbol{u}_i^{t-1} + \delta}} \right\|^2 \leq \frac{L^2 \beta_1^2}{(1 - \beta_1)^2} \frac{\alpha^2}{n} \sum_{i=1}^{n} \left\| \frac{\boldsymbol{m}_i^{t-1}}{\sqrt{\boldsymbol{u}_i^{t-1} + \delta}} \right\|^2 \leq \frac{\alpha^2 L^2 \beta_1^2 B^2}{\delta (1 - \beta_1)^2},$$

where the last inequality holds by $\| \boldsymbol{m}_i^{t-1} \| \leq B$ from (14). This completes the proof. $\qquad\square$

**Lemma A.5** *Under Assumptions 1 and 4, it holds that*

$$\sum_{t=0}^{T-1} \mathbb{E} \left[ \| \boldsymbol{z}^{t+1} - \boldsymbol{z}^t \|^2 \right] \leq \frac{2\beta_1^2 \alpha^2 d B_\infty^2}{\delta (1 - \beta_1)^2} \tag{36}$$

$$+ 2\alpha^2 \left( \frac{24}{n\delta} T B^2 + \frac{6L^2}{n\delta} \mathbb{E} \left[ \sum_{t=0}^{T-1} \| \mathbf{X}_\perp^t \|^2 \right] + \frac{6}{\delta} \mathbb{E} \left[ \sum_{t=0}^{T-1} \| \nabla f(\overline{\boldsymbol{x}}^t) \|^2 \right] + \sum_{t=0}^{T-1} \frac{2d}{\delta^2} \left( 1 - \beta_2^{t+1} \right) B_\infty^4 \right).$$

*Proof.* By (32) and Young's inequality, we have

$$\sum_{t=0}^{T-1} \mathbb{E}\left[\left\|\boldsymbol{z}^{t+1}-\boldsymbol{z}^{t}\right\|^{2}\right] \leq \mathbb{E}\left[2\sum_{t=0}^{T-1}\left\|\frac{\beta_1}{1-\beta_1}\frac{\alpha}{n}\sum_{i=1}^{n}\boldsymbol{m}_i^{t-1}\circ\left(\frac{1}{\sqrt{\boldsymbol{u}_i^{t-1}+\delta}}-\frac{1}{\sqrt{\boldsymbol{u}_i^{t}+\delta}}\right)\right\|^{2}\right]$$
$$+\mathbb{E}\left[2\sum_{t=0}^{T-1}\left\|\frac{\alpha}{n}\sum_{i=1}^{n}\frac{\boldsymbol{g}_i^{t}}{\sqrt{\boldsymbol{u}_i^{t}+\delta}}\right\|^{2}\right]. \tag{37}$$

To bound the first term in the RHS of (37), we notice

$$\sum_{t=0}^{T-1}\left\|\frac{1}{n}\sum_{i=1}^{n}\boldsymbol{m}_i^{t-1}\circ\left(\frac{1}{\sqrt{\boldsymbol{u}_i^{t-1}+\delta}}-\frac{1}{\sqrt{\boldsymbol{u}_i^{t}+\delta}}\right)\right\|^{2}$$
$$\leq\sum_{t=0}^{T-1}\frac{1}{n}\sum_{i=1}^{n}\left\|\boldsymbol{m}_i^{t-1}\circ\left(\frac{1}{\sqrt{\boldsymbol{u}_i^{t-1}+\delta}}-\frac{1}{\sqrt{\boldsymbol{u}_i^{t}+\delta}}\right)\right\|^{2}$$
$$\leq B_\infty^2\sum_{t=0}^{T-1}\frac{1}{n}\sum_{i=1}^{n}\left\|\frac{1}{\sqrt{\boldsymbol{u}_i^{t-1}+\delta}}-\frac{1}{\sqrt{\boldsymbol{u}_i^{t}+\delta}}\right\|^{2}$$
$$\leq B_\infty^2\sum_{t=0}^{T-1}\frac{1}{n}\sum_{i=1}^{n}\left\|\frac{1}{\sqrt{\boldsymbol{u}_i^{t-1}+\delta}}-\frac{1}{\sqrt{\boldsymbol{u}_i^{t}+\delta}}\right\|_{1}\left\|\frac{1}{\sqrt{\boldsymbol{u}_i^{t-1}+\delta}}-\frac{1}{\sqrt{\boldsymbol{u}_i^{t}+\delta}}\right\|_{\infty}. \tag{38}$$

In addition, it holds $\left\|\frac{1}{\sqrt{\boldsymbol{u}_i^{t-1}+\delta}}-\frac{1}{\sqrt{\boldsymbol{u}_i^{t}+\delta}}\right\|_{\infty} \leq \frac{1}{\sqrt{\delta}}$, and

$$\sum_{t=0}^{T-1}\frac{1}{n}\sum_{i=1}^{n}\left\|\frac{1}{\sqrt{\boldsymbol{u}_i^{t-1}+\delta}}-\frac{1}{\sqrt{\boldsymbol{u}_i^{t}+\delta}}\right\|_{1}=\sum_{t=0}^{T-1}\frac{1}{n}\sum_{i=1}^{n}\left(\left\|\frac{1}{\sqrt{\boldsymbol{u}_i^{t-1}+\delta}}\right\|_{1}-\left\|\frac{1}{\sqrt{\boldsymbol{u}_i^{t}+\delta}}\right\|_{1}\right)\leq\frac{d}{\sqrt{\delta}}, \tag{39}$$

where the equality holds because $\boldsymbol{u}_i^t$ is nondecreasing with $t$ for each $i \in [n]$. Therefore, it follows from (38) that

$$\sum_{t=0}^{T-1}\left\|\frac{1}{n}\sum_{i=1}^{n}\boldsymbol{m}_i^{t-1}\circ\left(\frac{1}{\sqrt{\boldsymbol{u}_i^{t-1}+\delta}}-\frac{1}{\sqrt{\boldsymbol{u}_i^{t}+\delta}}\right)\right\|^{2}\leq\frac{dB_\infty^2}{\delta}. \tag{40}$$

To bound the second term in RHS of (37), we first apply Young's inequality to have

$$\mathbb{E}\left[\sum_{t=0}^{T-1}\left\|\frac{1}{n}\sum_{i=1}^{n}\frac{\boldsymbol{g}_i^{t}}{\sqrt{\boldsymbol{u}_i^{t}+\delta}}\right\|^{2}\right]$$
$$\leq 2\mathbb{E}\left[\sum_{t=0}^{T-1}\left\|\frac{1}{n}\sum_{i=1}^{n}\frac{\boldsymbol{g}_i^{t}}{\sqrt{\overline{\boldsymbol{u}}^{t-1}+\delta}}\right\|^{2}\right]+2\mathbb{E}\left[\sum_{t=0}^{T-1}\left\|\frac{1}{n}\sum_{i=1}^{n}\left(\frac{\boldsymbol{g}_i^{t}}{\sqrt{\overline{\boldsymbol{u}}^{t-1}+\delta}}-\frac{\boldsymbol{g}_i^{t}}{\sqrt{\boldsymbol{u}_i^{t}+\delta}}\right)\right\|^{2}\right]. \tag{41}$$

Notice

$$\sum_{t=0}^{T-1}\left\|\frac{1}{n}\sum_{i=1}^{n}\left(\frac{\boldsymbol{g}_i^{t}}{\sqrt{\overline{\boldsymbol{u}}^{t-1}+\delta}}-\frac{\boldsymbol{g}_i^{t}}{\sqrt{\boldsymbol{u}_i^{t}+\delta}}\right)\right\|^{2}\leq\sum_{t=0}^{T-1}\frac{1}{n}\sum_{i=1}^{n}\left\|\frac{\boldsymbol{g}_i^{t}}{\sqrt{\overline{\boldsymbol{u}}^{t-1}+\delta}}-\frac{\boldsymbol{g}_i^{t}}{\sqrt{\boldsymbol{u}_i^{t}+\delta}}\right\|^{2}$$
$$\leq B_\infty^2\sum_{t=0}^{T-1}\frac{1}{n}\sum_{i=1}^{n}\left\|\frac{1}{\sqrt{\overline{\boldsymbol{u}}^{t-1}+\delta}}-\frac{1}{\sqrt{\boldsymbol{u}_i^{t}+\delta}}\right\|_{1}\left\|\frac{1}{\sqrt{\overline{\boldsymbol{u}}^{t-1}+\delta}}-\frac{1}{\sqrt{\boldsymbol{u}_i^{t}+\delta}}\right\|_{\infty}. \tag{42}$$

In addition, it holds that $\left\| \frac{1}{\sqrt{\overline{\boldsymbol{u}}^{t-1}+\delta}} - \frac{1}{\sqrt{\boldsymbol{u}_i^t+\delta}} \right\|_\infty \leq \frac{1}{\sqrt{\delta}}$, and

$$\left\| \frac{1}{\sqrt{\overline{\boldsymbol{u}}^{t-1}+\delta}} - \frac{1}{\sqrt{\boldsymbol{u}_i^t+\delta}} \right\|_1 = \left\| \frac{\overline{\boldsymbol{u}}^{t-1} - \boldsymbol{u}_i^t}{\sqrt{(\boldsymbol{u}_i^t+\delta)(\overline{\boldsymbol{u}}^{t-1}+\delta)}\left(\sqrt{\boldsymbol{u}_i^t+\delta}+\sqrt{\overline{\boldsymbol{u}}^{t-1}+\delta}\right)} \right\|_1$$

$$\leq \frac{1}{2\delta^{\frac{3}{2}}}\|\boldsymbol{u}_i^t - \overline{\boldsymbol{u}}^{t-1}\|_1 \leq \frac{d}{2\delta^{\frac{3}{2}}}\|\boldsymbol{u}_i^t - \overline{\boldsymbol{u}}^{t-1}\|_\infty \overset{(14)}{\leq} \frac{d}{\delta^{\frac{3}{2}}}\left(1-\beta_2^{t+1}\right)B_\infty^2. \tag{43}$$

Therefore, we have from (41) and (42) that

$$\mathbb{E}\left[\sum_{t=0}^{T-1}\left\|\frac{1}{n}\sum_{i=1}^n \frac{\boldsymbol{g}_i^t}{\sqrt{\boldsymbol{u}_i^t+\delta}}\right\|^2\right] \leq 2\mathbb{E}\left[\sum_{t=0}^{T-1}\left\|\frac{1}{n}\sum_{i=1}^n \frac{\boldsymbol{g}_i^t}{\sqrt{\overline{\boldsymbol{u}}^{t-1}+\delta}}\right\|^2\right] + \sum_{t=0}^{T-1}\frac{2d}{\delta^2}\left(1-\beta_2^{t+1}\right)B_\infty^4. \tag{44}$$

Now we bound the first term in the RHS of (44) as follows

$$2\mathbb{E}\left[\sum_{t=0}^{T-1}\left\|\frac{1}{n}\sum_{i=1}^n \frac{\boldsymbol{g}_i^t}{\sqrt{\overline{\boldsymbol{u}}^{t-1}+\delta}}\right\|^2\right]$$

$$\leq \frac{2}{\delta}\mathbb{E}\left[\sum_{t=0}^{T-1}\left\|\frac{1}{n}\sum_{i=1}^n \left(\boldsymbol{g}_i^t - \nabla f_i(\boldsymbol{x}_i^t) + \nabla f_i(\boldsymbol{x}_i^t) - \nabla f(\overline{\boldsymbol{x}}^t) + \nabla f(\overline{\boldsymbol{x}}^t)\right)\right\|^2\right]$$

$$\leq \frac{6}{\delta}\mathbb{E}\left[\sum_{t=0}^{T-1}\left\|\frac{1}{n}\sum_{i=1}^n \left(\boldsymbol{g}_i^t - \nabla f_i(\boldsymbol{x}_i^t)\right)\right\|^2 + \sum_{t=0}^{T-1}\left\|\frac{1}{n}\sum_{i=1}^n \left(\nabla f_i(\boldsymbol{x}_i^t) - \nabla f_i(\overline{\boldsymbol{x}}^t)\right)\right\|^2 + \sum_{t=0}^{T-1}\left\|\nabla f(\overline{\boldsymbol{x}}^t)\right\|^2\right]$$

$$= \frac{6}{n^2\delta}\mathbb{E}\left[\sum_{t=0}^{T-1}\sum_{i=1}^n \left\|\boldsymbol{g}_i^t - \nabla f_i(\boldsymbol{x}_i^t)\right\|^2\right] + \frac{6}{\delta}\mathbb{E}\left[\sum_{t=0}^{T-1}\left\|\frac{1}{n}\sum_{i=1}^n \left(\nabla f_i(\boldsymbol{x}_i^t) - \nabla f_i(\overline{\boldsymbol{x}}^t)\right)\right\|^2\right] + \frac{6}{\delta}\mathbb{E}\left[\sum_{t=0}^{T-1}\left\|\nabla f(\overline{\boldsymbol{x}}^t)\right\|^2\right]$$

$$\leq \frac{6}{n^2\delta}4nTB^2 + \frac{6L^2}{n\delta}\mathbb{E}\left[\sum_{t=0}^{T-1}\left\|\mathbf{X}_\perp^t\right\|^2\right] + \frac{6}{\delta}\mathbb{E}\left[\sum_{t=0}^{T-1}\left\|\nabla f(\overline{\boldsymbol{x}}^t)\right\|^2\right], \tag{45}$$

where the equality holds because $\boldsymbol{g}_1^t, \boldsymbol{g}_2^t, \ldots, \boldsymbol{g}_n^t$ are conditionally independent of each other, and in the last inequality, we have used (34). Plugging (45) into (44) gives

$$\mathbb{E}\left[\sum_{t=0}^{T-1}\left\|\frac{1}{n}\sum_{i=1}^n \frac{\boldsymbol{g}_i^t}{\sqrt{\boldsymbol{u}_i^t+\delta}}\right\|^2\right] \tag{46}$$

$$\leq \frac{24}{n\delta}TB^2 + \frac{6L^2}{n\delta}\mathbb{E}\left[\sum_{t=0}^{T-1}\left\|\mathbf{X}_\perp^t\right\|^2\right] + \frac{6}{\delta}\mathbb{E}\left[\sum_{t=0}^{T-1}\left\|\nabla f_i(\overline{\boldsymbol{x}}^t)\right\|^2\right] + \sum_{t=0}^{T-1}\frac{2d}{\delta^2}\left(1-\beta_2^{t+1}\right)B_\infty^4.$$

The proof is then completed by substituting (40) and (46) into (37). $\qquad\square$

**Lemma A.6** *Under Assumptions 1 and 4, it holds*

$$\mathbb{E}_t\left[\left\langle \nabla f\left(\boldsymbol{z}^t\right), \frac{1}{n}\sum_{i=1}^n \frac{\boldsymbol{g}_i^t}{\sqrt{\overline{\boldsymbol{u}}^{t-1}+\delta}}\right\rangle\right] \tag{47}$$

$$\geq \frac{1}{2\sqrt{B_\infty^2+\delta}}\|\nabla f\left(\overline{\boldsymbol{x}}^t\right)\|^2 - \frac{L^2}{n}\left(\frac{\sqrt{B_\infty^2+\delta}}{\delta} + \frac{1}{2\sqrt{\delta}}\right)\|\mathbf{X}_\perp^t\|^2 - \frac{\alpha^2\beta_1^2 L^2 B^2}{\delta(1-\beta_1)^2}\left(\frac{\sqrt{B_\infty^2+\delta}}{\delta} + \frac{1}{2\sqrt{\delta}}\right).$$

*Proof.* By Assumption 4, it holds that

$$
\mathbb{E}_t\left[\left\langle \nabla f\left(\boldsymbol{z}^t\right), \frac{1}{n}\sum_{i=1}^n \frac{\boldsymbol{g}_i^t}{\sqrt{\overline{\boldsymbol{u}}^{t-1}+\delta}}\right\rangle\right] = \left\langle \nabla f\left(\boldsymbol{z}^t\right), \frac{1}{n}\sum_{i=1}^n \frac{\nabla f_i(\boldsymbol{x}_i^t)}{\sqrt{\overline{\boldsymbol{u}}^{t-1}+\delta}}\right\rangle
$$

$$
= \left\langle \nabla f\left(\boldsymbol{z}^t\right) - \nabla f(\overline{\boldsymbol{x}}^t), \frac{1}{n\sqrt{\overline{\boldsymbol{u}}^{t-1}+\delta}}\sum_{i=1}^n\left(\nabla f_i(\boldsymbol{x}_i^t)-\nabla f(\overline{\boldsymbol{x}}^t)\right)\right\rangle + \left\langle \nabla f\left(\boldsymbol{z}^t\right) - \nabla f(\overline{\boldsymbol{x}}^t), \frac{\nabla f(\overline{\boldsymbol{x}}^t)}{\sqrt{\overline{\boldsymbol{u}}^{t-1}+\delta}}\right\rangle
$$

$$
+ \left\langle \nabla f(\overline{\boldsymbol{x}}^t), \frac{1}{n\sqrt{\overline{\boldsymbol{u}}^{t-1}+\delta}}\sum_{i=1}^n\left(\nabla f_i(\boldsymbol{x}_i^t)-\nabla f(\overline{\boldsymbol{x}}^t)\right)\right\rangle + \left\langle \nabla f(\overline{\boldsymbol{x}}^t), \frac{\nabla f(\overline{\boldsymbol{x}}^t)}{\sqrt{\overline{\boldsymbol{u}}^{t-1}+\delta}}\right\rangle. \tag{48}
$$

Next we bound each of the four terms in the RHS of (48). For the first term in the RHS of (48), we use Young's inequality, (34), and (35) to have

$$
\left\langle \nabla f\left(\boldsymbol{z}^t\right) - \nabla f(\overline{\boldsymbol{x}}^t), \frac{1}{n\sqrt{\overline{\boldsymbol{u}}^{t-1}+\delta}}\sum_{i=1}^n\left(\nabla f_i(\boldsymbol{x}_i^t)-\nabla f(\overline{\boldsymbol{x}}^t)\right)\right\rangle
$$

$$
\geq -\frac{1}{2\sqrt{\delta}}\left(\left\|\nabla f\left(\boldsymbol{z}^t\right)-\nabla f(\overline{\boldsymbol{x}}^t)\right\|^2 + \left\|\frac{1}{n}\sum_{i=1}^n\left(\nabla f_i\left(\boldsymbol{x}_i^t\right)-\nabla f_i\left(\overline{\boldsymbol{x}}^t\right)\right)\right\|^2\right)
$$

$$
\geq -\frac{1}{2\sqrt{\delta}}\left(\frac{\alpha^2\beta_1^2 L^2 B^2}{\delta(1-\beta_1)^2} + \frac{L^2}{n}\|\mathbf{X}_\perp^t\|^2\right), \tag{49}
$$

where we have used $\overline{\boldsymbol{u}}^{t-1}\geq\mathbf{0}$ in the first inequality. For the second term in the RHS of (48), we have

$$
\left\langle \nabla f\left(\boldsymbol{z}^t\right) - \nabla f(\overline{\boldsymbol{x}}^t), \frac{\nabla f(\overline{\boldsymbol{x}}^t)}{\sqrt{\overline{\boldsymbol{u}}^{t-1}+\delta}}\right\rangle \tag{50}
$$

$$
\geq -\frac{\left\|\nabla f\left(\overline{\boldsymbol{x}}^t\right)\right\|^2}{4\sqrt{B_\infty^2+\delta}} - \frac{\sqrt{B_\infty^2+\delta}}{\delta}\left\|\nabla f\left(\boldsymbol{z}^t\right)-\nabla f(\overline{\boldsymbol{x}}^t)\right\|^2 \overset{(35)}{\geq} -\frac{\left\|\nabla f\left(\overline{\boldsymbol{x}}^t\right)\right\|^2}{4\sqrt{B_\infty^2+\delta}} - \frac{\alpha^2\beta_1^2 L^2\sqrt{B_\infty^2+\delta}B^2}{\delta^2(1-\beta_1)^2},
$$

where the first inequality follows from Young's inequality and $\overline{\boldsymbol{u}}^{t-1}\geq\mathbf{0}$. For the third term in the RHS of (48), we have from Young's inequality and (34) that

$$
\left\langle \nabla f(\overline{\boldsymbol{x}}^t), \frac{1}{n\sqrt{\overline{\boldsymbol{u}}^{t-1}+\delta}}\sum_{i=1}^n\left(\nabla f_i(\boldsymbol{x}_i^t)-\nabla f(\overline{\boldsymbol{x}}^t)\right)\right\rangle
$$

$$
\geq -\frac{\left\|\nabla f\left(\overline{\boldsymbol{x}}^t\right)\right\|^2}{4\sqrt{B_\infty^2+\delta}} - \frac{\sqrt{B_\infty^2+\delta}}{\delta}\left\|\frac{1}{n}\sum_{i=1}^n\left(\nabla f_i\left(\boldsymbol{x}_i^t\right)-\nabla f_i\left(\overline{\boldsymbol{x}}^t\right)\right)\right\|^2
$$

$$
\geq -\frac{\left\|\nabla f\left(\overline{\boldsymbol{x}}^t\right)\right\|^2}{4\sqrt{B_\infty^2+\delta}} - \frac{L^2\sqrt{B_\infty^2+\delta}}{n\delta}\|\mathbf{X}_\perp^t\|^2. \tag{51}
$$

The last term in the RHS of (48) can be bounded as

$$
\left\langle \nabla f\left(\overline{\boldsymbol{x}}^t\right), \frac{1}{n}\sum_{i=1}^n \frac{\nabla f\left(\overline{\boldsymbol{x}}^t\right)}{\sqrt{\overline{\boldsymbol{u}}^{t-1}+\delta}}\right\rangle \geq \frac{1}{\sqrt{B_\infty^2+\delta}}\|\nabla f\left(\overline{\boldsymbol{x}}^t\right)\|^2. \tag{52}
$$

Substituting (49)–(52) into (48) and rearranging terms yields the desired result. $\qquad\square$

Now we are ready to show the main convergence result of Algorithm 1.

**Theorem A.1 (Complete statement of Theorem 4.1)** *Under Assumptions 1–4, let $\alpha > 0$ and $\gamma > 0$ satisfy*

$$
\alpha \leq \min\left\{\frac{\delta}{24L\sqrt{B_\infty^2+\delta}}, \frac{(1-\eta^2)^2}{32}\right\}, \quad \gamma \leq \min\left\{\frac{1-\widehat{\rho}^2}{60\eta}, \frac{1-\eta^2}{25}, \frac{\alpha}{\eta}\right\}. \tag{53}
$$

*Then with C defined in (15), it holds*

$$
\frac{\alpha}{4\sqrt{B_\infty^2+\delta}} \sum_{t=0}^{T-1} \mathbb{E}\left[\|\nabla f\left(\overline{\boldsymbol{x}}^t\right)\|^2\right] \le \mathbb{E}\left[f\left(\boldsymbol{z}^0\right) - f\left(\boldsymbol{z}^T\right)\right] + \frac{\alpha\beta_1}{1-\beta_1}\frac{dB_\infty^2}{\sqrt{\delta}} + \frac{L\beta_1^2\alpha^2 dB_\infty^2}{\delta(1-\beta_1)^2}
$$

$$
+ \frac{24L\alpha^2}{n\delta}TB^2 + \left(\frac{6L^3\alpha^2}{n\delta} + \frac{L^2\alpha}{n}\left(\frac{\sqrt{B_\infty^2+\delta}}{\delta} + \frac{1}{2\sqrt{\delta}}\right)\right)\alpha^2 CT + L\alpha^2 B_\infty^4 \frac{2d}{\delta^2}\sum_{t=0}^{T-1}\left(1-\beta_2^{t+1}\right) \quad (54)
$$

$$
+ \alpha\sum_{t=0}^{T-1}\left(\frac{dB_\infty^4}{\delta^{\frac{3}{2}}}\left(1-\beta_2^{t+1}\right) + \frac{\alpha^2\beta_1^2 L^2 B^2}{\delta(1-\beta_1)^2}\left(\frac{\sqrt{B_\infty^2+\delta}}{\delta} + \frac{1}{2\sqrt{\delta}}\right)\right).
$$

*Proof.* By the $L$-smoothness of $f$, we have

$$
f\left(\boldsymbol{z}^{t+1}\right) \le f\left(\boldsymbol{z}^t\right) + \left\langle\nabla f\left(\boldsymbol{z}^t\right), \boldsymbol{z}^{t+1} - \boldsymbol{z}^t\right\rangle + \frac{L}{2}\left\|\boldsymbol{z}^{t+1} - \boldsymbol{z}^t\right\|^2,
$$

which together with (32) gives

$$
f\left(\boldsymbol{z}^{t+1}\right) \le f\left(\boldsymbol{z}^t\right) - \alpha\left\langle\nabla f\left(\boldsymbol{z}^t\right), \frac{1}{n}\sum_{i=1}^n \frac{\boldsymbol{g}_i^t}{\sqrt{\boldsymbol{u}_i^t+\delta}}\right\rangle + \frac{L}{2}\left\|\boldsymbol{z}^{t+1} - \boldsymbol{z}^t\right\|^2
$$

$$
+ \frac{\beta_1}{1-\beta_1}\left\langle\nabla f\left(\boldsymbol{z}^t\right), \frac{\alpha}{n}\sum_{i=1}^n \boldsymbol{m}_i^{t-1}\circ\left(\frac{1}{\sqrt{\boldsymbol{u}_i^{t-1}+\delta}} - \frac{1}{\sqrt{\boldsymbol{u}_i^t+\delta}}\right)\right\rangle.
$$

Take expectation, sum up over $t$, and rearrange terms of the above inequality. We have

$$
\alpha\sum_{t=0}^{T-1}\mathbb{E}\left[\left\langle\nabla f\left(\boldsymbol{z}^t\right), \frac{1}{n}\sum_{i=1}^n \frac{\boldsymbol{g}_i^t}{\sqrt{\boldsymbol{u}_i^t+\delta}}\right\rangle\right] \le \mathbb{E}\left[f\left(\boldsymbol{z}^0\right) - f\left(\boldsymbol{z}^T\right)\right] + \frac{L}{2}\mathbb{E}\sum_{t=0}^{T-1}\left[\left\|\boldsymbol{z}^{t+1}-\boldsymbol{z}^t\right\|^2\right]
$$
$$
+ \frac{\alpha\beta_1}{1-\beta_1}\sum_{t=0}^{T-1}\mathbb{E}\left[\left\langle\nabla f\left(\boldsymbol{z}^t\right), \frac{1}{n}\sum_{i=1}^n \boldsymbol{m}_i^{t-1}\circ\left(\frac{1}{\sqrt{\boldsymbol{u}_i^{t-1}+\delta}} - \frac{1}{\sqrt{\boldsymbol{u}_i^t+\delta}}\right)\right\rangle\right]. \quad (55)
$$

Below we bound the two inner-product terms on the LHS and RHS of (55). First,

$$
\sum_{t=0}^{T-1}\mathbb{E}\left[\left\langle\nabla f\left(\boldsymbol{z}^t\right), \frac{1}{n}\sum_{i=1}^n \boldsymbol{m}_i^{t-1}\circ\left(\frac{1}{\sqrt{\boldsymbol{u}_i^{t-1}+\delta}} - \frac{1}{\sqrt{\boldsymbol{u}_i^t+\delta}}\right)\right\rangle\right]
$$

$$
\le \sum_{t=0}^{T-1}\mathbb{E}\left[\|\nabla f\left(\boldsymbol{z}^t\right)\|_\infty\left\|\frac{1}{n}\sum_{i=1}^n \boldsymbol{m}_i^{t-1}\circ\left(\frac{1}{\sqrt{\boldsymbol{u}_i^{t-1}+\delta}} - \frac{1}{\sqrt{\boldsymbol{u}_i^t+\delta}}\right)\right\|_1\right]
$$

$$
\le \sum_{t=0}^{T-1}\mathbb{E}\left[\|\nabla f\left(\boldsymbol{z}^t\right)\|_\infty\frac{1}{n}\sum_{i=1}^n\|\boldsymbol{m}_i^{t-1}\|_\infty\left\|\frac{1}{\sqrt{\boldsymbol{u}_i^{t-1}+\delta}} - \frac{1}{\sqrt{\boldsymbol{u}_i^t+\delta}}\right\|_1\right]
$$

$$
\le \sum_{t=0}^{T-1}B_\infty^2\mathbb{E}\left[\frac{1}{n}\sum_{i=1}^n\left\|\frac{1}{\sqrt{\boldsymbol{u}_i^{t-1}+\delta}} - \frac{1}{\sqrt{\boldsymbol{u}_i^t+\delta}}\right\|_1\right] \overset{(39)}{\le} \frac{dB_\infty^2}{\sqrt{\delta}}, \quad (56)
$$

where in the third inequality, we have used $\|\nabla f(\boldsymbol{z}^t)\|_\infty \le B_\infty$ and $\|\boldsymbol{m}_i^{t-1}\|_\infty \le B_\infty$ by Assumption 4 and Lemma A.1. Second, we write

$$\left\langle \nabla f(\boldsymbol{z}^t), \frac{1}{n}\sum_{i=1}^n \frac{\boldsymbol{g}_i^t}{\sqrt{\boldsymbol{u}_i^t + \delta}} \right\rangle \tag{57}$$

$$= \left\langle \nabla f(\boldsymbol{z}^t), \frac{1}{n}\sum_{i=1}^n \left( \frac{\boldsymbol{g}_i^t}{\sqrt{\boldsymbol{u}_i^t + \delta}} - \frac{\boldsymbol{g}_i^t}{\sqrt{\overline{\boldsymbol{u}}^{t-1} + \delta}} \right) \right\rangle + \left\langle \nabla f(\boldsymbol{z}^t), \frac{1}{n}\sum_{i=1}^n \frac{\boldsymbol{g}_i^t}{\sqrt{\overline{\boldsymbol{u}}^{t-1} + \delta}} \right\rangle.$$

By $\|f(\boldsymbol{z}^t)\|_\infty \le B_\infty$ and $\|\boldsymbol{g}_i^t\|_\infty \le B_\infty$ from Assumption 4, we have

$$\left\langle \nabla f(\boldsymbol{z}^t), \frac{1}{n}\sum_{i=1}^n \left( \frac{\boldsymbol{g}_i^t}{\sqrt{\boldsymbol{u}_i^t + \delta}} - \frac{\boldsymbol{g}_i^t}{\sqrt{\overline{\boldsymbol{u}}^{t-1} + \delta}} \right) \right\rangle \tag{58}$$

$$\ge -B_\infty^2 \frac{1}{n}\sum_{i=1}^n \left\| \frac{1}{\sqrt{\overline{\boldsymbol{u}}^{t-1} + \delta}} - \frac{1}{\sqrt{\boldsymbol{u}_i^t + \delta}} \right\|_1 \overset{(43)}{\ge} -\frac{d}{\delta^{\frac{3}{2}}}\left(1 - \beta_2^{t+1}\right)B_\infty^4.$$

Substituting (47) and (58) into (57) after taking conditional expectation, we obtain

$$\mathbb{E}_t\left[ \left\langle \nabla f(\boldsymbol{z}^t), \frac{1}{n}\sum_{i=1}^n \frac{\boldsymbol{g}_i^t}{\sqrt{\boldsymbol{u}_i^t + \delta}} \right\rangle \right] \ge -\frac{dB_\infty^4}{\delta^{\frac{3}{2}}}\left(1 - \beta_2^{t+1}\right) + \frac{1}{2\sqrt{B_\infty^2 + \delta}}\|\nabla f(\overline{\boldsymbol{x}}^t)\|^2 \tag{59}$$

$$-\frac{L^2}{n}\left( \frac{\sqrt{B_\infty^2 + \delta}}{\delta} + \frac{1}{2\sqrt{\delta}} \right)\|\mathbf{X}_\perp^t\|^2 - \frac{\alpha^2 \beta_1^2 L^2 B^2}{\delta(1-\beta_1)^2}\left( \frac{\sqrt{B_\infty^2 + \delta}}{\delta} + \frac{1}{2\sqrt{\delta}} \right).$$

Now plugging (36), (56) and (59) after full expectation into (55) and rearranging terms gives

$$\left( \frac{\alpha}{2\sqrt{B_\infty^2 + \delta}} - \frac{6L\alpha^2}{\delta} \right)\sum_{t=0}^{T-1}\mathbb{E}[\|\nabla f(\overline{\boldsymbol{x}}^t)\|^2] \le \mathbb{E}\left[ f(\boldsymbol{z}^0) - f(\boldsymbol{z}^T) \right] + \frac{\alpha\beta_1}{1-\beta_1}\frac{dB_\infty^2}{\sqrt{\delta}}$$

$$+ \frac{L}{2}\left( \frac{2\beta_1^2\alpha^2 dB_\infty^2}{\delta(1-\beta_1)^2} + 2\alpha^2\left( \frac{24}{n\delta}TB^2 + \frac{6L^2}{n\delta}\mathbb{E}\left[ \sum_{t=0}^{T-1}\|\mathbf{X}_\perp^t\|^2 \right] + \sum_{t=0}^{T-1}\frac{2d}{\delta^2}\left(1 - \beta_2^{t+1}\right)B_\infty^4 \right) \right)$$

$$+ \alpha\sum_{t=0}^{T-1}\left( \frac{dB_\infty^4}{\delta^{\frac{3}{2}}}\left(1 - \beta_2^{t+1}\right) + \frac{L^2}{n}\left( \frac{\sqrt{B_\infty^2 + \delta}}{\delta} + \frac{1}{2\sqrt{\delta}} \right)\mathbb{E}[\|\mathbf{X}_\perp^t\|^2] + \frac{\alpha^2\beta_1^2 L^2 B^2}{\delta(1-\beta_1)^2}\left( \frac{\sqrt{B_\infty^2 + \delta}}{\delta} + \frac{1}{2\sqrt{\delta}} \right) \right).$$

Plug (15) into the inequality above, notice $\frac{\alpha}{2\sqrt{B_\infty^2 + \delta}} - \frac{6L\alpha^2}{\delta} \ge \frac{\alpha}{4\sqrt{B_\infty^2 + \delta}}$, and rearrange terms. We obtain the desired result and complete the proof. $\qquad\square$

Below we give a few choices of the algorithmic parameters and give the convergence rate of Algorithm 1.

**Theorem A.2 (Complete statement of Theorem 4.2)** *Suppose that the conditions assumed in Theorem A.1 hold with $\alpha = \frac{4\theta\sqrt{n(B_\infty^2 + \delta)}}{\sqrt{T}}$ for a sufficiently large $T \ge n$ and some $\theta \in (\frac{n}{Le}, \frac{n}{L})$. Then with $C$ defined in (15), we have the following results.*

*(i) Let $\delta = \omega^2 B_\infty^2 \frac{\sqrt{T}}{\sqrt{n}}$ for some universal constant $\omega > 0$. Set $\beta_2 \in \left[ \frac{T}{T+1}, \left(\frac{\theta L}{n}\right)^{\frac{1}{T}} \right]$. Then*

$$\frac{1}{T}\sum_{t=0}^{T-1}\mathbb{E}\left[ \|\nabla f(\overline{\boldsymbol{x}}^t)\|^2 + \frac{1}{n}\|\mathbf{X}_\perp^t\|^2 \right]$$

$$= O\left( \frac{1}{\sqrt{nT}}\left( f(\boldsymbol{z}^0) - f^* + LB^2 + dB_\infty^2(1+L) + C(1+L^2)B_\infty^2 \right) + C(1+L^2)\frac{B_\infty^2}{T} + \frac{nL^2B^2}{T} \right). \tag{60}$$

*(ii) Let $\delta = O(1)$ be a universal positive constant. Suppose $\frac{T}{T+1} \leq (\frac{\theta L}{\sqrt{nT}})^{\frac{1}{T}} < 1$. Set $\beta_2 \in \left[\frac{T}{T+1}, (\frac{\theta L}{\sqrt{nT}})^{\frac{1}{T}}\right]$. Then*

$$\frac{1}{T}\sum_{t=0}^{T-1}\mathbb{E}\left[\|\nabla f\left(\overline{\boldsymbol{x}}^t\right)\|^2 + \frac{1}{n}\|\mathbf{X}_\perp^t\|^2\right] \tag{61}$$

$$= O\left(\frac{1}{\sqrt{nT}}\left(f(\boldsymbol{z}^0) - f^* + dB_\infty^3 + LB^2B_\infty^2 + dB_\infty^2 L + dLB_\infty^5\right) + \frac{CB_\infty^2}{T}\left(1 + L^2B_\infty^2\right) + \frac{L^2B^2B_\infty^4}{T}\right).$$

*Proof.* From $\alpha \leq \frac{\delta}{24L\sqrt{B_\infty^2+\delta}}$, it follows that $\frac{L\alpha}{\sqrt{\delta}} \leq \frac{\sqrt{\delta}}{24\sqrt{B_\infty^2+\delta}} \leq \frac{1}{24}$. Hence,

$$\frac{6L^3\alpha^2}{n\delta} \leq \frac{L^2\alpha}{4n\sqrt{\delta}},$$

$$\frac{\alpha\beta_1}{1-\beta_1}\frac{dB_\infty^2}{\sqrt{\delta}} + \frac{L\beta_1^2\alpha^2 dB_\infty^2}{\delta(1-\beta_1)^2} \leq \frac{\alpha\beta_1}{1-\beta_1}\frac{dB_\infty^2}{\sqrt{\delta}}\left(1 + \frac{\beta_1}{24(1-\beta_1)}\right) = \frac{\alpha\beta_1(24-23\beta_1)}{24(1-\beta_1)^2}\frac{dB_\infty^2}{\sqrt{\delta}},$$

and

$$L\alpha^2 B_\infty^4 \frac{2d}{\delta^2}\sum_{t=0}^{T-1}\left(1-\beta_2^{t+1}\right) + \alpha\sum_{t=0}^{T-1}\frac{dB_\infty^4}{\delta^{\frac{3}{2}}}\left(1-\beta_2^{t+1}\right)$$

$$\leq \frac{13\alpha dB_\infty^4}{12\delta^{\frac{3}{2}}}\sum_{t=0}^{T-1}\left(1-\beta_2^{t+1}\right) = \frac{13\alpha dB_\infty^4}{12\delta^{\frac{3}{2}}}\left(T - \frac{\beta_2(1-\beta_2^T)}{1-\beta_2}\right).$$

Now dividing both sides of (54) by $\frac{\alpha T}{4\sqrt{B_\infty^2+\delta}} = \theta\sqrt{nT}$, using the three inequalities above, and noticing $f(\boldsymbol{z}^T) \geq f^*$ from Assumption 1, we have

$$\frac{1}{T}\sum_{t=0}^{T-1}\mathbb{E}\left[\|\nabla f\left(\overline{\boldsymbol{x}}^t\right)\|^2\right] \leq \frac{1}{\theta\sqrt{nT}}\left(f(\boldsymbol{z}^0) - f^* + \frac{\alpha\beta_1(24-23\beta_1)}{24(1-\beta_1)^2}\frac{dB_\infty^2}{\sqrt{\delta}}\right) + \frac{96L\alpha\sqrt{B_\infty^2+\delta}}{n\delta}B^2$$

$$+ 4\sqrt{B_\infty^2+\delta}\left(\frac{3L^2\alpha}{4n\sqrt{\delta}} + \frac{L^2\alpha}{n}\frac{\sqrt{B_\infty^2+\delta}}{\delta}\right)\alpha C$$

$$+ \frac{1}{\theta\sqrt{nT}}\frac{13\alpha dB_\infty^4}{12\delta^{\frac{3}{2}}}\left(T - \frac{\beta_2(1-\beta_2^T)}{1-\beta_2}\right) + 4\sqrt{B_\infty^2+\delta}\left(\frac{\alpha^2\beta_1^2 L^2 B^2}{\delta(1-\beta_1)^2}\left(\frac{\sqrt{B_\infty^2+\delta}}{\delta} + \frac{1}{2\sqrt{\delta}}\right)\right).$$

Replacing $\alpha$ by $\frac{4\theta\sqrt{n}\sqrt{B_\infty^2+\delta}}{\sqrt{T}}$ in the above inequality and adding the resulting inequality to (15), we obtain

$$\frac{1}{T}\sum_{t=0}^{T-1}\mathbb{E}\left[\|\nabla f\left(\overline{\boldsymbol{x}}^t\right)\|^2 + \frac{1}{n}\|\mathbf{X}_\perp^t\|^2\right] \leq \frac{1}{\theta\sqrt{nT}}\left(f(\boldsymbol{z}^0) - f^* + \frac{\theta\sqrt{n}\sqrt{B_\infty^2+\delta}\beta_1(24-23\beta_1)}{6(1-\beta_1)^2}\frac{dB_\infty^2}{\sqrt{\delta T}}\right)$$

$$+ \frac{384\theta L(B_\infty^2+\delta)}{\delta\sqrt{nT}}B^2 + C\left(\frac{3}{4\sqrt{\delta}} + \frac{\sqrt{B_\infty^2+\delta}}{\delta}\right)\frac{64\theta^2 L^2(B_\infty^2+\delta)^{\frac{3}{2}}}{T} + \frac{16\theta^2(B_\infty^2+\delta)}{T}C$$

$$+ \frac{13dB_\infty^4}{3\delta^{\frac{3}{2}}}\frac{\sqrt{B_\infty^2+\delta}}{T}\left(T - \frac{\beta_2(1-\beta_2^T)}{1-\beta_2}\right) \tag{62}$$

$$+ \left(\frac{\beta_1^2 B^2}{\delta(1-\beta_1)^2}\left(\frac{\sqrt{B_\infty^2+\delta}}{\delta} + \frac{1}{2\sqrt{\delta}}\right)\right)\frac{64\theta^2 L^2 n(B_\infty^2+\delta)^{\frac{3}{2}}}{T}.$$

Case (i): When $\delta = \omega^2 B_\infty^2 \frac{\sqrt{T}}{\sqrt{n}}$, we have $\frac{B_\infty^2+\delta}{\delta} = 1 + \frac{\sqrt{n}}{\omega^2\sqrt{T}} = \Theta(1)$ because $n \leq T$. In addition, notice $\frac{\beta_2}{T(1-\beta_2)} \geq 1$ from $\beta_2 \geq \frac{T}{T+1}$ and thus

$$\frac{1}{T}\left(T - \frac{\beta_2(1-\beta_2^T)}{1-\beta_2}\right) = 1 - \frac{\beta_2(1-\beta_2^T)}{T(1-\beta_2)} \leq \beta_2^T \leq \frac{\theta L}{n}, \tag{63}$$

where the last inequality follows from $\beta_2 \leq (\frac{\theta L}{n})^{\frac{1}{T}}$. Therefore, by ignoring universal constants, we have from (15) and (62) that

$$\frac{1}{T}\sum_{t=0}^{T-1}\mathbb{E}\left[\|\nabla f\left(\overline{\boldsymbol{x}}^t\right)\|^2 + \frac{1}{n}\|\mathbf{X}_\perp^t\|^2\right] \tag{64}$$
$$=O\left(\frac{1}{\sqrt{nT}}\left(f(\boldsymbol{z}^0) - f^* + \frac{dB_\infty^2\sqrt{n}}{\sqrt{T}} + LB^2 + dB_\infty^2 L + C(1+L^2)B_\infty^2\right) + C(1+L^2)\frac{B_\infty^2}{T} + \frac{nL^2B^2}{T}\right).$$

Because $n \leq T$, we have (60) from (64).

Case (ii): Define $h(\beta_2) = \frac{1}{T}\left(T - \frac{\beta_2(1-\beta_2^T)}{1-\beta_2}\right)$. Notice that $h(\beta_2) = \frac{1}{T}\sum_{t=0}^{T-1}(1-\beta_2^{t+1})$ and thus $h$ is decreasing on $[0,1)$. Hence $h(\beta_2) \leq h(\frac{pT}{pT+1})$ for any $\beta_2 \in [\frac{pT}{pT+1}, 1)$. In addition, $(\frac{pT}{pT+1})^T = (1 - \frac{1}{pT+1})^{(pT+1)\cdot\frac{T}{pT+1}} \leq \frac{1}{e^{\frac{T}{pT+1}}}$ and $\frac{\beta_2}{T(1-\beta_2)} = p$ for $\beta_2 = \frac{pT}{pT+1}$. Thus it holds that for any $\beta_2 \in [\frac{pT}{pT+1}, 1)$,

$$h(\beta_2) \leq h\left(\frac{pT}{pT+1}\right) = 1 - p\left(1 - \left(\frac{pT}{pT+1}\right)^T\right) \leq 1 - p\left(1 - e^{-\frac{T}{pT+1}}\right). \tag{65}$$

By Taylor expansion of $e^{-\frac{T}{pT+1}}$ at the origin point, we have $e^{-\frac{T}{pT+1}} \leq 1 - \frac{T}{pT+1} + \frac{T^2}{2(pT+1)^2}$, which together with (65) gives

$$h(\beta_2) \leq \frac{1}{pT+1} + \frac{pT^2}{2(pT+1)^2} \leq \frac{1}{pT} + \frac{1}{2p}, \forall \beta_2 \in \left[\frac{pT}{pT+1}, 1\right). \tag{66}$$

With the choice of $T \geq n$ and $p = \sqrt{nT}$, we have $h(\beta_2) \leq \frac{1}{\sqrt{nT}}$. Hence, when $\delta > 0$ is independent of $T$ and $n$, we have from (62) that by ignoring universal constants,

$$\frac{1}{T}\sum_{t=0}^{T-1}\mathbb{E}\left[\|\nabla f\left(\overline{\boldsymbol{x}}^t\right)\|^2 + \frac{1}{n}\|\mathbf{X}_\perp^t\|^2\right]$$
$$=O\left(\frac{1}{\sqrt{nT}}\left(f(\boldsymbol{z}^0) - f^* + \frac{dB_\infty^3\sqrt{n}}{\sqrt{T}} + LB^2B_\infty^2 + dB_\infty^2 L + dB_\infty^5\right) + \frac{CB_\infty^2}{T}\left(1+L^2B_\infty^2\right) + \frac{L^2B^2B_\infty^4}{T}\right),$$

which indicates (61) by $n \leq T$. $\qquad\square$

**Remark 3 (Linear speed up and topology independence)** *In Case (i) of Theorem A.2, with the chosen $\alpha$ and $\delta$, if*

$$T = \Omega\left(\max\left\{\frac{n^3}{(1-\widehat{\rho}^2)^4}, \frac{n^{\frac{7}{3}}}{(1-\eta^2)^{\frac{8}{3}}}\right\}\right), \tag{67}$$

*we have $C = O(1)$ and $\frac{n}{T} = O(\frac{1}{\sqrt{nT}})$, and thus (60) implies*

$$\frac{1}{T}\sum_{t=0}^{T-1}\mathbb{E}\left[\|\nabla f\left(\overline{\boldsymbol{x}}^t\right)\|^2 + \frac{1}{n}\|\mathbf{X}_\perp^t\|^2\right] = O\left(\frac{1}{\sqrt{nT}}\right). \tag{68}$$

*For Case (ii) of Theorem A.2, with the chosen $\alpha$ and $\delta$, if*

$$T = \Omega\left(\max\left\{\frac{n^3}{(1-\widehat{\rho}^2)^4}, \frac{n^2}{(1-\eta^2)^2}\right\}\right), \tag{69}$$

*then $\frac{C}{T} = O(\frac{1}{\sqrt{nT}})$ and $\frac{1}{T} = O(\frac{1}{\sqrt{nT}})$. Thus (61) also implies (68). Notice that letting $\tau$ be selected from $\{0,\dots,T-1\}$ uniformly at random, we have from (68) that $\mathbb{E}\left[\nabla f(\overline{\boldsymbol{x}}^\tau) + \frac{1}{n}\|\mathbf{X}_\perp^\tau\|^2\right] = O\left(\frac{1}{\sqrt{nT}}\right)$. For a given $\varepsilon > 0$, we need $T = \Theta\left(\frac{1}{n\varepsilon^4}\right)$ iterations to produce an $\varepsilon$-stationary solution in expectation, i.e., $\mathbb{E}\left[\nabla f(\overline{\boldsymbol{x}}^\tau) + \frac{1}{n}\|\mathbf{X}_\perp^\tau\|^2\right] \leq \varepsilon^2$. If $\varepsilon$ is sufficiently small such that $T = \Theta\left(\frac{1}{n\varepsilon^4}\right)$ satisfies the conditions in (67) and (69), we obtain linear speed up with respect to $n$, and the parameters $\alpha$ and $\delta$ are independent of the communication graph.*

# B  Convergence Rate Results of Algorithm 2

In this section, we analyze the convergence rate of Algorithm 2.

## B.1  Noncompressed case of Algorithm 2

For ease of reading, we first consider the special noncompressed case, where $\gamma_x = \gamma_g = 1$ and $\mathcal{Q} = \mathbf{I}$. In this case, we can write the update in the more compact matrix form as follows

$$\mathbf{G}^{t-\frac{1}{2}} = \mathbf{G}^{t-1} + \nabla \mathbf{F}^t - \nabla \mathbf{F}^{t-1}, \tag{70}$$

$$\mathbf{G}^t = \mathbf{G}^{t-\frac{1}{2}} \mathbf{W}, \tag{71}$$

$$\mathbf{M}^t = \beta_1 \mathbf{M}^{t-1} + (1 - \beta_1) \mathbf{G}^t, \tag{72}$$

$$\mathbf{X}^{t+\frac{1}{2}} = \mathbf{X}^t - \alpha \mathbf{M}^t, \tag{73}$$

$$\mathbf{X}^{t+1} = \mathbf{X}^{t+\frac{1}{2}} \mathbf{W}, \tag{74}$$

where we use notations in Section 1.3 by letting

$$\mathbf{G}^t = \left[ \boldsymbol{g}_1^t, \boldsymbol{g}_2^t, \ldots, \boldsymbol{g}_n^t \right], \ \mathbf{G}^{t-\frac{1}{2}} = \left[ \boldsymbol{g}_1^{t-\frac{1}{2}}, \boldsymbol{g}_2^{t-\frac{1}{2}}, \ldots, \boldsymbol{g}_n^{t-\frac{1}{2}} \right].$$

Under Assumption 5, one can easily have

$$\mathbb{E}_t \left[ \frac{1}{n} \sum_{i=1}^n \boldsymbol{g}_i^t \right] = \frac{1}{n} \sum_{i=1}^n \nabla f_i(\boldsymbol{x}_i^t). \tag{75}$$

In addition, it is straightforward to have

$$\frac{1}{n} \sum_{i=1}^n \boldsymbol{g}_i^t = \frac{1}{n} \sum_{i=1}^n \nabla \mathbf{F}_i(\boldsymbol{x}_i^t, \xi_i^t). \tag{76}$$

With these, we first show the following lemma.

**Lemma B.1** *Under Assumptions 1 and 5, it holds that*

$$\mathbb{E}\left[ \|\bar{\boldsymbol{g}}^t\|^2 \right] = \mathbb{E}\left[ \left\| \frac{1}{n} \sum_{i=1}^n \boldsymbol{g}_i^t \right\|^2 \right] \le \frac{\sigma^2}{n} + 2\mathbb{E}\left[ \left\| \nabla f(\bar{\boldsymbol{x}}^t) \right\|^2 \right] + \frac{2L^2}{n} \mathbb{E}\left[ \|\mathbf{X}_\perp^t\|^2 \right]. \tag{77}$$

*Proof.* We have

$$\mathbb{E}\left[ \left\| \frac{1}{n} \sum_{i=1}^n \boldsymbol{g}_i^t \right\|^2 \right] = \mathbb{E}\left[ \left\| \frac{1}{n} \sum_{i=1}^n \boldsymbol{g}_i^t - \frac{1}{n} \sum_{i=1}^n \nabla f_i(\boldsymbol{x}_i^t) + \frac{1}{n} \sum_{i=1}^n \nabla f_i(\boldsymbol{x}_i^t) \right\|^2 \right]$$

$$\overset{(75)}{=} \mathbb{E}\left[ \left\| \frac{1}{n} \sum_{i=1}^n \boldsymbol{g}_i^t - \frac{1}{n} \sum_{i=1}^n \nabla f_i(\boldsymbol{x}_i^t) \right\|^2 \right] + \mathbb{E}\left[ \left\| \frac{1}{n} \sum_{i=1}^n \nabla f_i(\boldsymbol{x}_i^t) \right\|^2 \right]$$

$$\overset{(76)}{=} \mathbb{E}\left[ \left\| \frac{1}{n} \sum_{i=1}^n \nabla \mathbf{F}_i(\boldsymbol{x}_i^t, \xi_i^t) - \frac{1}{n} \sum_{i=1}^n \nabla f_i(\boldsymbol{x}_i^t) \right\|^2 \right] + \mathbb{E}\left[ \left\| \frac{1}{n} \sum_{i=1}^n \nabla f_i(\boldsymbol{x}_i^t) \right\|^2 \right]$$

$$\le \mathbb{E}\left[ \left\| \frac{1}{n} \sum_{i=1}^n \nabla \mathbf{F}_i(\boldsymbol{x}_i^t, \xi_i^t) - \frac{1}{n} \sum_{i=1}^n \nabla f_i(\boldsymbol{x}_i^t) \right\|^2 \right] + 2\mathbb{E}\left[ \left\| \nabla f(\bar{\boldsymbol{x}}^t) \right\|^2 \right] + 2\mathbb{E}\left[ \left\| \frac{1}{n} \sum_{i=1}^n \nabla f_i(\boldsymbol{x}_i^t) - \nabla f(\bar{\boldsymbol{x}}^t) \right\|^2 \right]$$

$$\le \frac{\sigma^2}{n} + 2\mathbb{E}\left[ \left\| \nabla f(\bar{\boldsymbol{x}}^t) \right\|^2 \right] + \frac{2L^2}{n} \mathbb{E}\left[ \|\mathbf{X}_\perp^t\|^2 \right],$$

where the first inequality follows from Young's inequality, and the second one holds from the independence of $\{\xi_i^t\}$, Assumption 5, and (34). The proof is then completed. $\qquad \square$

**Lemma B.2** *Under Assumptions 1 and 5, let $\alpha \leq \frac{1}{4L}$. Then*

$$
\begin{aligned}
\mathbb{E}\left[\left\|\nabla f\left(\overline{\boldsymbol{x}}^t\right)\right\|^2\right] \leq & \frac{4}{\alpha}\left(\mathbb{E}\left[f\left(\boldsymbol{z}^t\right)\right]-\mathbb{E}\left[f\left(\boldsymbol{z}^{t+1}\right)\right]\right)+\frac{2\alpha L\sigma^2}{n} \\
& +\left(\frac{4\alpha L^3}{n}+\frac{6L^2}{n}\right)\mathbb{E}\left[\|\mathbf{X}_\perp^t\|^2\right]+\frac{6L^2\alpha^2\beta_1^2}{(1-\beta_1)^2}\mathbb{E}\left[\|\overline{\boldsymbol{m}}^{t-1}\|^2\right].
\end{aligned}
\tag{78}
$$

*Proof.* By (55) with $\boldsymbol{u}_i^t = \boldsymbol{0}$ and $\delta = 1$, we have

$$
\mathbb{E}\left[\left\langle\nabla f\left(\boldsymbol{z}^t\right),\frac{1}{n}\sum_{i=1}^n\boldsymbol{g}_i^t\right\rangle\right] \leq \frac{1}{\alpha}\mathbb{E}\left[f\left(\boldsymbol{z}^t\right)\right]-\mathbb{E}\left[f\left(\boldsymbol{z}^{t+1}\right)\right]+\frac{L}{2\alpha}\mathbb{E}\left[\left\|\boldsymbol{z}^{t+1}-\boldsymbol{z}^t\right\|^2\right].
\tag{79}
$$

In addition, it holds

$$
\begin{aligned}
\mathbb{E}_t&\left[\left\langle\nabla f\left(\boldsymbol{z}^t\right),\frac{1}{n}\sum_{i=1}^n\boldsymbol{g}_i^t\right\rangle\right] \overset{(75)}{=} \left\langle\nabla f\left(\boldsymbol{z}^t\right),\frac{1}{n}\sum_{i=1}^n\nabla f_i\left(\boldsymbol{x}_i^t\right)\right\rangle \\
&=\left\langle\nabla f\left(\boldsymbol{z}^t\right)-\nabla f\left(\overline{\boldsymbol{x}}^t\right),\frac{1}{n}\sum_{i=1}^n\nabla f_i\left(\boldsymbol{x}_i^t\right)-\nabla f\left(\overline{\boldsymbol{x}}^t\right)\right\rangle+\left\langle\nabla f\left(\boldsymbol{z}^t\right)-\nabla f\left(\overline{\boldsymbol{x}}^t\right),\nabla f\left(\overline{\boldsymbol{x}}^t\right)\right\rangle \\
&\quad+\left\langle\nabla f\left(\overline{\boldsymbol{x}}^t\right),\frac{1}{n}\sum_{i=1}^n\nabla f_i\left(\boldsymbol{x}_i^t\right)-\nabla f\left(\overline{\boldsymbol{x}}^t\right)\right\rangle+\left\|\nabla f\left(\overline{\boldsymbol{x}}^t\right)\right\|^2 \\
&\overset{(34)}{\geq} -\frac{L^2}{2}\left\|\boldsymbol{z}^t-\overline{\boldsymbol{x}}^t\right\|^2-\frac{L^2}{2n}\left\|\mathbf{X}_\perp^t\right\|^2-L^2\left\|\boldsymbol{z}^t-\overline{\boldsymbol{x}}^t\right\|^2-\frac{1}{4}\left\|\nabla f\left(\overline{\boldsymbol{x}}^t\right)\right\|^2 \\
&\quad-\frac{1}{4}\left\|\nabla f\left(\overline{\boldsymbol{x}}^t\right)\right\|^2-\frac{L^2}{n}\left\|\mathbf{X}_\perp^t\right\|^2+\left\|\nabla f\left(\overline{\boldsymbol{x}}^t\right)\right\|^2 \\
&=-\frac{3L^2}{2}\left\|\boldsymbol{z}^t-\overline{\boldsymbol{x}}^t\right\|^2-\frac{3L^2}{2n}\left\|\mathbf{X}_\perp^t\right\|^2+\frac{1}{2}\left\|\nabla f\left(\overline{\boldsymbol{x}}^t\right)\right\|^2 \\
&=-\frac{3L^2\alpha^2\beta_1^2}{2(1-\beta_1)^2}\left\|\overline{\boldsymbol{m}}^{t-1}\right\|^2-\frac{3L^2}{2n}\left\|\mathbf{X}_\perp^t\right\|^2+\frac{1}{2}\left\|\nabla f\left(\overline{\boldsymbol{x}}^t\right)\right\|^2,
\end{aligned}
\tag{80}
$$

where we have used (31) and (33) with $\boldsymbol{u}_i^t = \boldsymbol{0}$ and $\delta = 1$ in the last equality.

Moreover, by (32) with $\boldsymbol{u}_i^t = \boldsymbol{0}$ and $\delta = 1$, we have $\boldsymbol{z}^{t+1}-\boldsymbol{z}^t=-\frac{\alpha}{n}\sum_{i=1}^n\boldsymbol{g}_i^t$. Thus

$$
\frac{L}{2\alpha}\mathbb{E}\left[\left\|\boldsymbol{z}^{t+1}-\boldsymbol{z}^t\right\|^2\right]=\frac{\alpha L}{2}\mathbb{E}\left[\left\|\frac{1}{n}\sum_{i=1}^n\boldsymbol{g}_i^t\right\|^2\right]\overset{(77)}{\leq}\frac{\alpha L\sigma^2}{2n}+\alpha L\mathbb{E}\left[\|\nabla f(\overline{\boldsymbol{x}}^t)\|^2\right]+\frac{\alpha L^3}{n}\mathbb{E}\left[\|\mathbf{X}_\perp^t\|^2\right].
\tag{81}
$$

Substituting (80) and (81) into (79) yields

$$
\begin{aligned}
\left(\frac{1}{2}-\alpha L\right)\mathbb{E}\left[\left\|\nabla f\left(\overline{\boldsymbol{x}}^t\right)\right\|^2\right] \leq & \frac{1}{\alpha}\left(\mathbb{E}\left[f\left(\boldsymbol{z}^t\right)\right]-\mathbb{E}\left[f\left(\boldsymbol{z}^{t+1}\right)\right]\right)+\frac{\alpha L\sigma^2}{2n} \\
& +\left(\frac{\alpha L^3}{n}+\frac{3L^2}{2n}\right)\mathbb{E}\left[\|\mathbf{X}_\perp^t\|^2\right]+\frac{3L^2\alpha^2\beta_1^2}{2(1-\beta_1)^2}\mathbb{E}\left[\|\overline{\boldsymbol{m}}^{t-1}\|^2\right].
\end{aligned}
\tag{82}
$$

Now notice $\frac{1}{2}-\alpha L \geq \frac{1}{4}$ from the condition on $\alpha$, and we complete the proof by (82). $\qquad\square$

**Lemma B.3** *Under Assumptions 1, 2, 3, and 5, let $\alpha \leq \frac{1}{4L}$. Then for all $t \geq 0$,*

$$
\mathbb{E}\left[\left\|\overline{\boldsymbol{m}}^t\right\|^2\right] \leq \beta_1\mathbb{E}\left[\left\|\overline{\boldsymbol{m}}^{t-1}\right\|^2\right]+(1-\beta_1)\left(\frac{\sigma^2}{n}+2\mathbb{E}\left[\|\nabla f(\overline{\boldsymbol{x}}^t)\|^2\right]+\frac{2L^2}{n}\mathbb{E}\left[\|\mathbf{X}_\perp^t\|^2\right]\right).
\tag{83}
$$

*Proof.* By (72), it holds

$$
\mathbb{E}\left[\left\|\overline{\boldsymbol{m}}^t\right\|^2\right] = \mathbb{E}\left[\left\|\beta_1\overline{\boldsymbol{m}}^{t-1} + (1-\beta_1)\overline{\boldsymbol{g}}^t\right\|^2\right]
$$

$$
\leq \beta_1\mathbb{E}\left[\left\|\overline{\boldsymbol{m}}^{t-1}\right\|^2\right] + (1-\beta_1)\mathbb{E}\left[\left\|\overline{\boldsymbol{g}}^t\right\|^2\right] \tag{84}
$$

$$
\stackrel{(77)}{\leq} \beta_1\mathbb{E}\left[\left\|\overline{\boldsymbol{m}}^{t-1}\right\|^2\right] + (1-\beta_1)\left(\frac{\sigma^2}{n} + 2\mathbb{E}\left[\left\|\nabla f(\overline{\boldsymbol{x}}^t)\right\|^2\right] + \frac{2L^2}{n}\mathbb{E}\left[\|\mathbf{X}_\perp^t\|^2\right]\right),
$$

which completes the proof. $\qquad\square$

**Lemma B.4** *Under Assumptions 1, 2, 3, and 5, it holds that for all $t \geq 0$,*

$$
\mathbb{E}\left[\left\|\mathbf{G}_\perp^t\right\|^2\right] \leq \frac{1+\rho^2}{2}\mathbb{E}\left[\left\|\mathbf{G}_\perp^{t-1}\right\|^2\right] + \frac{\rho^2(1+\rho^2)}{1-\rho^2}\Bigg(2n\sigma^2 \tag{85}
$$

$$
+ L^2\left(8\mathbb{E}\left[\left\|\mathbf{X}_\perp^{t-1}\right\|^2\right] + 4\alpha^2\rho^2\mathbb{E}\left[\left\|\mathbf{M}_\perp^{t-1}\right\|^2\right] + 4\alpha^2 n\mathbb{E}\left[\left\|\overline{\boldsymbol{m}}^{t-1}\right\|^2\right]\right)\Bigg),
$$

$$
\mathbb{E}\left[\left\|\mathbf{M}_\perp^{t+1}\right\|^2\right] \leq \beta_1\mathbb{E}\left[\left\|\mathbf{M}_\perp^t\right\|^2\right] + (1-\beta_1)\mathbb{E}\left[\left\|\mathbf{G}_\perp^{t+1}\right\|^2\right]. \tag{86}
$$

*Proof.* By (70) and (71), we have

$$
\left\|\mathbf{G}_\perp^t\right\|^2 = \left\|\mathbf{G}^{t-\frac{1}{2}}(\mathbf{W}-\mathbf{J})\right\|^2 = \left\|\mathbf{G}^{t-1}(\mathbf{W}-\mathbf{J}) + \left(\nabla\mathbf{F}^t - \nabla\mathbf{F}^{t-1}\right)(\mathbf{W}-\mathbf{J})\right\|^2
$$

$$
= \left\|\mathbf{G}^{t-1}(\mathbf{I}-\mathbf{J})(\mathbf{W}-\mathbf{J})\right\|^2 + \left\|\left(\nabla\mathbf{F}^t - \nabla\mathbf{F}^{t-1}\right)(\mathbf{W}-\mathbf{J})\right\|^2
$$

$$
+ 2\left\langle\mathbf{G}^{t-1}(\mathbf{W}-\mathbf{J}), \left(\nabla\mathbf{F}^t - \nabla\mathbf{F}^{t-1}\right)(\mathbf{W}-\mathbf{J})\right\rangle
$$

$$
\leq \rho^2\left\|\mathbf{G}_\perp^{t-1}\right\|^2 + \rho^2\left\|\nabla\mathbf{F}^t - \nabla\mathbf{F}^{t-1}\right\|^2 + 2\left\langle\mathbf{G}^{t-1}(\mathbf{W}-\mathbf{J}), \left(\nabla\mathbf{F}^t - \nabla\mathbf{F}^{t-1}\right)(\mathbf{W}-\mathbf{J})\right\rangle, \tag{87}
$$

where we have used $\mathbf{J}\mathbf{W} = \mathbf{J}\mathbf{J} = \mathbf{J}$ and $\|\mathbf{W}-\mathbf{J}\|_2 \leq \rho$. For the third term on the RHS of (87), we have

$$
2\left\langle\mathbf{G}^{t-1}(\mathbf{W}-\mathbf{J}), \left(\nabla\mathbf{F}^t - \nabla\mathbf{F}^{t-1}\right)(\mathbf{W}-\mathbf{J})\right\rangle
$$

$$
\leq 2\left\|\mathbf{G}^{t-1}(\mathbf{W}-\mathbf{J})\right\| \cdot \left\|\left(\nabla\mathbf{F}^t - \nabla\mathbf{F}^{t-1}\right)(\mathbf{W}-\mathbf{J})\right\|
$$

$$
\leq 2\rho^2\left\|\mathbf{G}_\perp^{t-1}\right\| \cdot \left\|\nabla\mathbf{F}^t - \nabla\mathbf{F}^{t-1}\right\| \leq \frac{1-\rho^2}{2}\left\|\mathbf{G}_\perp^{t-1}\right\|^2 + \frac{2\rho^4}{1-\rho^2}\left\|\nabla\mathbf{F}^t - \nabla\mathbf{F}^{t-1}\right\|^2, \tag{88}
$$

where the second inequality holds because $\|\mathbf{W}-\mathbf{J}\|_2 \leq \rho$ and $\mathbf{W}-\mathbf{J} = (\mathbf{I}-\mathbf{J})(\mathbf{W}-\mathbf{J})$, and the third one follows from Young's inequality. Plugging (88) into (87) gives

$$
\left\|\mathbf{G}_\perp^t\right\|^2 \leq \frac{1+\rho^2}{2}\left\|\mathbf{G}_\perp^{t-1}\right\|^2 + \frac{\rho^2(1+\rho^2)}{1-\rho^2}\left\|\nabla\mathbf{F}^t - \nabla\mathbf{F}^{t-1}\right\|^2. \tag{89}
$$

In addition, we have

$$
\mathbb{E}\left\|\nabla\mathbf{F}^t - \nabla\mathbf{F}^{t-1}\right\|^2
$$

$$
= \mathbb{E}\left\|\nabla\mathbf{F}^t - \nabla\mathbf{F}^{t-1} - \nabla\boldsymbol{f}^t + \nabla\boldsymbol{f}^{t-1} + \nabla\boldsymbol{f}^t - \nabla\boldsymbol{f}^{t-1}\right\|^2
$$

$$
= \mathbb{E}\left\|\nabla\mathbf{F}^t - \nabla\mathbf{F}^{t-1} - \nabla\boldsymbol{f}^t + \nabla\boldsymbol{f}^{t-1}\right\|^2 + \mathbb{E}\left\|\nabla\boldsymbol{f}^t - \nabla\boldsymbol{f}^{t-1}\right\|^2
$$

$$
\leq 2n\sigma^2 + L^2\mathbb{E}\left[\|\mathbf{X}^t - \mathbf{X}^{t-1}\|^2\right]. \tag{90}
$$

Moreover, by (73) and (74), it holds

$$
\begin{aligned}
\left\|\mathbf{X}^t - \mathbf{X}^{t-1}\right\|^2 &= \left\|\left(\mathbf{X}^{t-1} - \alpha\mathbf{M}^{t-1}\right)\mathbf{W} - \mathbf{X}^{t-1}\right\|^2 \\
&\leq 2\left\|\mathbf{X}^{t-1}(\mathbf{W} - \mathbf{I})\right\|^2 + 2\alpha^2\left\|\mathbf{M}^{t-1}\mathbf{W}\right\|^2 \\
&= 2\left\|\mathbf{X}^{t-1}(\mathbf{W} - \mathbf{I}) - \overline{\mathbf{X}}^{t-1}(\mathbf{W} - \mathbf{I})\right\|^2 + 2\alpha^2\left\|\mathbf{M}^{t-1}\mathbf{W} - \overline{\mathbf{M}}^{t-1} + \overline{\mathbf{M}}^{t-1}\right\|^2 \\
&\leq 8\left\|\mathbf{X}_\perp^{t-1}\right\|^2 + 4\alpha^2\left\|\mathbf{M}^{t-1}(\mathbf{W} - \mathbf{J})\right\|^2 + 4\alpha^2\left\|\overline{\mathbf{M}}^{t-1}\right\|^2 \\
&= 8\left\|\mathbf{X}_\perp^{t-1}\right\|^2 + 4\alpha^2\left\|\mathbf{M}^{t-1}(\mathbf{I} - \mathbf{J})(\mathbf{W} - \mathbf{J})\right\|^2 + 4\alpha^2 n\left\|\overline{\boldsymbol{m}}^{t-1}\right\|^2 \\
&\leq 8\left\|\mathbf{X}_\perp^{t-1}\right\|^2 + 4\alpha^2\rho^2\left\|\mathbf{M}_\perp^{t-1}\right\|^2 + 4\alpha^2 n\left\|\overline{\boldsymbol{m}}^{t-1}\right\|^2.
\end{aligned} \tag{91}
$$

Now substituting (90) and (91) into (89) and taking full expectation yields (85).

Finally by (72) and the convexity of $\|\cdot\|^2$, we obtain (86) and complete the proof. $\qquad\square$

We are now ready to show the convergence rate of the noncompressed case of Algorithm 2, by combining Lemmas B.2–B.4 and the following inequality that is obtained from (17) with $\mathbf{Y}^t = \mathbf{M}^t$

$$
\left\|\mathbf{X}_\perp^{t+1}\right\|^2 \leq \frac{1 + \rho^2}{2}\left\|\mathbf{X}_\perp^t\right\|^2 + \frac{2\rho^2\alpha^2}{1 - \rho^2}\left\|\mathbf{M}_\perp^t\right\|^2. \tag{92}
$$

**Theorem B.1** *Under Assumptions 1, 2, 3, and 5, let $\{\mathbf{X}^t\}$ be generated from Algorithm 2 with $\gamma_x = \gamma_g = 1$, $\mathcal{Q} = \mathbf{I}$, and $0 < \alpha \leq \frac{1}{4L}$ satisfying*

$$
\frac{\rho^2(1 + \rho^2)}{1 - \rho^2}\left(\frac{32L^2\rho^2\alpha^2}{(1 - \rho^2)^2} + 4\alpha^2\rho^2 L^2 + 8\alpha^2 L^4\frac{4\rho^2\alpha^2}{(1 - \rho^2)^2}\right) \leq \frac{1 + \rho^2}{4}, \tag{93}
$$

$$
\frac{12L^2\alpha^2\beta_1^2}{(1 - \beta_1)^2} + 8\alpha^2 nL^2 C_g \leq \frac{1}{2}, \tag{94}
$$

*where $C_g$ is defined as*

$$
C_g = \frac{4\rho^2(1 + \rho^2)}{(1 - \rho^2)^2}\frac{4\rho^2\alpha^2}{(1 - \rho^2)^2}\left(\frac{4\alpha L^3}{n} + \frac{6L^2}{n} + \frac{12L^4\alpha^2\beta_1^2}{n(1 - \beta_1)^2}\right). \tag{95}
$$

*Then it holds*

$$
\frac{1}{T}\sum_{t=0}^{T-1}\mathbb{E}\left[\left\|\nabla f\left(\overline{\boldsymbol{x}}^t\right)\right\|^2\right] \leq \frac{8}{\alpha T}\left(f\left(\boldsymbol{z}^0\right) - f^*\right) + \frac{4\alpha L\sigma^2}{n} + 2C_g(2n\sigma^2 + 4\alpha^2 L^2\sigma^2), \tag{96}
$$

$$
\frac{1}{T}\sum_{t=0}^{T-1}\left\|\mathbf{X}_\perp^t\right\|^2 \leq \frac{4\rho^2\alpha^2}{(1 - \rho^2)^2}\frac{4\rho^2(1 + \rho^2)}{(1 - \rho^2)^2}\left(2n\sigma^2 + 4\alpha^2 L^2\sigma^2 + 8\alpha^2 nL^2\frac{1}{T}\sum_{t=0}^{T-1}\mathbb{E}\left[\left\|\nabla f(\overline{\boldsymbol{x}}^t)\right\|^2\right]\right). \tag{97}
$$

*Proof.* Sum up (92) over $t = 0$ To $T - 2$. We have

$$
\sum_{t=1}^{T-1}\left\|\mathbf{X}_\perp^t\right\|^2 \leq \frac{1 + \rho^2}{2}\sum_{t=0}^{T-2}\left\|\mathbf{X}_\perp^t\right\|^2 + \frac{2\rho^2\alpha^2}{1 - \rho^2}\sum_{t=0}^{T-2}\left\|\mathbf{M}_\perp^t\right\|^2,
$$

which together with $\mathbf{X}_\perp^0 = \mathbf{0}$ gives

$$
\sum_{t=0}^{T-1}\left\|\mathbf{X}_\perp^t\right\|^2 \leq \frac{4\rho^2\alpha^2}{(1 - \rho^2)^2}\sum_{t=0}^{T-2}\left\|\mathbf{M}_\perp^t\right\|^2. \tag{98}
$$

Similarly, summing up (86) over $t = -1$ to $T'$, noticing $\mathbf{M}_{\perp}^{-1} = \mathbf{0}$, and rearranging terms, we have

$$\sum_{t=0}^{T'-1} \mathbb{E}\left[\left\|\mathbf{M}_{\perp}^t\right\|^2\right] \leq \sum_{t=0}^{T'-1} \mathbb{E}\left[\left\|\mathbf{G}_{\perp}^t\right\|^2\right], \forall T' \geq 1. \tag{99}$$

Plugging (99) with $T' = T - 1$ into (98) yields

$$\sum_{t=0}^{T-1} \mathbb{E}\left[\left\|\mathbf{X}_{\perp}^t\right\|^2\right] \leq \frac{4\rho^2\alpha^2}{(1-\rho^2)^2} \sum_{t=0}^{T-2} \mathbb{E}\left[\left\|\mathbf{G}_{\perp}^t\right\|^2\right]. \tag{100}$$

In addition, summing up (83) over $t = 0$ to $T - 1$ gives

$$\begin{aligned}
\sum_{t=0}^{T-1} \mathbb{E}\left[\left\|\overline{\boldsymbol{m}}^t\right\|^2\right] \leq & \beta_1 \sum_{t=0}^{T-1} \mathbb{E}\left[\left\|\overline{\boldsymbol{m}}^{t-1}\right\|^2\right] + (1-\beta_1)\sum_{t=0}^{T-1}\left(\frac{\sigma^2}{n} + 2\mathbb{E}\left[\left\|\nabla f(\overline{\boldsymbol{x}}^t)\right\|^2\right] + \frac{2L^2}{n}\mathbb{E}\left[\left\|\mathbf{X}_{\perp}^t\right\|^2\right]\right) \\
& \stackrel{(100)}{\leq} \beta_1 \sum_{t=0}^{T-1} \mathbb{E}\left[\left\|\overline{\boldsymbol{m}}^{t-1}\right\|^2\right] + \frac{(1-\beta_1)T\sigma^2}{n} + 2\left(1-\beta_1\right)\sum_{t=0}^{T-1} \mathbb{E}\left[\left\|\nabla f(\overline{\boldsymbol{x}}^t)\right\|^2\right] \\
& + \frac{2(1-\beta_1)L^2}{n}\frac{4\rho^2\alpha^2}{(1-\rho^2)^2} \sum_{t=0}^{T-2} \mathbb{E}\left[\left\|\mathbf{G}_{\perp}^t\right\|^2\right].
\end{aligned} \tag{101}$$

Since $\overline{\boldsymbol{m}}^{-1} = \mathbf{0}$, we have from (101) that

$$\sum_{t=0}^{T-1} \mathbb{E}\left[\left\|\overline{\boldsymbol{m}}^t\right\|^2\right] \leq \frac{T\sigma^2}{n} + 2\sum_{t=0}^{T-1} \mathbb{E}\left[\left\|\nabla f(\overline{\boldsymbol{x}}^t)\right\|^2\right] + \frac{2L^2}{n}\frac{4\rho^2\alpha^2}{(1-\rho^2)^2} \sum_{t=0}^{T-2} \mathbb{E}\left[\left\|\mathbf{G}_{\perp}^t\right\|^2\right]. \tag{102}$$

Now sum up (85) over $t = 0$ to $T - 1$ and plug (99) with $T' = T - 1$, (100) and (102) into the resulting inequality. We have

$$\begin{aligned}
\sum_{t=0}^{T-1} \mathbb{E}\left[\left\|\mathbf{G}_{\perp}^t\right\|^2\right] \leq & \frac{1+\rho^2}{2}\mathbb{E}\sum_{t=0}^{T-1}\left[\left\|\mathbf{G}_{\perp}^{t-1}\right\|^2\right] + \frac{\rho^2(1+\rho^2)}{1-\rho^2}\sum_{t=0}^{T-1}\left(2n\sigma^2\right. \\
& \left. + L^2\left(8\mathbb{E}\left[\left\|\mathbf{X}_{\perp}^{t-1}\right\|^2\right] + 4\alpha^2\rho^2\mathbb{E}\left[\left\|\mathbf{M}_{\perp}^{t-1}\right\|^2\right] + 4\alpha^2n\mathbb{E}\left[\left\|\overline{\boldsymbol{m}}^{t-1}\right\|^2\right]\right)\right) \\
\leq & \frac{1+\rho^2}{2}\mathbb{E}\sum_{t=0}^{T-1}\left[\left\|\mathbf{G}_{\perp}^{t-1}\right\|^2\right] + \frac{\rho^2(1+\rho^2)}{1-\rho^2}\left(2n\sigma^2 T + \left(\frac{32L^2\rho^2\alpha^2}{(1-\rho^2)^2} + 4\alpha^2\rho^2L^2\right)\sum_{t=0}^{T-2}\mathbb{E}\left[\left\|\mathbf{G}_{\perp}^t\right\|^2\right]\right) \\
& + 4\alpha^2nL^2\frac{\rho^2(1+\rho^2)}{1-\rho^2}\left(\frac{T\sigma^2}{n} + 2\sum_{t=0}^{T-1}\mathbb{E}\left[\left\|\nabla f(\overline{\boldsymbol{x}}^t)\right\|^2\right] + \frac{2L^2}{n}\frac{4\rho^2\alpha^2}{(1-\rho^2)^2}\sum_{t=0}^{T-2}\mathbb{E}\left[\left\|\mathbf{G}_{\perp}^t\right\|^2\right]\right).
\end{aligned} \tag{103}$$

Define

$$\lambda = \frac{1+\rho^2}{2} + \frac{\rho^2(1+\rho^2)}{1-\rho^2}\left(\frac{32L^2\rho^2\alpha^2}{(1-\rho^2)^2} + 4\alpha^2\rho^2L^2 + 8\alpha^2L^4\frac{4\rho^2\alpha^2}{(1-\rho^2)^2}\right).$$

By the condition in (93), it holds $\lambda \leq \frac{3(1+\rho^2)}{4}$. Then $1 - \lambda \geq \frac{1-\rho^2}{4}$ and (103) indicates

$$\sum_{t=0}^{T-1} \mathbb{E}\left[\left\|\mathbf{G}_{\perp}^t\right\|^2\right] \leq \frac{4\rho^2(1+\rho^2)}{(1-\rho^2)^2}\left(2n\sigma^2 T + 4\alpha^2L^2T\sigma^2 + 8\alpha^2nL^2\sum_{t=0}^{T-1}\mathbb{E}\left[\left\|\nabla f(\overline{\boldsymbol{x}}^t)\right\|^2\right]\right). \tag{104}$$

Finally, sum up (78) over $t = 0$ to $T - 1$ and recall $f(\boldsymbol{z}) \geq f^*, \forall \boldsymbol{z}$ to have

$$
\begin{aligned}
\sum_{t=0}^{T-1} \mathbb{E}\left[\left\|\nabla f\left(\overline{\boldsymbol{x}}^t\right)\right\|^2\right] \leq & \frac{4}{\alpha}\left(f\left(\boldsymbol{z}^0\right) - f^*\right) + \frac{2\alpha L\sigma^2 T}{n} \\
& + \left(\frac{4\alpha L^3}{n} + \frac{6L^2}{n}\right) \sum_{t=0}^{T-1} \mathbb{E}\left[\|\mathbf{X}_\perp^t\|^2\right] + \frac{6L^2\alpha^2\beta_1^2}{(1-\beta_1)^2} \sum_{t=0}^{T-1} \mathbb{E}\left[\left\|\overline{\boldsymbol{m}}^{t-1}\right\|^2\right] \\
\leq & \frac{4}{\alpha}\left(f\left(\boldsymbol{z}^0\right) - f^*\right) + \frac{2\alpha L\sigma^2 T}{n} + \left(\frac{4\alpha L^3}{n} + \frac{6L^2}{n}\right) \frac{4\rho^2\alpha^2}{(1-\rho^2)^2} \sum_{t=0}^{T-2} \mathbb{E}\left[\left\|\mathbf{G}_\perp^t\right\|^2\right] \\
& + \frac{6L^2\alpha^2\beta_1^2}{(1-\beta_1)^2}\left(\frac{T\sigma^2}{n} + 2\sum_{t=0}^{T-1} \mathbb{E}\left[\left\|\nabla f(\overline{\boldsymbol{x}}^t)\right\|^2\right] + \frac{2L^2}{n}\frac{4\rho^2\alpha^2}{(1-\rho^2)^2}\sum_{t=0}^{T-2} \mathbb{E}\left[\left\|\mathbf{G}_\perp^t\right\|^2\right]\right) \\
\leq & \frac{4}{\alpha}\left(f\left(\boldsymbol{z}^0\right) - f^*\right) + \frac{2\alpha L\sigma^2 T}{n} + \frac{12L^2\alpha^2\beta_1^2}{(1-\beta_1)^2}\sum_{t=0}^{T-1} \mathbb{E}\left[\left\|\nabla f(\overline{\boldsymbol{x}}^t)\right\|^2\right] \quad (105) \\
& + C_g\left(2n\sigma^2 T + 4\alpha^2 L^2 T\sigma^2 + 8\alpha^2 nL^2\sum_{t=0}^{T-1} \mathbb{E}\left[\left\|\nabla f(\overline{\boldsymbol{x}}^t)\right\|^2\right]\right), \quad (106)
\end{aligned}
$$

where the second inequality follows from (100) and (102), the third one is obtained by plugging (104), and $C_g$ is defined in (95). By the condition in (94), we obtain (96) from (105). Plugging (104) into (100) gives (97) and completes the proof. $\qquad \square$

Below we specify the choice of $\alpha$ such that the conditions in (93) and (94) hold and simplify the convergence rate result in Theorem B.1 for Algorithm 2 with $\gamma_x = \gamma_g = 1$, and $\mathcal{Q} = \mathbf{I}$.

**Theorem B.2** *Suppose that the conditions assumed in Theorem B.1 hold. Let $T$ be large enough such that $\alpha = \frac{\theta\sqrt{n}}{\sigma\sqrt{T}}$ for some universal constant $\theta \in (0, 1)$ satisfies*

$$
\alpha \leq \min\left\{\frac{1-\rho^2}{8L}, \frac{(1-\rho^2)^{\frac{3}{2}}}{\rho L\sqrt{160}}, \frac{1-\beta_1}{\beta_1 L\sqrt{48}}, \frac{(1-\rho^2)^2}{16L\sigma^{\frac{1}{2}}(nT)^{\frac{1}{4}}}\right\}. \quad (107)
$$

*Then it holds*

$$
\frac{1}{T}\sum_{t=0}^{T-1} \mathbb{E}\left[\|\nabla f\left(\overline{\boldsymbol{x}}^t\right)\|^2 + \frac{1}{n}\|\mathbf{X}_\perp^t\|^2\right] = O\left(\frac{\sigma}{\sqrt{nT}}\right). \quad (108)
$$

*Proof.* By $\alpha \leq \frac{1-\rho^2}{\sqrt{8}L}$, it holds $8\alpha^2 L^4\frac{4\rho^2\alpha^2}{(1-\rho^2)^2} \leq 4\alpha^2\rho^2 L^2$. Hence,

$$
\frac{32L^2\rho^2\alpha^2}{(1-\rho^2)^2} + 4\alpha^2\rho^2 L^2 + 8\alpha^2 L^4\frac{4\rho^2\alpha^2}{(1-\rho^2)^2} \leq \frac{32L^2\rho^2\alpha^2}{(1-\rho^2)^2} + 8\alpha^2\rho^2 L^2 \leq \frac{40L^2\rho^2\alpha^2}{(1-\rho^2)^2} \leq \frac{1-\rho^2}{4},
$$

where the last inequality follows from $\alpha \leq \frac{(1-\rho^2)^{\frac{3}{2}}}{\rho L\sqrt{160}}$. Thus the condition in (93) holds by $\rho \leq 1$.

In addition, it follows from $\alpha \leq \frac{1-\beta_1}{\beta_1 L\sqrt{48}}$ that $\frac{12L^2\alpha^2\beta_1^2}{(1-\beta_1)^2} \leq \frac{1}{4}$. Moreover, by $\alpha \leq \frac{1}{4L}$ and $\alpha \leq \frac{1-\beta_1}{\beta_1 L\sqrt{48}}$, it holds $\frac{4\alpha L^3}{n} + \frac{6L^2}{n} + \frac{12L^4\alpha^2\beta_1^2}{n(1-\beta_1)^2} \leq \frac{8L^2}{n}$. Hence,

$$
C_g \leq \frac{4\rho^2(1+\rho^2)}{(1-\rho^2)^2}\frac{4\rho^2\alpha^2}{(1-\rho^2)^2}\frac{8L^2}{n} \quad (109)
$$

and

$$
8\alpha^2 nL^2 C_g \leq 8\alpha^2 nL^2\frac{4\rho^2(1+\rho^2)}{(1-\rho^2)^2}\frac{4\rho^2\alpha^2}{(1-\rho^2)^2}\frac{8L^2}{n} \leq \frac{1}{4},
$$

where the last inequality follows from $\alpha \leq \frac{1-\rho^2}{8L}$. This verifies the condition in (94). Therefore, both (96) and (97) hold.

Finally, notice that by (109) and $\alpha \leq \frac{(1-\rho^2)^2}{16L\sigma^{\frac{1}{2}}(nT)^{\frac{1}{4}}}$, we have

$$2C_g(2n\sigma^2 + 4\alpha^2 L^2\sigma^2) \leq \frac{4\rho^2(1+\rho^2)}{(1-\rho^2)^2}\frac{4\rho^2\alpha^2}{(1-\rho^2)^2}\frac{8L^2}{n}2(2n\sigma^2 + 4\alpha^2 L^2\sigma^2)$$

$$\leq \frac{\rho^4(1+\rho^2)}{n\sqrt{nT}\sigma}(2n\sigma^2 + 4\alpha^2 L^2\sigma^2) \leq \frac{2}{n\sqrt{nT}\sigma}(2n\sigma^2 + \sigma^2) = O\left(\frac{\sigma}{\sqrt{nT}}\right). \tag{110}$$

Hence, (96) implies $\frac{1}{T}\sum_{t=0}^{T-1}\mathbb{E}\left[\left\|\nabla f\left(\overline{\boldsymbol{x}}^t\right)\right\|^2\right] = O\left(\frac{\sigma}{\sqrt{nT}}\right)$. Also, by the same arguments in (110) and (97), we have $\frac{1}{T}\sum_{t=0}^{T-1}\left\|\mathbf{X}_\perp^t\right\|^2 = O\left(\frac{\sigma}{\sqrt{nT}}\right)$. Thus (108) follows. This completes the proof. $\qquad\square$

## B.2 General case of Algorithm 2

In this subsection, we analyze the convergence rate of Algorithm 2 in the general case. Again, we write the updates in the more compact matrix form:

$$\mathbf{G}^{t-\frac{1}{2}} = \mathbf{G}^{t-1} + \nabla\mathbf{F}^t - \nabla\mathbf{F}^{t-1},$$

$$\underline{\mathbf{G}}^t = \underline{\mathbf{G}}^{t-1} + \mathcal{Q}\left[\mathbf{G}^{t-\frac{1}{2}} - \underline{\mathbf{G}}^{t-1}\right],$$

$$\mathbf{G}^t = \mathbf{G}^{t-\frac{1}{2}} + \gamma_g\underline{\mathbf{G}}^t(\mathbf{W} - \mathbf{I}),$$

$$\mathbf{M}^t = \beta_1\mathbf{M}^{t-1} + (1-\beta_1)\mathbf{G}^t,$$

$$\mathbf{X}^{t+\frac{1}{2}} = \mathbf{X}^t - \alpha\mathbf{M}^t,$$

$$\underline{\mathbf{X}}^{t+1} = \underline{\mathbf{X}}^t + \mathcal{Q}\left[\mathbf{X}^{t+\frac{1}{2}} - \underline{\mathbf{X}}^t\right],$$

$$\mathbf{X}^{t+1} = \mathbf{X}^{t+\frac{1}{2}} + \gamma_x\underline{\mathbf{X}}^{t+1}(\mathbf{W} - \mathbf{I}).$$

Let $\widehat{\mathbf{W}}_x = \gamma_x\mathbf{W} + (1-\gamma_x)\mathbf{I}$ and $\widehat{\mathbf{W}}_g = \gamma_g\mathbf{W} + (1-\gamma_g)\mathbf{I}$. Then we can write the $\mathbf{X}$ and $\mathbf{G}$ updates to

$$\mathbf{X}^{t+1} = \mathbf{X}^{t+\frac{1}{2}}\widehat{\mathbf{W}}_x + \gamma_x\left(\underline{\mathbf{X}}^{t+1} - \mathbf{X}^{t+\frac{1}{2}}\right)(\mathbf{W} - \mathbf{I}), \tag{111}$$

$$\mathbf{G}^{t+1} = \mathbf{G}^{t+\frac{1}{2}}\widehat{\mathbf{W}}_g + \gamma_g\left(\underline{\mathbf{G}}^{t+1} - \mathbf{G}^{t+\frac{1}{2}}\right)(\mathbf{W} - \mathbf{I}). \tag{112}$$

Again when $\mathbf{W}$ satisfies the conditions in Assumption 2, $\widehat{\mathbf{W}}_x$ and $\widehat{\mathbf{W}}_g$ also satisfy all three conditions. Indeed, we have

$$\widehat{\rho}_x := \left\|\widehat{\mathbf{W}}_x - \mathbf{J}\right\|_2 < 1, \ \widehat{\rho}_g := \left\|\widehat{\mathbf{W}}_g - \mathbf{J}\right\|_2 < 1.$$

The next lemma directly follows from Yan et al. (2023, Lemma 15).

**Lemma B.5** *Under Assumptions 1, 2, 3, and 5, it holds that*

$$\mathbb{E}\left[\left\|\underline{\mathbf{G}}^{t+1} - \mathbf{G}^{t+\frac{1}{2}}\right\|^2\right] \leq 2\eta^2\mathbb{E}\left[\left\|\mathbf{G}^t - \underline{\mathbf{G}}^t\right\|^2\right] + 6\eta^2 n\sigma^2 + 4\eta^2 L^2\mathbb{E}\left[\left\|\mathbf{X}^{t+1} - \mathbf{X}^t\right\|^2\right],$$

$$\mathbb{E}\left[\left\|\underline{\mathbf{G}}^{t+1} - \mathbf{G}^{t+\frac{1}{2}}\right\|^2\right] \leq \frac{1+\eta^2}{2}\mathbb{E}\left[\left\|\mathbf{G}^t - \underline{\mathbf{G}}^t\right\|^2\right] + \frac{6n\sigma^2}{1-\eta^2} + \frac{4L^2}{1-\eta^2}\mathbb{E}\left[\left\|\mathbf{X}^{t+1} - \mathbf{X}^t\right\|^2\right].$$

**Lemma B.6** *Under Assumptions 1, 2, 3, and 5, let $\alpha$ and $\gamma_x$ satisfy*

$$\alpha \leq \frac{(1-\eta^2)^2}{32}, \ \gamma_x \leq \min\left\{\frac{1-\widehat{\rho}_x^2}{60\eta}, \frac{1-\eta^2}{25}, \frac{\alpha}{\eta}, \frac{\sqrt{2}-1}{2\eta}\right\}.$$

*Then*

$$\mathbb{E}\left[\left\|\mathbf{X}_{\perp}^{t+1}\right\|^2\right] \leq \frac{3+\widehat{\rho}_x^2}{4}\mathbb{E}\left[\left\|\mathbf{X}_{\perp}^t\right\|^2\right] + \alpha^2\frac{4\widehat{\rho}_x^2}{1-\widehat{\rho}_x^2}\mathbb{E}\left[\left\|\mathbf{M}_{\perp}^t\right\|^2\right]$$
$$+ \frac{4\eta\gamma_x(1-\widehat{\rho}_x^2)}{3}\mathbb{E}\left[\left\|\mathbf{X}^t - \underline{\mathbf{X}}^t\right\|^2\right] + \frac{\alpha^2}{45}\mathbb{E}\left[\left\|\mathbf{M}^t\right\|^2\right],$$
$$\mathbb{E}\left[\left\|\mathbf{X}^{t+1} - \underline{\mathbf{X}}^{t+1}\right\|^2\right] \leq \frac{1}{1-\eta^2}\mathbb{E}\left[\left\|\mathbf{X}_{\perp}^t\right\|^2\right] + \frac{3+\eta^2}{4}\mathbb{E}\left[\left\|\mathbf{X}^t - \underline{\mathbf{X}}^t\right\|^2\right]$$
$$+ \frac{\alpha^2}{1-\eta^2}\mathbb{E}\left[\left\|\mathbf{M}^t\right\|^2\right] + \frac{4\alpha^2}{1-\eta^2}\mathbb{E}\left[\left\|\mathbf{M}_{\perp}^t\right\|^2\right],$$
$$\mathbb{E}\left[\left\|\mathbf{X}^{t+1} - \mathbf{X}^t\right\|^2\right] \leq 4\alpha^2\mathbb{E}\left[\left\|\mathbf{M}^t\right\|^2\right] + 12\mathbb{E}\left[\left\|\mathbf{X}_{\perp}^t\right\|^2\right] + 4\sqrt{2}\eta\gamma_x\mathbb{E}\left[\left\|\underline{\mathbf{X}}^t - \mathbf{X}^t\right\|^2\right],$$
$$\mathbb{E}\left[\left\|\mathbf{M}^t\right\|^2\right] = \mathbb{E}\left[\left\|\mathbf{M}_{\perp}^t\right\|^2\right] + n\mathbb{E}\left[\left\|\overline{\boldsymbol{m}}^t\right\|^2\right],$$

*Proof.* By (16) with $\widehat{\mathbf{W}}$ replaced by $\widehat{\mathbf{W}}_x$, we have that for any $\eta_1 > 0$,

$$\left\|\mathbf{X}_{\perp}^{t+1}\right\|^2 \leq (1+\eta_1)\left\|\mathbf{X}^{t+\frac{1}{2}}\widehat{\mathbf{W}}_x(\mathbf{I}-\mathbf{J})\right\|^2 + 4\left(1+\eta_1^{-1}\right)\gamma_x^2\left\|\left(\underline{\mathbf{X}}^{t+1} - \mathbf{X}^{t+\frac{1}{2}}\right)\right\|^2. \tag{113}$$

In addition, by (20) with $\mathbf{Y}$ replaced by $\mathbf{M}$, we have that for any $\eta_2 > 0$,

$$\mathbb{E}\left[\left\|\underline{\mathbf{X}}^{t+1} - \mathbf{X}^{t+\frac{1}{2}}\right\|^2\right] \leq \eta^2(1+\eta_2)\mathbb{E}\left[\left\|\mathbf{X}^t - \underline{\mathbf{X}}^t\right\|^2\right] + \eta^2\left(1+\eta_2^{-1}\right)\alpha^2\mathbb{E}\left[\left\|\mathbf{M}^t\right\|^2\right] \tag{114}$$

Moreover, we obtain from (17) with $\mathbf{Y}_{\perp}^t$ replaced by $\mathbf{M}_{\perp}^t$ and $\widehat{\mathbf{W}}$ replaced by $\widehat{\mathbf{W}}_x$ that

$$\left\|\mathbf{X}^{t+\frac{1}{2}}\widehat{\mathbf{W}}_x(\mathbf{I}-\mathbf{J})\right\|^2 \leq \frac{1+\widehat{\rho}_x^2}{2}\left\|\mathbf{X}_{\perp}^t\right\|^2 + \frac{2\widehat{\rho}_x^2\alpha^2}{1-\widehat{\rho}_x^2}\left\|\mathbf{M}_{\perp}^t\right\|^2. \tag{115}$$

Substituting (115) and (114) with $\eta_2 = 1$ into (113) with $\eta_1 = \frac{7\eta\gamma_x}{1-\widehat{\rho}^2}$, we obtain the first desired result by using (23).

By (25), we obtain that for any $\eta_3 > 0$,

$$\mathbb{E}\left[\left\|\mathbf{X}^{t+1} - \underline{\mathbf{X}}^{t+1}\right\|^2\right] \leq (1+\eta_3)(1+2\gamma_x)^2\mathbb{E}\left[\left\|\underline{\mathbf{X}}^{t+1} - \mathbf{X}^{t+\frac{1}{2}}\right\|^2\right] + \left(1+\eta_3^{-1}\right)4\gamma_x^2\mathbb{E}\left[\left\|\mathbf{X}_{\perp}^{t+\frac{1}{2}}\right\|^2\right]. \tag{116}$$

Also, it follows from (26) with $\mathbf{Y}$ replaced by $\mathbf{M}$ that

$$\mathbb{E}\left[\left\|\mathbf{X}_{\perp}^{t+\frac{1}{2}}\right\|^2\right] \leq 2\mathbb{E}\left[\left\|\mathbf{X}_{\perp}^t\right\|^2\right] + 2\alpha^2\mathbb{E}\left[\left\|\mathbf{M}_{\perp}^t\right\|^2\right]. \tag{117}$$

Substituting (117) and (114) with $\eta_2 = \frac{1-\eta^2}{2\eta^2}$ into (116) with $\eta_3 = \frac{1-\eta^2}{12}$, we obtain the second desired result by using (28).

To show the third desired inequality, we notice

$$\mathbb{E}\left[\left\|\mathbf{X}^{t+1} - \mathbf{X}^t\right\|^2\right] = \mathbb{E}\left[\left\|\mathbf{X}^{t+\frac{1}{2}}\widehat{\mathbf{W}}_x - \mathbf{X}^t + \gamma_x\left(\underline{\mathbf{X}}^{t+1} - \mathbf{X}^{t+\frac{1}{2}}\right)(\mathbf{W}-\mathbf{I})\right\|^2\right]$$

$$\leq (1+\eta_4)\,\mathbb{E}\left[\left\|\mathbf{X}^{t+\frac{1}{2}}\widehat{\mathbf{W}}_x - \mathbf{X}^t\right\|^2\right] + \left(1+\eta_4^{-1}\right)\mathbb{E}\left[\left\|\gamma_x\left(\underline{\mathbf{X}}^{t+1} - \mathbf{X}^{t+\frac{1}{2}}\right)(\mathbf{W}-\mathbf{I})\right\|^2\right]$$

$$\leq (1+\eta_4)\,\mathbb{E}\left[\left\|(\mathbf{X}^{t+\frac{1}{2}}-\mathbf{X}^t)\widehat{\mathbf{W}}_x + \mathbf{X}^t(\widehat{\mathbf{W}}_x - \mathbf{I})\right\|^2\right] + 4\left(1+\eta_4^{-1}\right)\gamma_x^2\mathbb{E}\left[\left\|\underline{\mathbf{X}}^{t+1} - \mathbf{X}^{t+\frac{1}{2}}\right\|^2\right]$$

$$\leq (1+\eta_4)\left(2\mathbb{E}\left[\left\|\mathbf{X}^{t+\frac{1}{2}}-\mathbf{X}^t\right\|^2\right] + 8\mathbb{E}\left[\left\|\mathbf{X}_\perp^t\right\|^2\right]\right) \tag{118}$$

$$+ 8\left(1+\eta_4^{-1}\right)\gamma_x^2\eta^2\left(\mathbb{E}\left[\left\|\mathbf{X}^{t+\frac{1}{2}}-\mathbf{X}^t\right\|^2\right] + \mathbb{E}\left[\left\|\underline{\mathbf{X}}^t - \mathbf{X}^t\right\|^2\right]\right)$$

$$\leq 4\mathbb{E}\left[\left\|\mathbf{X}^{t+\frac{1}{2}}-\mathbf{X}^t\right\|^2\right] + 12\mathbb{E}\left[\left\|\mathbf{X}_\perp^t\right\|^2\right] + 4\sqrt{2}\eta\gamma_x\mathbb{E}\left[\left\|\underline{\mathbf{X}}^t - \mathbf{X}^t\right\|^2\right]$$

$$= 4\alpha^2\mathbb{E}\left[\left\|\mathbf{M}^t\right\|^2\right] + 12\mathbb{E}\left[\left\|\mathbf{X}_\perp^t\right\|^2\right] + 4\sqrt{2}\eta\gamma_x\mathbb{E}\left[\left\|\underline{\mathbf{X}}^t - \mathbf{X}^t\right\|^2\right], \tag{119}$$

where $\eta_4$ is any positive scalar, the second inequality holds by $\|\mathbf{W}-\mathbf{I}\|_2 \leq 2$, the third one follows from (20), and in the fourth inequality, we take $\eta_4 = 2\gamma_x\eta$ and have from $\gamma_x \leq \frac{\sqrt{2}-1}{2\eta}$ that

$$2(1+\eta_4) + 8\left(1+\eta_4^{-1}\right)\gamma_x^2\eta^2 = 2(2\gamma_x\eta+1)^2 \leq 4, \ 8(1+\eta_4) \leq 12, \ 8\left(1+\eta_4^{-1}\right)\gamma_x^2\eta^2 \leq 4\sqrt{2}\eta\gamma_x.$$

This completes the proof of the third desired inequality. The fourth desired equation follows straightforwardly from the fact $\langle \mathbf{M}_\perp, \overline{\mathbf{M}}\rangle = 0$. $\qquad\square$

The following lemma bounds the consensus error and compression error of $\mathbf{G}$.

**Lemma B.7** *Under Assumptions 1, 2, 3, and 5, let $\gamma_g \leq \min\left\{\frac{\sqrt{1-\widehat{\rho}_g^2}}{12\eta}, \frac{1-\eta^2}{25}\right\}$. Then*

$$\mathbb{E}\left[\left\|\mathbf{G}_\perp^{t+1}\right\|^2\right] \leq \frac{2+\widehat{\rho}_g^2}{3}\mathbb{E}\left[\left\|\mathbf{G}_\perp^t\right\|^2\right] + \frac{48\eta^2\gamma_g^2}{1-\widehat{\rho}_g^2}\mathbb{E}\left[\left\|\mathbf{G}^t - \underline{\mathbf{G}}^t\right\|^2\right] + 11n\sigma^2 + \frac{5L^2}{1-\widehat{\rho}_g^2}\mathbb{E}\left[\left\|\mathbf{X}^{t+1} - \mathbf{X}^t\right\|^2\right],$$

$$\mathbb{E}\left[\left\|\mathbf{G}^{t+1} - \underline{\mathbf{G}}^{t+1}\right\|^2\right] \leq \frac{3+\eta^2}{4}\mathbb{E}\left[\left\|\mathbf{G}^t - \underline{\mathbf{G}}^t\right\|^2\right] + \frac{104\gamma_g^2}{1-\eta^2}\mathbb{E}\left[\left\|\mathbf{G}_\perp^t\right\|^2\right] + \frac{6L^2}{1-\eta^2}\mathbb{E}\left[\left\|\mathbf{X}^{t+1} - \mathbf{X}^t\right\|^2\right] + \frac{9n\sigma^2}{1-\eta^2}.$$

*Proof.* By the second inequality after (69) in Yan et al. (2023, Lemma 18) (with $\mathbf{Y}$ in Yan et al. (2023) replaced by our notation $\mathbf{G}$), we have the first desired inequality. The second desired inequality follows from the inductions after (71) in Yan et al. (2023, Lemma 18) (with $\mathbf{Y}$ in Yan et al. (2023) replaced by our notation $\mathbf{G}$). $\qquad\square$

Notice that Lemmas B.2, B.3, and (86) still hold for the compressed case. With these, we are ready to show the convergence rate of Algorithm 2 in the general case.

Denote

$$\Omega^t = \left(\frac{4}{\alpha}(\mathbb{E}[f(\boldsymbol{z}^{t+1})] - f^*), \mathbb{E}\left[\left\|\mathbf{X}_\perp^{t+1}\right\|^2\right], \mathbb{E}\left[\left\|\mathbf{X}^{t+1} - \underline{\mathbf{X}}^{t+1}\right\|^2\right], \mathbb{E}\left[\left\|\overline{\boldsymbol{m}}^t\right\|^2\right],\right.$$

$$\left.\mathbb{E}\left[\left\|\mathbf{M}_\perp^{t+1}\right\|^2\right], \mathbb{E}\left[\left\|\mathbf{G}_\perp^{t+1}\right\|^2\right], \mathbb{E}\left[\left\|\mathbf{G}^{t+1} - \underline{\mathbf{G}}^{t+1}\right\|^2\right]\right)^\top.$$

Then Lemmas B.2, B.3 and (86), together with Lemmas B.5–B.7, imply

$$\Omega^t \leq \mathbf{A}\Omega^{t-1} + \overline{\mathbf{A}}\Omega^t + \boldsymbol{b}^t + \mathbf{c}, \tag{120}$$

where

$$\mathbf{A} = \begin{pmatrix} 1 & \frac{4\alpha L^3}{n} + \frac{6L^2}{n} & 0 & \frac{6L^2\alpha^2\beta_1^2}{(1-\beta_1)^2} & 0 & 0 & 0 \\ 0 & \frac{3+\widehat{\rho}_x^2}{4} & \frac{4\eta\gamma_x(1-\widehat{\rho}_x^2)}{3} & 0 & \alpha^2\left(\frac{4\widehat{\rho}_x^2}{1-\widehat{\rho}_x^2} + \frac{1}{45}\right) & 0 & 0 \\ 0 & \frac{1}{1-\eta^2} & \frac{3+\eta^2}{4} & 0 & \frac{5\alpha^2}{1-\eta^2} & 0 & 0 \\ 0 & (1-\beta_1)\frac{2L^2}{n} & 0 & \beta_1 & 0 & 0 & 0 \\ 0 & 0 & 0 & 0 & \beta_1 & 0 & 0 \\ 0 & \frac{60L^2}{1-\widehat{\rho}_g^2} & \frac{20\sqrt{2}\eta\gamma_x L^2}{1-\widehat{\rho}_g^2} & 0 & \frac{20\alpha^2 L^2}{1-\widehat{\rho}_g^2} & \frac{2+\widehat{\rho}_g^2}{3} & \frac{48\eta^2\gamma_g^2}{1-\widehat{\rho}_g^2} \\ 0 & \frac{72L^2}{1-\eta^2} & \frac{24\sqrt{2}\eta\gamma_x L^2}{1-\eta^2} & 0 & \frac{24\alpha^2 L^2}{1-\eta^2} & \frac{104\gamma_g^2}{1-\eta^2} & \frac{3+\eta^2}{4} \end{pmatrix},$$

$$\overline{\mathbf{A}} = \begin{pmatrix} 0 & 0 & 0 & 0 & 0 & 0 & 0 \\ 0 & 0 & 0 & \frac{\alpha^2 n}{45} & 0 & 0 & 0 \\ 0 & 0 & 0 & \frac{\alpha^2 n}{1-\eta^2} & 0 & 0 & 0 \\ 0 & 0 & 0 & 0 & 0 & 0 & 0 \\ 0 & 0 & 0 & 0 & 0 & 1-\beta_1 & 0 \\ 0 & 0 & 0 & \frac{20\alpha^2 nL^2}{1-\widehat{\rho}_g^2} & 0 & 0 & 0 \\ 0 & 0 & 0 & \frac{24\alpha^2 nL^2}{1-\eta^2} & 0 & 0 & 0 \end{pmatrix}, \quad \mathbf{c} = \begin{pmatrix} \frac{2\alpha L\sigma^2}{n} \\ 0 \\ 0 \\ (1-\beta_1)\frac{\sigma^2}{n} \\ 0 \\ 11n\sigma^2 \\ \frac{9n\sigma^2}{1-\eta^2} \end{pmatrix},$$

$$\mathbf{b}^t = \begin{pmatrix} -\mathbb{E}\left[\|\nabla f(\overline{\boldsymbol{x}}^t)\|^2\right] \\ 0 \\ 0 \\ 2(1-\beta_1)\mathbb{E}\left[\|\nabla f(\overline{\boldsymbol{x}}^t)\|^2\right] \\ 0 \\ 0 \\ 0 \end{pmatrix}, \text{ for all } t \geq 0.$$

**Theorem B.3 (Complete statement of Theorem 4.3)** *Suppose Assumptions 1, 2, 3, and 5 hold. Let* $\gamma_x, \gamma_g$ *and* $\alpha$ *satisfy*

$$\gamma_x \leq \min\left\{\frac{1-\widehat{\rho}_x^2}{60\eta}, \frac{1-\eta^2}{25}, \frac{\alpha}{\eta}, \frac{\sqrt{2}-1}{2\eta}\right\}, \quad \gamma_g \leq \min\left\{\frac{1-\widehat{\rho}_g^2}{25\eta}, \frac{1-\widehat{\rho}_g^2}{25L}, \frac{1-\eta^2}{25}, \frac{1-\eta^2}{25L}\right\},$$

$$\alpha \leq \min\left\{\frac{1}{16b}, \frac{\gamma_g(1-\eta^2)}{32}, \frac{\gamma_g(1-\eta^2)}{12L\sqrt{n}}, \frac{\gamma_g(1-\widehat{\rho}_x^2)}{12\sqrt{b}}, \frac{\gamma_g(1-\widehat{\rho}_g^2)}{12L\sqrt{n}}, \right.$$

$$\left. \sqrt{\frac{(1-\beta_1)\gamma_g^2}{2L\left(\frac{6L\beta_1^2}{(1-\beta_1)^2} + \frac{b}{45(1-\widehat{\rho}_x^2)} + \frac{24L^2+1}{(1-\eta^2)^2} + \frac{20L^2}{(1-\widehat{\rho}_g^2)^2}\right)}} \right\} \tag{121}$$

*with* $b = 4\left(9L + 1 + \frac{72L^2+1}{(1-\eta^2)^2} + \frac{60L^2}{(1-\widehat{\rho}_g^2)^2}\right)$. *Then it holds that*

$$\frac{1}{2LT}\sum_{t=0}^{T-1}\mathbb{E}\left[\|\nabla f(\overline{\boldsymbol{x}}^t)\|^2\right] + \frac{1}{nT}\sum_{t=0}^{T-1}\mathbb{E}\left[\|\mathbf{X}_\perp^t\|^2\right]$$

$$\leq \frac{4}{\alpha LT}(f(\overline{\boldsymbol{x}}^0) - f^*) + \frac{2\alpha\sigma^2}{n} + \frac{\gamma_g^2\sigma^2(1-\beta_1)}{2Ln} + \frac{11\gamma_g^2\sigma^2}{1-\widehat{\rho}_g^2} + \frac{9\gamma_g^2\sigma^2}{(1-\eta^2)^2}.$$

*Proof.* For any $\mathbf{q} = (q_1, q_2, q_3, q_4, q_5, q_6, q_7)^\top \geq \mathbf{0}$, multiplying $\boldsymbol{q}^\top$ to both sides of (120) gives

$$(\mathbf{q}^\top - \mathbf{q}^\top\overline{\mathbf{A}})\Omega^t \leq \mathbf{q}^\top\mathbf{A}\Omega^{t-1} + \mathbf{q}^\top\boldsymbol{b}^t + \mathbf{q}^\top\mathbf{c}$$

$$= (\mathbf{q}^\top - \mathbf{q}^\top\overline{\mathbf{A}})\Omega^{t-1} + (\mathbf{q}^\top\mathbf{A} + \mathbf{q}^\top\overline{\mathbf{A}} - \mathbf{q}^\top)\Omega^{t-1} + \mathbf{q}^\top\boldsymbol{b}^t + \mathbf{q}^\top\mathbf{c}. \tag{122}$$

Let

$$q_1 = \frac{n}{L}, \ q_2 = \frac{b}{1 - \widehat{\rho}_x^2}, \ q_3 = \frac{1}{1 - \eta^2},$$

$$q_4 = \frac{n\alpha^2}{1 - \beta_1} \left( \frac{6L\beta_1^2}{(1 - \beta_1)^2} + \frac{b}{45(1 - \widehat{\rho}_x^2)} + \frac{24L^2 + 1}{(1 - \eta^2)^2} + \frac{20L^2}{(1 - \widehat{\rho}_g^2)^2} \right),$$

$$q_5 = \frac{\gamma_g^2}{6(1 - \beta_1)}, \ q_6 = \frac{\gamma_g^2}{1 - \widehat{\rho}_g^2}, \ q_7 = \frac{\gamma_g^2}{1 - \eta^2}.$$

With the choice of $\alpha$, we claim that

$$\boldsymbol{\lambda} := \mathbf{q}^\top \mathbf{A} + \mathbf{q}^\top \overline{\mathbf{A}} - \mathbf{q}^\top \leq (0, -1, 0, 0, 0, 0, 0). \tag{123}$$

First notice

$$q_4 \leq \frac{n\gamma_g^2}{2L} \leq \frac{n}{8L}, \ \gamma_g \leq \frac{n}{2L},$$

and it is straightforward to have $\lambda_1 = 0$. Second,

$$\lambda_2 = 4 \left( \frac{\alpha L^3}{n} + \frac{3L^2}{2n} \right) q_1 + \frac{3 + \widehat{\rho}_x^2}{4} q_2 + \frac{1}{1 - \eta^2} q_3 + (1 - \beta_1) \frac{2L^2}{n} q_4 + \frac{60L^2}{1 - \widehat{\rho}_g^2} q_6 + \frac{72L^2}{1 - \eta^2} q_7 - q_2$$

$$= 4 \left( \alpha L^2 + \frac{3L}{2} \right) - \frac{b}{4} + \frac{1}{(1 - \eta^2)^2} + (1 - \beta_1) \frac{2L^2}{n} q_4 + \frac{60\gamma_g^2 L^2}{(1 - \widehat{\rho}_g^2)^2} + \frac{72\gamma_g^2 L^2}{(1 - \eta^2)^2}$$

$$\leq 8L - \frac{b}{4} + \frac{1}{(1 - \eta^2)^2} + (1 - \beta_1) L + \frac{60\gamma_g^2 L^2}{(1 - \widehat{\rho}_g^2)^2} + \frac{72\gamma_g^2 L^2}{(1 - \eta^2)^2} \leq -1,$$

where the first inequality follows from $\alpha \leq \frac{1}{2L}$, and $q_4 \leq \frac{n}{2L}$, and the second inequality holds by $\gamma_g \leq 1$ and the definition of $b$. Third,

$$\lambda_3 = \frac{4\eta\gamma_x(1 - \widehat{\rho}_x^2)}{3} q_2 + \frac{3 + \eta^2}{4} q_3 + \frac{20\sqrt{2}\eta\gamma_x L^2}{1 - \widehat{\rho}_g^2} q_6 + \frac{24\sqrt{2}\eta\gamma_x L^2}{1 - \eta^2} q_7 - q_3$$

$$= \frac{4\eta\gamma_x b}{3} - \frac{1}{4} + \frac{20\sqrt{2}\eta\gamma_g^2\gamma_x L^2}{(1 - \widehat{\rho}_g^2)^2} + \frac{24\sqrt{2}\eta\gamma_g^2\gamma_x L^2}{(1 - \eta^2)^2} \leq \frac{4\alpha b}{3} - \frac{1}{4} + \frac{1}{12} + \frac{1}{12} \leq 0,$$

by the choice of $\gamma_g, \gamma_x$, and $\alpha \leq \frac{1}{16b}$. Fourth,

$$\lambda_4 = \frac{6L^2\alpha^2\beta_1^2}{(1 - \beta_1)^2} q_1 + \beta_1 q_4 - q_4 + \frac{\alpha^2 n}{45} q_2 + \frac{\alpha^2 n}{1 - \eta^2} q_3 + \frac{20\alpha^2 n L^2}{1 - \widehat{\rho}_g^2} q_6 + \frac{24\alpha^2 n L^2}{1 - \eta^2} q_7$$

$$= \frac{6Ln\alpha^2\beta_1^2}{(1 - \beta_1)^2} - (1 - \beta_1) q_4 + \frac{\alpha^2 n}{45} \frac{b}{1 - \widehat{\rho}_x^2} + \frac{\alpha^2 n}{(1 - \eta^2)^2} + \frac{20\alpha^2\gamma_g^2 n L^2}{(1 - \widehat{\rho}_g^2)^2} + \frac{24\alpha^2\gamma_g^2 n L^2}{(1 - \eta^2)^2} \leq 0,$$

by $\alpha, \gamma_g \leq 1$ and the choice of $q_4$. Fifth,

$$\lambda_5 = \alpha^2 \left( \frac{4\widehat{\rho}_x^2}{1 - \widehat{\rho}_x^2} + \frac{1}{45} \right) q_2 + \frac{5\alpha^2}{1 - \eta^2} q_3 + \beta_1 q_5 + \frac{20\alpha^2 L^2}{1 - \widehat{\rho}_g^2} q_6 + \frac{24\alpha^2 L^2}{1 - \eta^2} q_7 - q_5$$

$$= \alpha^2 \left( \frac{4\widehat{\rho}_x^2}{1 - \widehat{\rho}_x^2} + \frac{1}{45} \right) \frac{b}{1 - \widehat{\rho}_x^2} + \frac{5\alpha^2}{(1 - \eta^2)^2} - (1 - \beta_1) q_5 + \frac{20\alpha^2\gamma_g^2 L^2}{(1 - \widehat{\rho}_g^2)^2} + \frac{24\alpha^2\gamma_g^2 L^2}{(1 - \eta^2)^2} < 0,$$

by $\alpha \leq \min \left\{ \frac{\gamma_g(1 - \eta^2)}{32}, \frac{\gamma_g(1 - \eta^2)}{12L\sqrt{n}}, \frac{\gamma_g(1 - \widehat{\rho}_x^2)}{12\sqrt{b}}, \frac{\gamma_g(1 - \widehat{\rho}_g^2)}{12L\sqrt{n}} \right\}$ and $q_5 = \frac{\gamma_g^2}{6(1 - \beta_1)}$. Sixth,

$$\lambda_6 = (1 - \beta_1) q_5 + \frac{2 + \widehat{\rho}_g^2}{3} q_6 + \frac{104\gamma_g^2}{1 - \eta^2} q_7 - q_6 = \frac{\gamma_g^2}{6} - \frac{\gamma_g^2}{3} + \frac{104\gamma_g^4}{(1 - \eta^2)^2} \leq 0$$

by $\gamma_g \leq \frac{1-\eta^2}{25}$. Seventh,

$$\lambda_7 = \frac{48\eta^2\gamma_g^2}{1-\widehat{\rho}_g^2}q_6 + \frac{3+\eta^2}{4}q_7 - q_7 = \frac{48\eta^2\gamma_g^4}{(1-\widehat{\rho}_g^2)^2} - \frac{\gamma_g^2}{4} \leq 0$$

by $\gamma_g \leq \frac{1-\widehat{\rho}_g^2}{25\eta}$. Thus, (123) is obtained. From (122) and (123), it then holds

$$(0,1,0,0,0,0,0)\Omega^{t-1} \leq (\mathbf{q}^\top - \mathbf{q}^\top\overline{\mathbf{A}})\Omega^{t-1} - (\mathbf{q}^\top - \mathbf{q}^\top\overline{\mathbf{A}})\Omega^t + \mathbf{q}^\top\boldsymbol{b}^t + \mathbf{q}^\top\mathbf{c},$$

which implies

$$\mathbb{E}\left[\|\mathbf{X}_\perp^t\|^2\right] \leq (\mathbf{q}^\top - \mathbf{q}^\top\overline{\mathbf{A}})\Omega^{t-1} - (\mathbf{q}^\top - \mathbf{q}^\top\overline{\mathbf{A}})\Omega^t + \mathbf{q}^\top\boldsymbol{b}^t + \mathbf{q}^\top\mathbf{c}. \tag{124}$$

Summing up (124) over $t = 0, \ldots, T-1$ and then dividing by $nT$ gives

$$\frac{1}{nT}\sum_{t=0}^{T-1}\mathbb{E}\left[\|\mathbf{X}_\perp^t\|^2\right] \leq \frac{1}{nT}(\mathbf{q}^\top - \mathbf{q}^\top\overline{\mathbf{A}})\left(\Omega^{-1} - \Omega^{T-1}\right) + \frac{1}{nT}\sum_{t=0}^{T-1}\mathbf{q}^\top\boldsymbol{b}^t + \frac{1}{n}\mathbf{q}^\top\mathbf{c}. \tag{125}$$

Notice $\mathbf{q}^\top - \mathbf{q}^\top\overline{\mathbf{A}} = \boldsymbol{q}^\top\mathbf{A} - \boldsymbol{\lambda} \geq \mathbf{0}$ and $\Omega^{T-1} \geq \mathbf{0}$. Hence $(\mathbf{q}^\top - \mathbf{q}^\top\overline{\mathbf{A}})\Omega^{T-1} \geq \mathbf{0}$, and thus

$$\frac{1}{nT}(\mathbf{q}^\top - \mathbf{q}^\top\overline{\mathbf{A}})\left(\Omega^{-1} - \Omega^{T-1}\right) \leq \frac{1}{nT}(\mathbf{q}^\top - \mathbf{q}^\top\overline{\mathbf{A}})\Omega^{-1}$$

$$= \frac{4}{\alpha nT}q_1(f(\boldsymbol{z}^0) - f^*) = \frac{4}{\alpha LT}(f(\boldsymbol{z}^0) - f^*). \tag{126}$$

In addition, we know from $q_4 \leq \frac{n\gamma_g^2}{2L}$ that

$$\frac{1}{n}\mathbf{q}^\top\mathbf{c} \leq \frac{2\alpha\sigma^2}{n} + \frac{\gamma_g^2\sigma^2(1-\beta_1)}{2Ln} + \frac{11\gamma_g^2\sigma^2}{1-\widehat{\rho}_g^2} + \frac{9\gamma_g^2\sigma^2}{(1-\eta^2)^2}. \tag{127}$$

Moreover, it holds

$$\frac{1}{nT}\sum_{t=0}^{T-1}\mathbf{q}^\top\boldsymbol{b}^t = \frac{1}{nT}\sum_{t=0}^{T-1}\left(-\frac{n}{L}\mathbb{E}\left[\|\nabla f(\overline{\boldsymbol{x}}^t)\|^2\right] + 2q_4(1-\beta_1)\mathbb{E}\left[\|\nabla f(\overline{\boldsymbol{x}}^t)\|^2\right]\right)$$

$$\leq \frac{1}{nT}\sum_{t=0}^{T-1}\left(-\frac{n}{L}\mathbb{E}\left[\|\nabla f(\overline{\boldsymbol{x}}^t)\|^2\right] + \frac{n(1-\beta_1)}{4L}\mathbb{E}\left[\|\nabla f(\overline{\boldsymbol{x}}^{t-1})\|^2\right]\right)$$

$$= -\frac{1}{2LT}\sum_{t=0}^{T-1}\mathbb{E}\left[\|\nabla f(\overline{\boldsymbol{x}}^t)\|^2\right], \tag{128}$$

where the first inequality follows from $q_4 \leq \frac{n}{8L}$.

Substituting (126)–(128) into (125) yields

$$\frac{1}{2LT}\sum_{t=0}^{T-1}\mathbb{E}\left[\|\nabla f(\overline{\boldsymbol{x}}^t)\|^2\right] + \frac{1}{nT}\sum_{t=0}^{T-1}\mathbb{E}\left[\|\mathbf{X}_\perp^t\|^2\right]$$

$$\leq \frac{4}{\alpha LT}(f(\boldsymbol{z}^0) - f^*) + \frac{2\alpha\sigma^2}{n} + \frac{\gamma_g^2\sigma^2(1-\beta_1)}{2Ln} + \frac{11\gamma_g^2\sigma^2}{1-\widehat{\rho}_g^2} + \frac{9\gamma_g^2\sigma^2}{(1-\eta^2)^2}.$$

The proof is then completed by noticing $\boldsymbol{z}^0 = \overline{\boldsymbol{x}}^0$. $\qquad\square$

The theorem below directly follows from Theorem B.3 by plugging the specified algorithmic parameters and ignoring certain constants that are independent of $n$ and $T$.

**Theorem B.4** *Suppose that the conditions assumed in Theorem B.3 hold and $\alpha = \frac{\theta_1 \sqrt{n}}{\sigma \sqrt{T}}$, $\gamma_g = \frac{\theta_2 \sqrt{n}}{\sigma \sqrt{T}}$ and $\gamma_x = \frac{\theta_3 \sqrt{n}}{\sigma \sqrt{T}}$ for some $\theta_1, \theta_2, \theta_3 \in (0,1)$ independent of $T$ and $n$. Also, suppose $n \leq T$. Then*

$$\frac{1}{T} \sum_{t=0}^{T-1} \mathbb{E} \left[ \|\nabla f\left(\overline{\boldsymbol{x}}^t\right)\|^2 + \frac{1}{n}\|\mathbf{X}_\perp^t\|^2 \right] = O\left( \frac{\sigma}{\sqrt{nT}} + \frac{n}{T}\left( \frac{1}{1 - \widehat{\rho}_g^2} + \frac{1}{(1 - \eta^2)^2} \right) \right). \tag{129}$$

**Remark 4 (Linear speed up and topology independence)** *From Theorem B.4, we see that if*

$$T = \Omega\left( \max\left\{ \frac{n}{\sigma^2(1 - \widehat{\rho}_g^2)^6}, \frac{n}{\sigma^2(1 - \widehat{\rho}_x^2)^6}, \frac{n}{\sigma^2(1 - \eta^2)^6}, \frac{n^2}{\sigma^2(1 - \widehat{\rho}_g^2)^4}, \frac{n^2}{\sigma^2(1 - \widehat{\rho}_x^2)^4}, \frac{n^3}{\sigma^2(1 - \widehat{\rho}_g^2)^2}, \frac{n^3}{\sigma^2(1 - \eta^2)^4} \right\} \right), \tag{130}$$

*then $\alpha = O(\frac{\sqrt{n}}{\sigma\sqrt{T}})$, $\gamma_g = O(\frac{\sqrt{n}}{\sigma\sqrt{T}})$, $\gamma_x = O(\frac{\sqrt{n}}{\sigma\sqrt{T}})$, $\frac{n}{T}\left( \frac{1}{1-\widehat{\rho}_g^2} + \frac{1}{(1-\eta^2)^2} \right) = O(\frac{\sigma}{\sqrt{nT}})$, and the RHS of (129) becomes $O(\frac{\sigma}{\sqrt{nT}})$. Thus we have*

$$\frac{1}{T} \sum_{t=0}^{T-1} \mathbb{E} \left[ \|\nabla f\left(\overline{\boldsymbol{x}}^t\right)\|^2 + \frac{1}{n}\|\mathbf{X}_\perp^t\|^2 \right] = O\left( \frac{\sigma}{\sqrt{nT}} \right). \tag{131}$$

*For a given $\varepsilon > 0$, let $T = \Theta\left( \frac{\sigma^2}{n\varepsilon^4} \right)$ and $\tau$ be selected from $\{0, 1, \ldots, T-1\}$ uniformly at random. Then $\mathbf{X}^\tau$ is an $\varepsilon$-stationary solution in expectation, i.e., $\mathbb{E}\left[ \|\nabla f\left(\overline{\boldsymbol{x}}^\tau\right)\|^2 + \frac{1}{n}\|\mathbf{X}_\perp^\tau\|^2 \right] \leq \varepsilon^2$. When $\varepsilon$ is sufficiently small such that $T = \Theta\left( \frac{\sigma^2}{n\varepsilon^4} \right)$ satisfies the conditions in (130), we obtain linear speed up, and the algorithmic parameters are independent of the communication graph.*

## C  Examples of Compression Operators that satisfy Assumption 3

In this section, we provide a few concrete examples of compression operators that satisfy the condition in Assumption 3. More examples that satisfy Assumption 3 can be found in Chen et al. (2023a); Koloskova et al. (2019).

**Example C.1** *QSGD (Alistarh et al., 2017) compresses $\mathbf{x} \in \mathbb{R}^d$ by $Q_{sgd}(\mathbf{x}) = \frac{\text{sign}(\mathbf{x})\|\mathbf{x}\|}{s} \left\lfloor s \frac{|\mathbf{x}|}{\|\mathbf{x}\|} + \xi \right\rfloor$ where $\xi$ is uniformly distributed on $[0,1]^d$, $s$ is a parameter about compression level. Then $\mathcal{Q}(\mathbf{x}) := \frac{1}{\tau} Q_{ssgd}(\mathbf{x})$ with $\tau = \left( 1 + \min\left\{ d/s^2, \sqrt{d}/s \right\} \right)$ satisfies Assumption 3 with $\eta = 1 - \frac{1}{\tau}$.*

**Example C.2** *$Q_{sparse}(\mathbf{x})$ (Stich et al., 2018) randomly selects $k$ out of $d$ coordinates from $\boldsymbol{x}$, or the $k$ coordinates with the largest values in magnitude from $\boldsymbol{x}$. Then $Q_{sparse}(\mathbf{x})$ satisfies Assumption 3 with $\eta = \frac{k}{d}$.*

**Example C.3** *$Q_{gossip}(\mathbf{x})$ sets $Q_{gossip}(\mathbf{x}) = \boldsymbol{x}$ with probability $p \in [0,1]$ and $Q_{gossip}(\mathbf{x}) = 0$ with probability $1 - p$. Then $Q_{gossip}(\mathbf{x})$ satisfies Assumption 3 with $\eta = p$.*

## D  More Detailed Comparisons

In this section, we provide more detailed comparisons with related work.

### D.1  Term-by-term Comparison with Prior Work in Table 1

To complement Table 1, we provide a detailed term-by-term comparison of theoretical convergence results between our proposed algorithms (DAMSCo and DaSHCo) and related methods listed in Table 1.

**CMP, AG, MMT**: DADAM(Nazari et al., 2022), DAGM(Chen et al., 2023b), and our proposed DAMSCo incorporate both adaptive gradient updates and momentum acceleration. However, unlike DAMSCo, DADAM and DAGM do not utilize compressed communication. Choco-SGD(Koloskova et al., 2019),

BEER(Zhao et al., 2022a), and CDProxSGT (Yan et al., 2023) perform compressed communication but do not apply adaptive gradient updates or momentum acceleration. The compression effects in these methods are represented through a simple reduction factor of $(1 - \eta)$ per communication round, while this is absent from DADAM and DAGM.

**DH**: DADAM, DAGM, Choco-SGD, and our first method DAMSCo require the bounded gradient assumption. This limits their theoretical guarantees in scenarios with DH. In contrast, our second method DaSHCo, and CDProxSGT and BEER do not need this assumption, thus being able to handle DH.

**CMR**: In terms of communication rounds, the number required per iteration varies depending on the use of gradient tracking. BEER, CDProxSGT, DAGM, and our proposed DaSHCo perform gradient tracking and transmit both model parameters and local gradients at each iteration, leading to two communication rounds. In contrast, Choco-SGD and our proposed DAMSCo, without gradient tracking, require only one communication round per iteration.

**LS, TI, convergence rate**: Here, LS is characterized by a clear $\frac{1}{n}$ dependency in the convergence rate, showing that the convergence rate improves linearly with the number of agents ($n$); TI refers to the capability of choosing learning rate independent of the topology parameter $\rho$. We do not compare to the convergence rate of Choco-SGD as it assume strong convexity, which is not directly comparable with our considered nonconvex case. When $T$ is sufficiently large, all methods except Choco-SGD in Table 1 have the convergence rate of $O\left(\frac{1}{\sqrt{nT}}\right)$ and thus achieve LS and TI.

### D.2 Comparisons to more related work

In this section, we compare our proposed algorithms (DAMSCo and DaSHCo) to several more existing decentralized stochastic optimization methods.

Our first method DAMSCo is designed with adaptive gradient updates, i.e., Adam-type updates. Its convergence rate is $O(1/\sqrt{T})$ and aligns with the rates established for existing (nondistributed or distributed) stochastic nonconvex optimization algorithms that use Adam-type updates.

For momentum-based algorithms, a few recent works enhance theoretical convergence rates by incorporating variance reduction (VR) techniques. However, we observe that, in practice, VR-based methods can exhibit unstable performance and poor generalization, particularly in training large-scale deep neural networks (DNNs). Specifically, through additional experiments comparing our algorithms DAMSCo and DaSHCo against the VR-based method DoCoM (Yau & Wai, 2022), we found that our methods significantly outperform DoCoM; see Figures 9 and 10 in Appendix E.

Below we compare the convergence rate of our second method DaSHCo with four recent notable decentralized algorithms, along with the difference of their assumptions.

- SQuARM-SGD (Singh et al., 2021) achieves the same convergence rate as DaSHCo. But unlike DaSHCo, SQuARM-SGD does not use the gradient tracking technique. It requires either a bounded gradient dissimilarity assumption or a bounded gradient assumption, which is stronger than our assumption for analyzing DaSHCo.

- DoCoM (Yau & Wai, 2022) achieves better sample complexity than ours. It relies on the VR technique and requires the so-called mean-squared smoothness assumption, which is stronger than our assumption on smoothness of the population function. Moreover, VR techniques can yield poor generalization performance, as we observed from Figures 9 and 10 in Appendix E.

- Cedas (Huang & Pu, 2024) achieves the same convergence rate as our method DaSHCo in a nonconvex setting. But it makes stronger assumptions on the mixing matrix $W$, requiring $W$ to be positive semidefinite and symmetric, while we do not need these restrictive conditions.

- Islamov et al. (Islamov et al., 2024) give two methods: MoTEF and MoTEF-VR. MoTEF has the same convergence rate as our method DaSHCo but needs stronger assumption than ours in $W$. More precisely, it additionally needs $W$ to be symmetric. Similar to DoCoM, MoTEF-VR achieves the

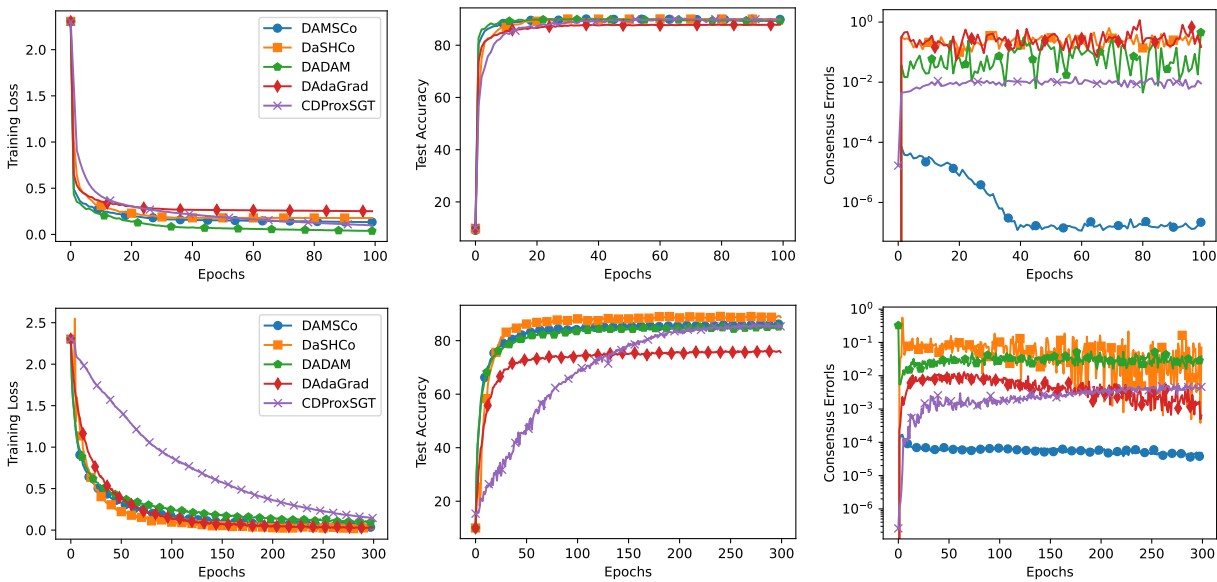

Figure 4: **Results with homogeneous data:** Plotted above are (from left to right) the training loss, test accuracy, and consensus error with respect to epoch for the FashionMNIST (top) and CIFAR-10 (bottom) datasets, comparing DAMSCo and DaSHCo with CDProxSGT (Yan et al., 2023), Distributed Adam (Nazari et al., 2022; Chen et al., 2023b), and Distributed AdaGrad (Duchi et al., 2011) with Top-$k$ compression.

optimal convergence rate, by utilizing the VR technique and assuming the stronger mean-squared smoothness assumption.

## E  Additional Numerical Plots

In this section, we first include our experimental results plotted with epochs as the x-axis instead of communication rounds. Figure 4 gives the results for homogeneous data and Figure 5 for heterogeneous data. We also include QSGD quantization (Alistarh et al., 2017) of compression instead of Top-$k$ in Figure 6. Here, we train the model on FashionMNIST with both heterogeneous and homogeneous data. We note that our proposed algorithms perform quite similarly to what we observed for Top-$k$ compression for the homogeneous case, while Distributed AdaGrad failed to generalize. For the heterogeneous data, we note that DaSHCo can successfully train the model and achieve the highest test accuracy, though convergence is slower. We suspect that 4-bit quantization is more aggressive than Top-$k(0.3)$ and thus leads to less competitive results.

Further, we include a comparison between grid and ring topologies for the communication network. Our primary experiments utilized a ring topology, and these additional experiments demonstrate the ability of our methods to generalize to additional network topologies. For the experiments given in Figure 7 and Figure 8, we run with 9 MPI ranks in a ring or a $3 \times 3$ grid on the FashionMNIST data. We run with homogeneous data in Figure 7 and heterogeneous data in Figure 8. For the heterogeneous case, we only perform training and testing on 9 label classes, with each MPI rank training on a single class. We note that the difference in results between the differing topologies is minimal for DAMSCo and DaSHCo, though there is an apparent difference in results with the higher number of MPI ranks. We further note that we observed instability with DADAM training with heterogeneous data and our test hyperparameters. Future work will focus on hyperparameter tuning for larger MPI ranks and a more in depth study of the effects of MPI rank counts, communication topology, and their interplay with the performance of the proposed algorithms.

Additionally, we include a comparison to the state-of-the-art DoCoM optimizer (Yau & Wai, 2023). We select DoCoM for comparison, as its theoretical convergence rate is tighter than ours and the other suggested SOTA methods, and the code and experimental data used by the authors are readily available. We directly use the hyperparameters the authors tuned for the LeNet5 network on FashionMNIST dataset, and replicate

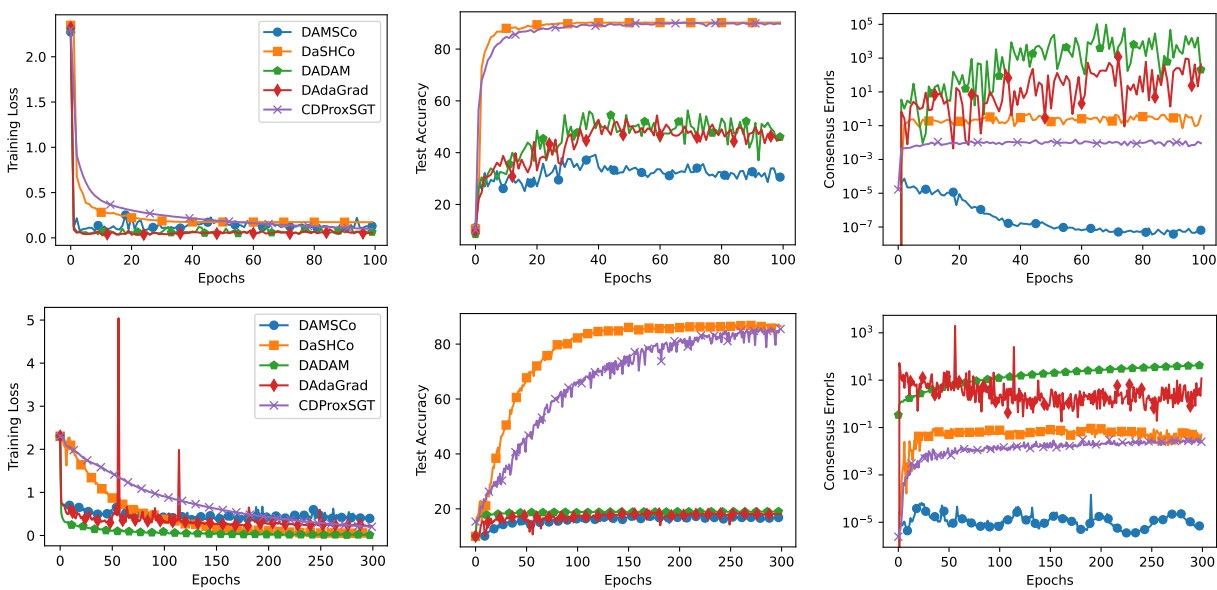

Figure 5: **Results with heterogeneous data:** Plotted above are (from left to right) the training loss, test accuracy, and consensus error with respect to epoch for the FashionMNIST (top) and CIFAR-10 (bottom) datasets, comparing DAMSCo and DaSHCo with CDProxSGT, Distributed Adam, and Distributed AdaGrad with Top-$k$ compression.

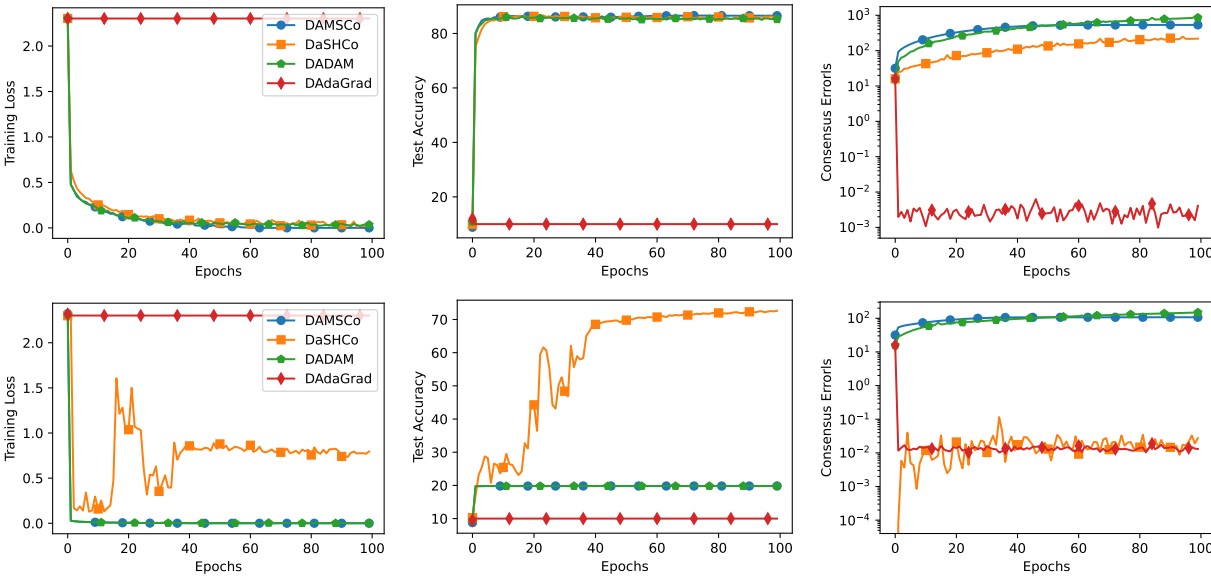

Figure 6: **Results with QSGD:** Plotted above are (from left to right) the training loss, test accuracy, and consensus error with respect to epoch for the FashionMNIST datasets using homogenous data (top) and heterogenous data (bottom), comparing DAMSCo and DaSHCo with Distributed Adam and Distributed AdaGrad with 4-bit QSGD compression.

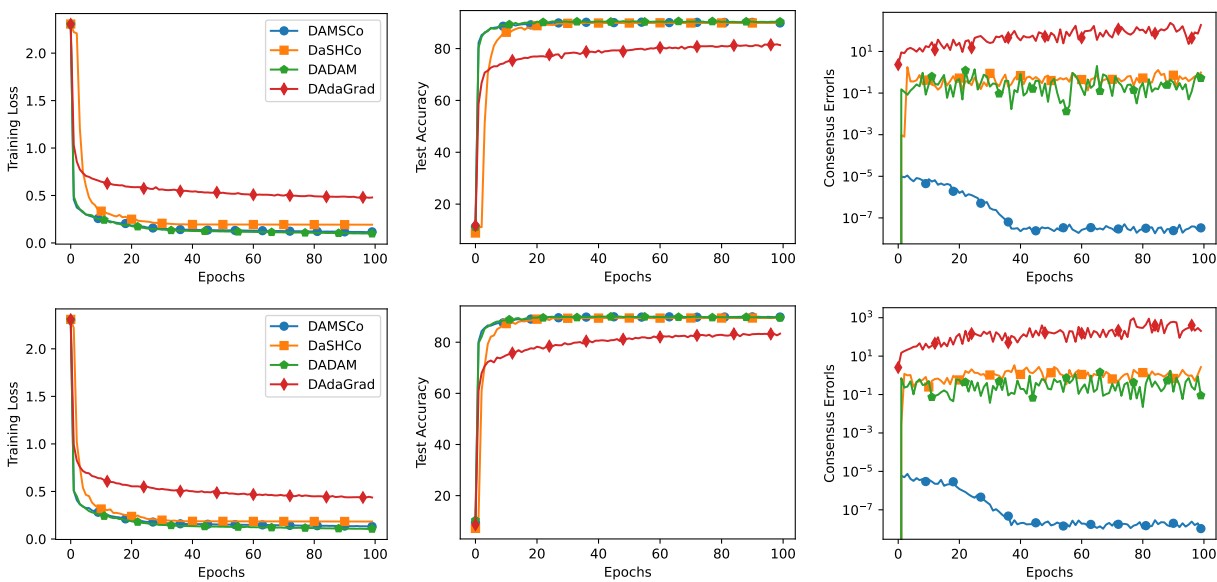

Figure 7: **Grid versus ring communication networks comparison with homogeneous data:** Plotted above are (from left to right) the training loss, test accuracy, and consensus error with respect to epoch for the FashionMNIST datasets running on Ring (top) and Grid (bottom) topologies for the communication network, comparing DAMSCo and DaSHCo with Distributed Adam and Distributed AdaGrad with Top-$k$ compression.

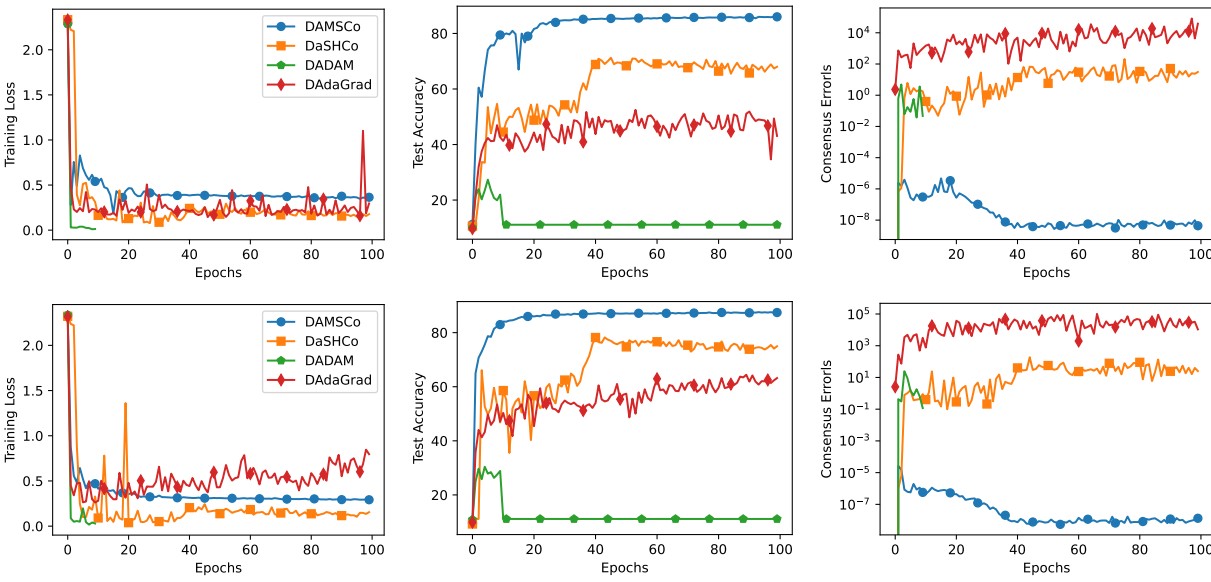

Figure 8: **Grid versus ring communication networks comparison with heterogeneous data:** Plotted above are (from left to right) the training loss, test accuracy, and consensus error with respect to epoch for the FashionMNIST datasets running on Ring (top) and Grid (bottom) topologies for the communication network, comparing DAMSCo and DaSHCo with Distributed Adam and Distributed AdaGrad with Top-$k$ compression.

the experiments from Figures 1, 2, and 3. We give these updated figures below in Figures 9, 10, and 11. A preliminary study suggests that hyper-parameter tuning with DoCoM by itself is unlikely to significantly improve upon the results presented here.

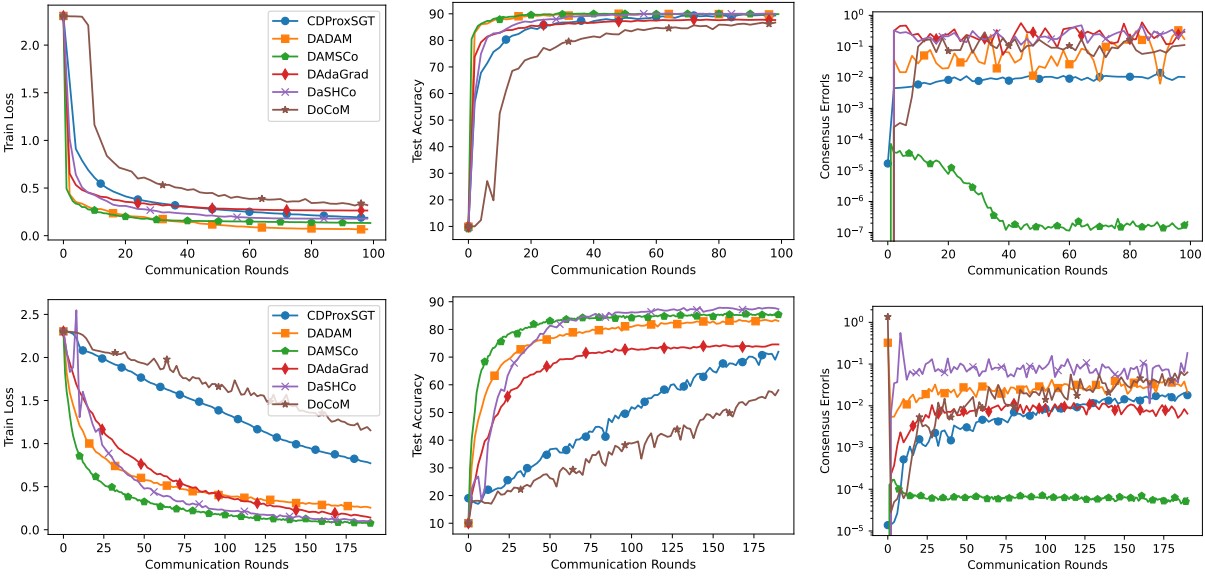

Figure 9: **Results with homogeneous data:** Plotted above are (from left to right) the training loss, test accuracy, and consensus error per communication round for the FashionMNIST on LeNet5 (top) and CIFAR-10 on Fixup-ResNet-20 (bottom) benchmarks, comparing DAMSCo and DaSHCo with DoCoM, CDProxSGT, Distributed AdaGrad, and Distributed Adam with Top-$k$ compression.

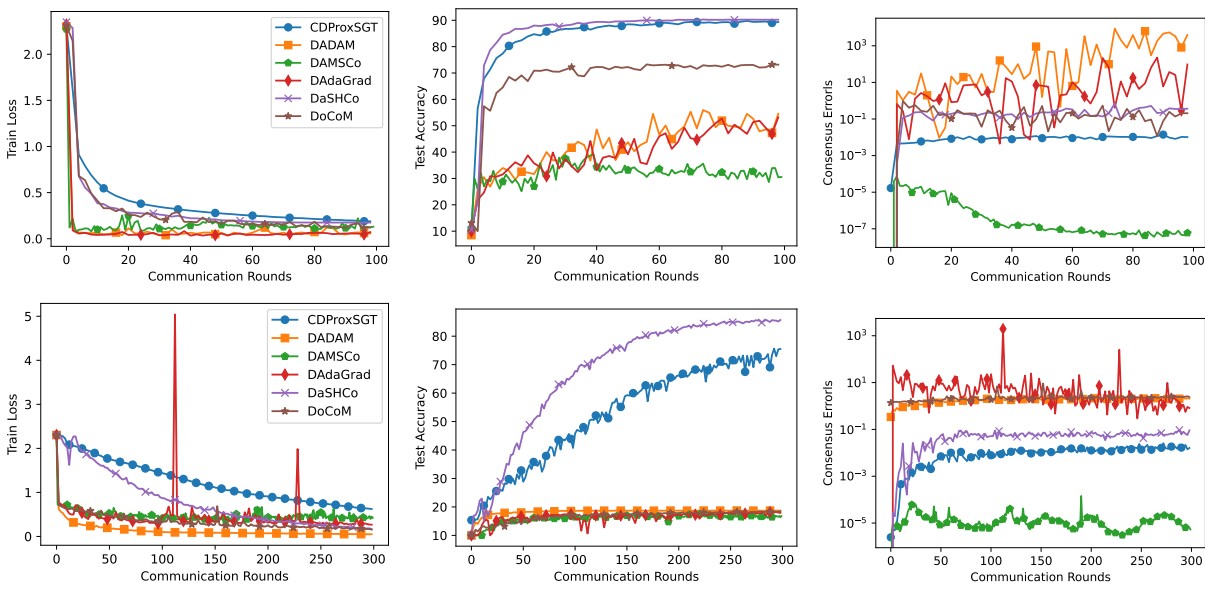

Figure 10: **Results with heterogeneous data:** Plotted above are (from left to right) the training loss, test accuracy, and consensus error per communication round for the FashionMNIST on LeNet5 (top) and CIFAR-10 on Fixup-ResNet-20 (bottom) benchmarks, comparing DAMSCo and DaSHCo with DoCoM, CDProxSGT, Distributed AdaGrad, and Distributed Adam with Top-$k$ compression.

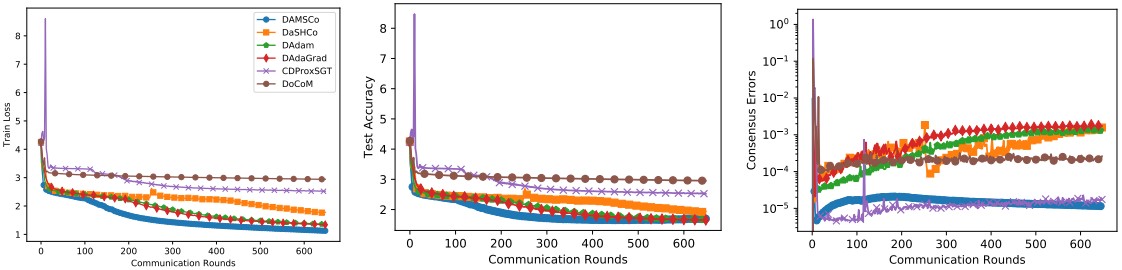

Figure 11: **GPT Results with homogeneous data:** Plotted above are (from left to right) the training loss, validation loss, and consensus error per communication round for the Shakespeare dataset, comparing DAMSCo and DaSHCo with DoCoM, CDProxSGT, Distributed AdaGrad, and Distributed Adam with Top-$k$ compression.

Lastly, we demonstrate a linear speedup for DAMSCo and DaSHCo, resulting from varying the number of agents to 5, 9, and 16 in a ring toplogy. We utilize the 5 and 9 agent results on FashionMNIST with homogeneous data, as given in Figures 4 and 7 and we include an additional experiment with 16 agents. These are plotted in Figure 12. We note that the close overlap of curves for loss and accuracy provides experimental validation of our theoretical result of linear speedup.

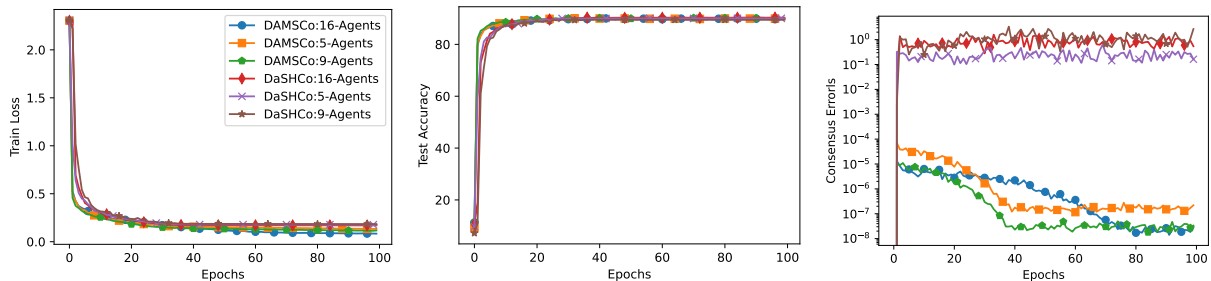

Figure 12: **Results demonstrating linear speedup:** Plotted above are (from left to right) the training loss, test accuracy, and consensus error per communication round for the FashionMNIST datasets on LeNet5, comparing DAMSCo and DaSHCo with 5, 9 and 16 agents in a ring topology.

## F   NanoGPT Qualitative Sample Results

In this section, we show the inference outputs of our NanoGPT LLMs trained on the tiny-shakespeare dataset. We provide the prompt "To be, or not to be?" for each model trained by each optimizer. The outputs are generated via the sampling procedure implemented in the NanoGPT code base (Andrej, 2022). For each optimizer, we generate 3 samples per prompt. We do no curation and give the first three outputs produced. These outputs can be seen in Figure 13, Figure 14, and Figure 15. We note that there is a similar performance between DaSHCo, DAMSCo, DAdaGrad, and DAdam, as the minimum loss achieved is very similar. Thus, the outputs produce relatively comprehensive and coherent english sentences. However, we do notice a difference in CDProxSGT, where the outputs produced are significantly less coherent. This falls in line with poorer generalization in the validation loss as seen in Figure 11.

```
To be, or not to be?

CAPULET:
Not with the constary: so much be stander-day
the change of his foul and to be determine,
And then I for a will our absence of your brother;
And that you were he will intend over-once
Than I will bring told under all the mind:
And what have they yet lived thy foot this good:
I will make me only son brother of the morning,
Then, in the soul's blood
That hands upon the rest.
Why, therefore I am forget.

CLIFFORD:
Here comes, you say and learn against you this?

First Lord:
What must I c
---------------
To be, or not to be?

QUEEN MARGARET:
You cannot learn her; I'll meet my heart.

JULIET:
I cannot speak to thee.

PAULINA:
Yes, my lord;
And though I live it should be desire moved:
And therefore I will be so;
And not he be seen to say, 'tis my suit:
Here is your foes, for a gentleman.

PAULINA:
And was so! why long will your flatter.

POLIXENES:
I shall be so:
My lord; I'll not be resolved of thee.

LEONTES:
O my gracious lady what thou canst be see
The babuse that I may for my will,---

PAULINA:
O much of the chi
---------------
To be, or not to be?
Therefore, we saw the field in my sweet scape.

LADY CAPULET:
Else your honourable seen as their sink
To whose finding the trumpets between child,
Were as bring tI have a child been as ince,
And I will blame a cursed for in my wife's my
hand; I think, I will thou forget me for my life.
Have I come to some fair of his crown,
And that am I be sit in my dressing you;
And then I will cruel upon my part, but I will.

LEONTES:
I never weep:
And I will not bee you to light:
They are but a lady, and ev
---------------
```

(a) DAMSCo

```
To be, or not to be?

CORIOLANUS:
O, behold!

CORIOLANUS:
More is taken them with hands: but he put of his back:
I'll be away, and hand, with to heavy some fortune.

All:
This is even well, I heard it the overture.

First Gentleman:
Well, no man it is my children and all the tyranny.

HERMIONE:
Alas, whom I sworn to kill a pass which is as word,
And in The shame-stone of the rest: and noble the son;
To have for thy fled, that fight doth of them;
Let's too discaladine and Edward's in child;
And your husband's for hi
---------------
To be, or not to be?

QUEEN ELIZABETH:
Here chance me! I have seen an many word.
Are you would I have no successed in the friends.

DUKE OF YORK:
My lord?

KING EDWARD IV:
Now would indeedst to the grace?

HENRY BOLINGBROKE:
What all you to loss as this good deserve,
To make heavy bear that vice and way so live.
That I may so welcome to speak:
We I say, well, and play'd in the eagly
Of this offices at ornignor of the king;
And wilt it good, both speedly like to the hand:
If being a chooice wounder for Gloucester's
---------------
To be, or not to be?
The conscience. This is the common with the view,
Than till of anon oath of his womb:
For we mine is for worthy in her borne to Warwick.
The scale brave Edward and haste and in close.

LADY ANNE:
The mind men, therefore his conscient of his gentleman
and deserve your Henry's mother; and therefore thee
day as the victory unto the life absence it is state,
And tradenure the noble hath been seems pillain,
Report to the bable honourable to the tongue.

KING RICHARD II:
Uncle, I am come on: make wel
---------------
```

(b) DaSHCo

Figure 13: **LLM Output:** Plotted above are the outputs of NanoGPT from the prompt "To be, or not to be?" comparing DAMSCo and DaSHCo trained on the tiny-shakespeare dataset with Top-$k$ compression.

```
To be, or not to be?

CLAUDIO:
Not with that we service.

LUCIO:
I am great thee, have been thy purpose and as die.

ISABELLA:
O heaven, I beseech you, faith, and be thee not
To be right to hang for Hereford to thee,
And give him by self: now, and let me no more.

Clown:
My word will, sir, I pray you, sir.

LUCIO:
My lord, I do but Clifford, and so bid him
As if well as we for us.

Clown:
What comes it now, from this Montague?

Provost:
Here is gone!

ANGELO:
What a man are for him? Knockes and you?

MISTRESS OVERD
---------------
To be, or not to be?

PETRUCHIO:
Why, son I was behold in an house, nor by me
new a man that such men and us bear.

Third Servant:
Faith, let me not.

TRANIO:
So see I do before thee, diest before thee?

First Gentleman:
Ay, my lord.

Second Gentleman:
You shall not, sir; and let him know your virtues
a known of that honour so like a crown,
and all suitors.

LUCIO:

ISABELLA:
Come, come, come, come; though it constant
appointo you;
of a comfort, sir, like a brow storm more of cols.

Clown:
Would you give me well.

---------------
To be, or not to be?
Therefore, be soften and wonder--what news
To have prevail'd your choacter should see,
To see the galland.

Shepherd:
For this is it so?

ISABELLA:
I do for wise I have a child, and as indeed
Men known our foul most report in the news.

DUKE VINCENTIO:
But you had all, and have releason made me livery
of me; for thence must I have but but a with him;
you would yet, though I will not be redeem and not
so before much about me: he must be will not call
you go with him: instantly.

BRUTUS:
What wer
---------------
```

(a) DADAM

```
To be, or not to be?

CORIOLANUS:
O, affection!

SICINIUS:
By Angelo?

ORCORIOLANUS:
My lord, I put him to be determined away.

ROMEO:
In am I must off the first contract;
And but I in entreation of Hermione to thee
Will be revenged well in a war.

VOLUMNIA:
I would banished, yet let me know thee gods
And the self-place of the golden is all,
And both be consul, as if he were return'd
In breathe so; some the fire-run leong,
That Prince I am trade, being by you:
Saddle the Earl of Wiltshire are no vex'd
The noble Nep
---------------
To be, or not to be?

QUEEN ELIZABETH:
He is that hath had ever his noble two liver
and that he is so noble counted of courtesy.

SICINIUS:
I would be thus worst: he did with all the maid
Made most grave their whole doubly company,
To the golden change their duke of it.

VIRGILIA:
Heaven as heaven, what said it is;
When he should be strived me so I say;
Not in him that I have seen your time
To time to our fellow upon my mother.
Thou art not born.

BUCKINGHAM:
No, my good lord.

BRUTUS:
If I patch you, God will in e
---------------
To be, or not to be?
Therefore, we may shall have kill'd me up
And here prove take all the war. Fie, we must be:
Think you means his body to your words; and yet,
And bear me false a wIn report in his land,
And yet me keep you me at my breather
And witness what you will, if that still fear,
Disprame us the spring thoughts, the base of York
Makes here a man age of it is conduct of thee;
And the noble hath been so much precious to the
lavely express made him to a tongue's fore you.

SICINIUS:
What is the mannerled wel
---------------
```

(b) DAdaGrad

Figure 14: **LLM Output:** Plotted aboveare the outputs of NanoGPT from the prompt "To be, or not to be?" comparing DADAM and DAdaGrad trained on the tiny-shakespeare dataset with Top-$k$ compression.

```
To be, or not to be?

THUSINRDY:
Aure aissthew y hellllll n amopetelives;
Pothy wou m thake o Windo wher eringh t ath dour wish eshire s poongower ore
Thak d nderurt f sor; irind m:

Than inle onthe se Prerd I Som.

HENRY ERD E:
Po I:
Shosal this ghest hoin ccur ayo teyo ryous chan t ce wi
---------------
To be, or not to be?

QUETAM:
And shirdse anot whe m sono anghy tou nours.

MARINA:
Thy shat su in soth llethend, wild
ch fre my shinth s l.

ERMENIO:
My wousthor edourd helllvet:
A d anot to oshink hed m tish:
Shed he be fe flator:
Whet Clo ghasundist, is duche n,
To me thall heatake bed hich und wan s s with sel ngond the weld nonch id,
This h isurd we.

MMEOLIXnGIUS:
I thity se thistou tised the n sen e sutan wiplyth ou whand nghitt chus.

ESBRTES:
Hall t thyou t prit.

BELORUS:
Myowout ir f an;
Tonknd inot
---------------
To be, or not to be?
Ther, now, we heald, I he shik; ghead n the is,
I spr thisand allon bay ho su andesen,
Thes thinds se te wofoingin ind tof ther We a wowhid chin blare aned hyou aInd
my a theint d polof the an benketind l menat m inor wesing brwimngise.

MELINIO:
But d dotho y willl wher mperel tshou!
A'NGTolllct ourjur'st o t wonck u thinour hes m sbe l itond chinous,
Prtr Whound the y shand fsurs:
Hend o ndone y nthe sthis t faban s hor mqu thit.

HEG mETESCK:
Thy, d, clige fincore yothes arr whollds e

---------------
```

(a) CDProxSGT

Figure 15: **LLM Output:** Plotted above is the output of NanoGPT from the prompt "To be, or not to be?" using CDProxSGT trained on the tiny-shakespeare dataset with Top-$k$ compression.

