# OpenReview forum: "Compressed Decentralized Momentum Stochastic Gradient Methods for Nonconvex Optimization"
_TMLR — Accepted by TMLR_

### Review · Reviewer_KRgU · 2025-04-19

**Summary Of Contributions:**

This paper integrates communication compression and momentum acceleration for decentralized stochastic non-convex optimization and proposes two algorithms, one for the homogeneous and the other for the heterogeneous setting. The paper theoretically analyzes their convergence rates, which are shown to match the known lower bound of stochastic nonconvex optimization, and further achieve linear speedup with respect to the number of machines in the decentralized setting. Empirical studies are also conducted to validate the effectiveness in reducing the communication rounds needed for convergence.

**Audience:**

Yes

**Broader Impact Concerns:**

No broader impact is of concern.

**Claims And Evidence:**

Yes

**Requested Changes:**

1. Can the authors elaborate on the detailed technical challenge of integrating communication compression techniques with momentum acceleration?
2. Can the authors include, perhaps in the appendix, an explicit term-by-term comparison of the theoretical results with the previous baselines mentioned in Table 1, and elaborate on which terms popped up / vanished / changed due to the additional techniques applied, e.g., momentum, compressed communication?
3. Can the authors provide some experiments for the performance on different numbers of machines to illustrate, e.g., the linear speedup?

**Strengths And Weaknesses:**

**Strengths:**
1. The paper integrates the communication compression techniques in decentralized optimization and momentum acceleration for stochastic nonconvex optimization to reduce the communication complexity and accelerate convergence.
2. Both homogeneous and heterogeneous settings are studied.
3. Both theoretical analysis and empirical validation are provided, which are technically solid.

**Weaknesses:**
1. The proposed approaches are based on integrating existing methods for accelerating decentralized stochastic non-convex optimization, e.g., the DADAM line of work and those with compressed communication to reduce the communication cost, making the results seem slightly incremental.
2. Even though theoretical analyses are provided for both algorithms proposed, and the difference in settings is listed in Table 1, it's unclear how the theoretical result compares with previous methods on which the proposed methods build/improve. Meeting the stochastic non-convex $\mathcal{O}(\epsilon^{-2})$ lower bound in terms of gradient norm minimization is quite standard in recent stochastic non-convex analysis, which is not by itself informative of the acceleration techniques applied, and the linear speedup is also standard in decentralized optimization, which is not informative of the compression techniques applied.

---

> ### Author Response · Authors · 2025-06-10
> **Response to Weakness and Requested Changes by Reviewer KRgU**
>
> $\textbf{Weakness 1.}$
>
> We understand the reviewer’s concern regarding the perceived incremental nature of our work. While it is true that we build upon existing techniques such as ADAM and compression, our paper  resolves the core analytical challenge introduced by the interaction of compressed communication and adaptive gradient updates: their coupling makes it difficult to upper bound the consensus error and stationarity violation. The difficulty is further intensified in a decentralized scenario (i.e., without a central server to perform global average or sum). Our main technical innovation lies in carefully controlling these error terms to preserve both communication efficiency and convergence guarantees. We will emphasize the challenges of analysis more clearly in the revised version to highlight the novelty of our contribution.
>
> $\textbf{Weakness 2.}$
>
> By Table 1, we would like to highlight the important techniques adopted by each method. But we agree that Table 1 does not fully clarify the theoretical improvements of our work over existing ones. We will include more details to compare the theoretical results of different methods term by term; please refer to our response to Requested Change 2.
>
> Although meeting the standard
> $O(\epsilon^{-2})$ convergence rate is expected in stochastic nonconvex optimization, it is highly nontrivial and unknown for a method that combines several important techniques including compressed communication, adaptive gradient update, and decentralized update. As we responded to Weakness 2, our contribution lies in tackling the significant challenges.
>
>
> $\textbf{Requested Change 1.}$ Elaboration on the Technical Challenge of Integrating Communication Compression and Momentum Acceleration
>
> Thanks for the suggestion. The coupling of compressed communication and momentum acceleration or adaptive gradient update makes it difficult to upper bound the consensus error and stationarity violation. The difficulty is further intensified in a decentralized scenario (i.e., without a central server to perform global average or sum). In our analysis, we design Lyapunov functions that carefully decompose the error into compression error, consensus error, and stochastic error, and we balance these using tailored stepsizes and mixing parameters (e.g., see Theorems 4.1 and 4.3). We will add these discussions in Section 3.1 to elaborate the technical challenge.
>
> $\textbf{Requested Change 2.}$ Explicit Term-by-Term Comparison
>
> We appreciate this suggestion. We will add in the appendix a term-by-term comparison of our theoretical results with those from the baseline methods in Table 1. Specifically, we highlight the terms in our convergence rate bound that reflect different algorithmic properties. The term exhibiting a $\frac{1}{n}$ relationship explicitly demonstrates a linear speedup with respect to the number of workers. The term involving $\rho$ quantifies the dependence of convergence on the network topology, indicating the algorithm’s robustness to different graph structures. The term involving $\eta$ captures the impact of communication compression, revealing how lower communication overhead can accelerate convergence. Finally, we discuss the role of momentum acceleration in our numerical experiments, showing its effect on empirical performance.
>
> $\textbf{Requested Change 3.}$ Experiments with Different Numbers of Machines
>
> We appreciate the suggestion to show linear speedup. In the current manuscript, we actually have some limited numerical results to demonstrate the linear speedup. In top row of Figure 4, we used n=5 agents, and in top row of Figure 7, we used n=9 agents on training the same neural network by the same training data. The curves in terms of epoch by our methods are almost the same. This indicates our methods can indeed achieve linear speed up. As suggested, we will add more numerical results to verify linear speed up. We also include limited additional experimental validation with plots using n=16 agents in the linked anonymous repository: https://anonymous.4open.science/r/TMLR-B8F6/Referee_plots.pdf

---

### Review · Reviewer_66WN · 2025-05-10

**Summary Of Contributions:**

The paper introduces two methods for decentralized nonconvex optimization that use momentum and reduce communication costs through compression. The first method, DAMSCo, is a decentralized version of AMSGrad combined with compression. It works under the assumption of bounded sample gradients. The authors show that it achieves optimal iteration complexity. However, this assumption implies that clients have similar data, which often doesn’t hold in real-world applications like federated learning.

To address the heterogeneous case, the authors propose a second method, DaSHCo. It is a decentralized heavy-ball method with gradient tracking, designed to handle data heterogeneity without requiring bounded gradients. They also achieve the optimal convergence rate for this method.

**Audience:**

Yes

**Claims And Evidence:**

Yes

**Requested Changes:**

A Future Work section would be a nice addition.

**Strengths And Weaknesses:**

Strengths:
- The paper is well written and easy to follow.
- It presents a general framework that combines compression, momentum, and decentralization.
- It includes strong theoretical results, with optimal convergence rates.


Question:

Is it possible to relax the bounded gradient assumption for DAMSCo?

---

> ### Author Response · Authors · 2025-06-10
> **Response to Question and Requested Changes by Reviewer 66WN**
>
> We appreciate this insightful question. In our current theoretical analysis of Algorithm 1 (named as DAMSCo), we rely on the bounded gradient assumption to bound the consensus error and ensure convergence guarantees under nonconvex settings with compression. Though there are papers relaxing the bounded gradient assumption for analyzing Adam in a $\textbf{non-distributed}$ setting, we could not find any paper in the literature that does not require bounded gradient to analyze Adam-type methods in a decentralized setting. We believe the challenge is caused by the interaction of adaptive gradient update and decentralized communication. For our case, the challenge becomes more significant because of the compressed communication. Let us elaborate it in more detail below.
>
> $\textbf{1. Key Technical Obstacle in Theorem 4.1.}$ We establish convergence in Theorem 4.1 by constructing a Lyapunov function that couples the consensus error and the adaptive learning rate terms. We heavily use the boundedness of the gradients to control the error terms that arise from the decentralization and the compression operator. Without the bounded gradient assumption, we could not control these error terms effectively. In particular, the gradient error term in the left hand side of  Lemma A.6 relies on bounding the gradient magnitude to ensure that the expected descent in the objective dominates the accumulated error.
>
>
> $\textbf{2. Why Algorithm 2 only needs a weaker assumption.}$ Algorithm 2 does not need the gradient boundedness assumption for convergence guarantee because it adopts a gradient tracking step to control the error in the gradient estimates, at the cost of one more round of communication per update than Algorithm 1. Gradient tracking is a powerful technique that helps agents estimate the global gradient, thereby mitigating the variance that arises from decentralization and stochasticity.
>
> $\textbf{Potential future exploration:}$ We recognize these theoretical difficulties and agree that extending DAMSCo’s analysis to relax the assumption of unbounded gradients is a valuable direction for future work. To address this issue, we plan to use the mean-variance bound assumption in Assumption 5 of our paper and explore modifying DAMSCo by either tracking the stochastic gradient or tracking the second momentum term. This way can potentially weaken the assumption but requires one more round of communication per update. Hence, we plan to make the exploration for the case with heterogeneous data. We will revise our paper by explicitly mentioning these challenges in the discussion section and outline the potential explorations in a Future Work section.

---

### Review · Reviewer_W3sM · 2025-05-30

**Summary Of Contributions:**

This paper introduces two decentralized algorithms, DAMSCo (Decentralized AMSGrad with Compressed Communication) and DaSHCo (Decentralized Stochastic Heavy-ball with Compressed Communication), for nonconvex decentralized optimization with efficient communication. In particular,
- DAMSCo is an adaptive method (AMSGrad-style) with compression and is analyzed under the bounded gradients condition. The authors claim this is the first decentralized adaptive stochastic gradient method with compressed communication.
- DaSHCo employs a heavy-ball momentum technique with gradient tracking and compression, aimed at scenarios with data heterogeneity where gradients may not be bounded.

Both algorithms are claimed to achieve an optimal convergence rate of $O(1/\sqrt{T})$ and can attain linear speedup with topology-independent property under certain conditions. The paper provides theoretical analyses and empirical results on DNNs and Transformers.

**Audience:**

Yes

**Broader Impact Concerns:**

I do not identify any specific broader impact concerns that would necessitate a Broader Impact Statement beyond those typical for general optimization algorithms.

**Claims And Evidence:**

No

**Requested Changes:**

- "The authors should provide a comprehensive discussion and theoretical comparison of DaMSCo and DaSHCo with the relevant works mentioned in the weaknesses part, highlighting why the proposed methods offer advantages over existing approaches. In addition, a discussion on the underlying assumptions and the novelty of the theoretical analysis is necessary to better position this work.
- Following the previous point, the authors should elaborate more on the DaSHCo and DAMSCo algorithms. For now, I do not find any explanation right after Algorithms 1 and 2.
- Could the authors explain why the compressed gradient $\underline{g}_i$ of each agent $i$ needs to be shared in the DaSHCo algorithm, but not in DAMSCo? Is it possible to eliminate this additional communication burden in the DaSHCo algorithm?
- "In the simulation, beyond evaluating performance based on metric versus communication rounds, could the authors also compare their algorithms with SOTA methods in terms of communication costs, ensuring that the exchange of auxiliary parameters is also accounted for?

**Strengths And Weaknesses:**

## Strengths

The experiments on LeNet5, Fixup-ResNet-20, and NanoGPT demonstrate the performance of DAMSCo on homogeneous data and DaSHCo on heterogeneous data against several baselines.

## Weaknesses
- The rate of convergence in this work is not tight enough, i.e., $O(1/\sqrt{T})$, while some recent work such as [Yau and Wai, 2023] leads to near-optimal $O(T^{-2/3})$ rate.
- The paper fails to discuss and compare its contributions against several highly relevant works that also address decentralized optimization with momentum / adaptive gradient and compression, e.g., [Singh et al., 2020], [Yau and Wai, 2023], [Huang and Pu, 2024], [Islamov et al., 2025]. This makes it difficult to assess the precise novelty and significance of the proposed methods.

N. Singh, D. Data, J. George, and S. Diggavi. "SQuARM-SGD: Communication-efficient momentum SGD for decentralized optimization." IEEE JSAIT, 2021.

C. Y. Yau, and H. T. Wai. "DoCoM: Compressed Decentralized Optimization with Near-Optimal Sample Complexity." TMLR, 2023.

K. Huang, and S. Pu. "Cedas: A compressed decentralized stochastic gradient method with improved convergence." IEEE TAC, 2024.

R. Islamov, Y. Gao, and S. U. Stich. "Towards Faster Decentralized Stochastic Optimization with Communication Compression." ICLR, 2025.

---

> ### Author Response · Authors · 2025-06-13
> **Response to Question and Requested Changes by Reviewer W3sM**
>
> $\textbf{Weakness 1}.$
>
> We acknowledge the reviewer’s concern that our current theoretical convergence rate is $O(1/\sqrt{T})$, while some recent works achieve a faster rate of $O(T^{-2/3})$. We clarify the following:
>
> 1. Our first method DAMSCo is designed with $\textit{adaptive gradient updates}$, i.e., Adam-type updates. Its convergence rate is $O(1/\sqrt{T})$ and aligns with the rates established for existing (nondistributed or distributed) stochastic nonconvex optimization algorithms that use Adam-type updates.
>
> 2. For momentum-based algorithms, a few recent works enhance theoretical convergence rates by incorporating variance reduction (VR) techniques. However, we observe that, in practice, VR-based methods can exhibit unstable performance or even degrade generalization, particularly in training large-scale deep neural networks (DNNs). Specifically, through additional experiments comparing our algorithms DAMSCo and DaSHCo against the VR-based method DoCoM, we found that our methods significantly outperform DoCoM in training relatively large-scale models, including LeNet5, Fixup-ResNet-20, and NanoGPT. The detailed numerical results demonstrating this advantage are available at: https://anonymous.4open.science/r/TMLR-B8F6/Referee_plots.pdf.
>
> $\textbf{Weakness 2}.$
>
> Compared to the four mentioned works, our first method DAMSCo effectively combines Adam-type update with compressed communication to achieve fast convergence. In contrast, none of these methods incorporate Adam-type updates. We provide a detailed comparison of each method against our second method DaSHCo.
>
> 1. SQuARM-SGD achieves the same convergence rate as DaSHCo. But unlike DaSHCo, SQuARM-SGD does not use the gradient tracking technique. It requires either a bounded gradient dissimilarity assumption or a bounded gradient assumption, which is stronger than our assumption for analyzing DaSHCo.
>
> 2. DoCoM achieves better sample complexity than ours. But it relies on VR techniques and requires the so-called mean-squared smoothness assumption, which is stronger than our assumption on smoothness of the population function. Moreover, VR techniques can be less effective in large-scale experiments, as we observed in the additional numerical results available at https://anonymous.4open.science/r/TMLR-B8F6/Referee_plots.pdf.
>
> 3. Cedas achieves the same convergence rate as our method DaSHCo in a nonconvex setting. But it makes stronger assumptions. More precisely, it assumes the mixing matrix $W$ is positive semidefinite and symmetric, while we do not need these restrictive conditions.
>
> 4. Islamov et al. give two methods: MoTEF and MoTEF-VR. MoTEF has the same convergence rate as our method DaSHCo but needs stronger assumption than ours in $W$. More precisely, it additionally needs $W$ to be symmetric. Similar to DoCoM, MoTEF-VR achieves the optimal convergence rate. It relies on VR techniques and needs the stronger mean-squared smoothness assumption. Though we have yet compared to MoTEF-VR, we observed that VR techniques can be less effective in large-scale experiments.
>
>
> $\textbf{Change 1}.$
> We refer the reviewer to our response to the weakness part.
>
> $\textbf{Change 2}.$
> We agree that not many explanations are provided under Alg. 1 and 2. We will revise the paper by adding more detailed descriptions of the key algorithmic components. Specifically, Line 5-6 in Alg. 1 follow AMSGrad and perform local update to the first and second momentum. Line 7 performs a local update to the model; $\underline{\mathbf{x}}_i$ is used to estimate the local model; we compress the estimate error. Line 8 performs a neighbor communication, which can be realized through communicating the compressed vector, as we explained in the third paragraph under Alg. 1.
>
> In Alg. 2, we communicate not only model but also stochastic gradient. Line 5 uses the gradient tracking technique to handle data heterogeneity. Similar to the model compression, we use $\underline{\mathbf{g}}_i^t$ as an estimate of $\mathbf{g}_i^{t-\frac{1}{2}}$ and compress the estimate error. Line 6 mixes local gradients and updates the first momentum. Line 7 performs the local update to model by using the first momentum term and then compresses the model estimate error. Line 8 is similar to Line 8 in Alg. 1.
>
> $\textbf{Change 3}.$
> We emphasize that the design choice in DaSHCo helps control the impact of possibly unbounded gradients by allowing agents to track the global gradient. If the gradients are bounded, DaSHCo will not need communicating gradients to have convergence. We will revise the paper to clarify this design, highlighting that communicating gradients in DaSHCo is motivated by the need to control errors without assuming bounded gradient.
>
> $\textbf{Change 4}.$
> We appreciate the suggestion. In addition to the communication rounds vs. performance, we will show curves in terms of the total communication cost (e.g., total bits), accounting for the transmission of auxiliary parameters.

---

### Decision · Action_Editor_VjSi · 2025-07-09

**Recommendation:** Accept with minor revision

**Additional Comments:**

The authors should incorporate the term-by-term theoretical analysis and additional experiments from the rebuttal into the revised version. The contributions are notable, especially the detailed analysis regarding communication compression and momentum acceleration in the decentralized setting.

**Audience:**

Yes

**Audience Explanation:**

Yes, I believe that many individuals in TMLR's audience would find the findings of this paper quite relevant and engaging. The methods proposed, particularly in the context of decentralized training, address important challenges in the field and offer practical solutions that could benefit researchers and practitioners alike. The clarity and depth of the analysis, combined with the timely nature of the topic, make it likely that the insights presented will resonate with a wide range of readers interested in machine learning and optimization techniques.

**Claims And Evidence:**

Yes

**Claims Explanation:**

The claims made in the submission are well-supported by a solid combination of theoretical analysis and empirical evidence. The authors have effectively demonstrated the performance of their methods through comprehensive experiments, addressing potential concerns raised by reviewers. Their detailed responses in the rebuttal further clarified any ambiguities, showcasing the practical implications of their work. Overall, the evidence presented convincingly substantiates their claims, making a strong case for the contributions they are making in the field of decentralized optimization.